# Autonomous transposons tune their sequences to ensure somatic suppression

İbrahim Avşar Ilık[1,6], Petar Glažar[1,6], Kevin Tse[2], Björn Brändl[3,4], David Meierhofer[5], Franz-Josef Müller[3,4], Zachary D. Smith[2] & Tuğçe Aktaş[1✉]

Transposable elements (TEs) are a major constituent of human genes, occupying approximately half of the intronic space. During pre-messenger RNA synthesis, intronic TEs are transcribed along with their host genes but rarely contribute to the final mRNA product because they are spliced out together with the intron and rapidly degraded. Paradoxically, TEs are an abundant source of RNA-processing signals through which they can create new introns[1], and also functional[2] or non-functional chimeric transcripts[3]. The rarity of these events implies the existence of a resilient splicing code that is able to suppress TE exonization without compromising host pre-mRNA processing. Here we show that SAFB proteins protect genome integrity by preventing retrotransposition of L1 elements while maintaining splicing integrity, via prevention of the exonization of previously integrated TEs. This unique dual role is possible because of L1's conserved adenosine-rich coding sequences that are bound by SAFB proteins. The suppressive activity of SAFB extends to tissue-specific, giant protein-coding cassette exons, nested genes and Tigger DNA transposons. Moreover, SAFB also suppresses LTR/ERV elements in species in which they are still active, such as mice and flies. A significant subset of splicing events suppressed by SAFB in somatic cells are activated in the testis, coinciding with low SAFB expression in postmeiotic spermatids. Reminiscent of the division of labour between innate and adaptive immune systems that fight external pathogens, our results uncover SAFB proteins as an RNA-based, pattern-guided, non-adaptive defence system against TEs in the soma, complementing the RNA-based, adaptive Piwi-interacting RNA pathway of the germline.

Transposable elements (TEs) are genomic parasites of virtually all living organisms[4–6]. Continuous TE activity threatens the integrity of genes and genomes, necessitating strategies to silence their activity. Despite extensive strategies to suppress TE expression and propagation[7], at least 40% of the extant human genome consists of TE-derived DNA, a fraction that appears to be increasing with active expansion of L1, *Alu* and the SVA family of retrotransposons and the creation of polymorphic insertions in the human population[8]. When these insertions land within genes they can affect the expression of the host gene by either altering the epigenetic landscape of the locus or interfering with post-transcriptional RNA processing[9]. Owing to the relative simplicity of splice-site sequences, intronic TE insertions can create alternatively spliced exons in both healthy tissues[10] and cancer[3], leading to 'exonization' of TEs. The vast majority of intronic TEs are, however, never exonized, suggesting that nuclear factors are generally able to discriminate against splice sites in TEs while maintaining the 'splicing code'.

To determine new factors that regulate the splicing of intronic TEs in the human genome we identified binding sites of 33 RNA-binding proteins (RBPs) using FLASH in HEK293 cells[11], with a focus on SR/SR-like and hnRNP proteins, which play important roles in the promotion and suppression of splicing, respectively (Extended Data Fig. 1a). Our data show high correlation between replicates (Supplementary Fig. 2) and grouping of proteins with similar architecture (Supplementary Figs. 2 and 3).

To study shared and unique RBP sites in this complex dataset we compiled all sites bound by any FLASH-profiled proteins into a unified peak file. We then counted FLASH reads from each RBP profile against these unified peaks, resulting in a count matrix of 135,891 (total number of peaks) × 72 (33 RBPs, three controls, two replicates each). We projected this matrix onto a two-dimensional plane using uniform manifold approximation and projection (UMAP)[12] (Fig. 1) and identified clusters with the HDBSCAN[13] algorithm (Extended Data Fig. 1b). We identified two side-by-side clusters that were enriched with peaks mapping to *Alu* retrotransposons, which are non-autonomous short interspersed nuclear elements (SINEs) that are mobilized by the L1 machinery. Curiously, cluster 2 is specifically enriched for antisense *Alu* elements, which are inserted on the strand opposite the host gene,

[1]Otto Warburg Laboratories, Max Planck Institute for Molecular Genetics, Berlin, Germany. [2]Department of Genetics, Yale Stem Cell Center, Yale School of Medicine, New Haven, CT, USA. [3]Universitätsklinikum Schleswig-Holstein Campus Kiel, Zentrum für Integrative Psychiatrie, Kiel, Germany. [4]Department of Genome Regulation, Max Planck Institute for Molecular Genetics, Berlin, Germany. [5]Mass Spectrometry Joint Facilities Scientific Service, Max Planck Institute for Molecular Genetics, Berlin, Germany. [6]These authors contributed equally: İbrahim Avşar Ilık, Petar Glažar. ✉e-mail: aktas@molgen.mpg.de

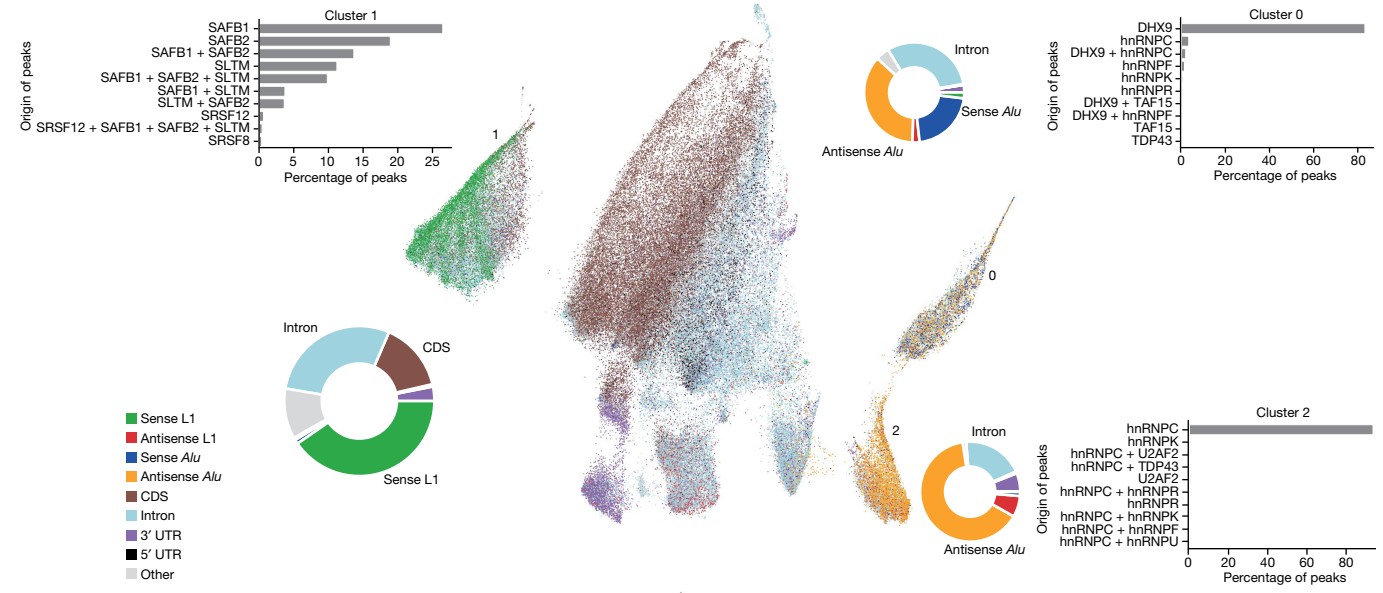

**Fig. 1 | FLASH screen of 33 RBPs showing sequence and structure determinants of RNA–protein interactions.** UMAP representation of all FLASH data. Each dot represents a peak identified in one or more proteins profiled by FLASH (total number of peaks, 135,891). UTR, untranslated region.

whereas cluster 0 contains a mixture of sense and antisense *Alu* insertions (Fig. 1, bottom right). Analysis of the peaks showed that roughly 93% of cluster 2 peaks originated from hnRNPC whereas around 83% of cluster 0 peaks originated from DHX9, with a smaller contribution of hnRNPC (3%, a further 2% are peaks shared by DHX9 and hnRNPC). Interestingly, previous work has shown that hnRNPC interacts exclusively with single-stranded, uracil-rich segments of antisense *Alu* insertions[14] whereas DHX9 interacts with long, double-stranded RNA formed by dense *Alu* insertions on opposite strands[15], suggesting that, even though they target the same transposon families, their respective targeting is heavily influenced by genomic context, transcriptional direction and secondary structure. Notably, all of these features were captured by our FLASH peak clustering without previous assumptions about the targets and binding modes of these proteins (Fig. 1). These observations indicate that our analysis captures biological insights behind the binding data generated by FLASH.

## SAFB proteins bind to L1 RNA

Encouraged by these results, we measured TE enrichment in each cluster and found that cluster 1 was specifically enriched for sense L1 RNA (Fig. 1, left). The peaks from cluster 1 originate mostly from three SR-like proteins with ER-type repeats—Scaffold Attachment Factor B1 (SAFB1), SAFB2 and SAFB-like transcriptional modulator (SLTM)—the three proteins that constitute the SAFB protein family in mammals (Extended Data Fig. 1a; roughly 87% of the peaks), with a small contribution of SRSF12 (around 1%), a testis-enriched SR protein of unknown function[16] (Fig. 1). SAFB1, SAFB2 and SLTM are characterized by a DNA-binding SAP domain at the N terminus and an RNA-binding RRM domain in the middle, followed by an ER-rich repeat at the C terminus (Extended Data Fig. 1a). The similarity in molecular architecture and sequence is reflected in the binding patterns, showing a large degree of overlap in the targets of the SAFB family (Fig. 2a,b). Consistent with the UMAP representation, sense L1 RNA is the TE most enriched in SAFB data (Fig. 2a,c). Surprisingly, we also detected an enrichment of sense Tigger DNA transposons (Fig. 2b,c), which are functionally extinct for at least 40 million years[17]. Both antisense L1 and antisense Tigger elements were specifically depleted from SAFB peaks, pointing to a shared sequence pattern within the sense strand of both transposons

that is probably depleted from the antisense strand (Fig. 2c and Supplementary Fig. 4a,b). FLASH coverage over strand-separated elements of L1 (Extended Data Fig. 2) or Tigger (Extended Data Fig. 3) confirmed that SAFB proteins indeed bind to L1 and Tigger repeats inserted on the same strand as the host gene and avoid antisense insertions. We verified these interactions with RNA immunoprecipitation–quantitative PCR (RIP–qPCR) using a monoclonal antibody against SAFB1 in HEK293 cells (Extended Data Fig. 4a).

## SAFB prevents L1 retrotransposition

Because L1 elements are the only autonomous transposons currently active in the human genome, interactions with sense L1 RNA can have implications beyond RNA processing of host genes if their binding partners affect the life cycle of L1 transposons. All three SAFB proteins showed binding preference for somewhat unfragmented, long L1 elements (Fig. 2d) and coverage almost exclusive to the coding segments of L1 repeats ORF1p and ORF2p (Supplementary Fig. 4a). As such, we tested whether SAFB proteins are involved in suppression of L1 retrotransposition. Using a luciferase-based assay utilizing a pCAG-driven L1 element[18] (Fig. 2e, top), we observed that individual depletion of SAFB1 increased retrotransposition efficiency whereas SAFB2 and SLTM had little or no effect (Fig. 2e). Next, we depleted all three SAFB proteins simultaneously to determine whether they might have redundant functions in the regulation of L1 retrotransposition. Indeed, the triple knockdown (referred to as SAFB KD hereafter) led to a much higher increase in L1 retrotransposition efficiency compared with single depletions (Fig. 2e). We next investigated the fate of L1 RNA using RNA fluorescence in situ hybridization (RNA–FISH) without the overexpression of a reporter construct. In control small interfering RNA-treated cells, L1 RNA was found to be strictly nuclear (Fig. 2f; and Extended Data Fig. 4g for probe specificity). By contrast, SAFB-depleted cells showed a marked increase in cytoplasmic L1 RNA (Fig. 2f), suggesting that SAFB proteins bind to and retain L1 RNA in the nucleus, thereby preventing their retrotransposition into new genetic loci.

The coding segments of L1 RNAs bound by SAFB proteins are significantly adenine rich in mammals and can be altered by optimization of codon sequences towards a higher guanine:cytosine (GC) content, as conducted for the hyperactive L1 variant ORFeus (Supplementary

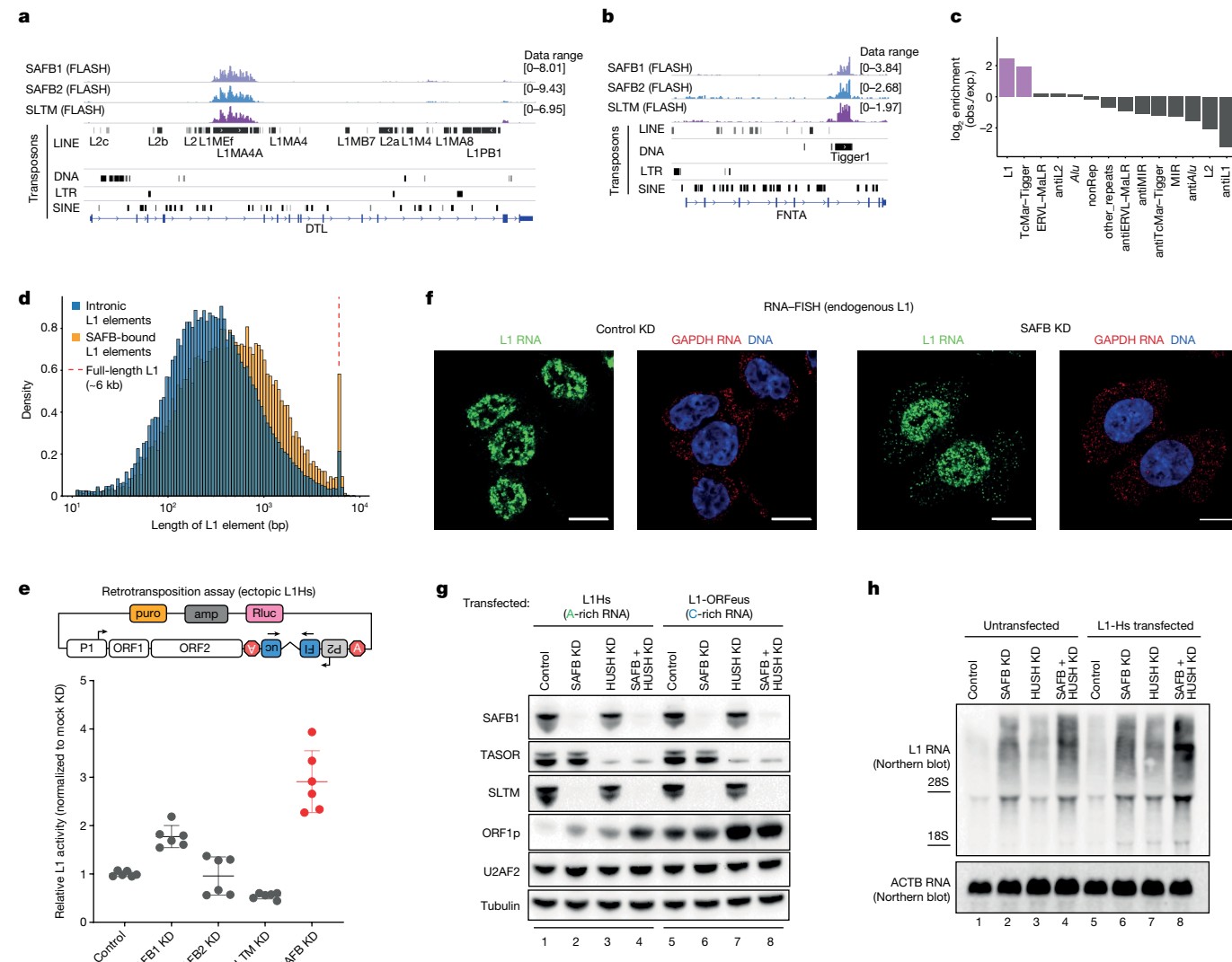

**Fig. 2 | SAFB proteins bind L1 elements and prevent their retrotransposition.** **a**, Integrated Genomics Viewer (IGV) snapshot of the gene *DTL* showing extensive binding of SAFB1, SAFB2 and SLTM to a 5,461 bp L1MA4 retrotransposon inserted on the same strand as the host gene. **b**, IGV snapshot of the gene *FNTA* showing extensive binding of SAFB1, SAFB2 and SLTM to a 2,307 bp Tigger1 DNA transposon inserted on the same strand as the host gene. **c**, Enrichment and depletion of TEs in SAFB peaks ($n = 23,136$) relative to all peak-hosting genes ($n = 8,881$). obs., observed; exp., expected. **d**, Length distribution of SAFB-bound L1 elements (orange, $n = 28,734$) compared with all intronic L1 elements (blue, $n = 1,001,410$). **e**, Luciferase-based L1 retrotransposition assay carried out in HeLa cells. The plasmid used for the assay is depicted above. L1 expression is driven by a pCAG promoter. Error bars show s.d. of six data points from two biological replicates carried out in technical triplicates (SAFB KD, simultaneous depletion of SAFB1, SAFB2 and SLTM). **f**, RNA–FISH in HCT116 cells in control versus SAFB depletion (SAFB KD, simultaneous depletion of SAFB1 and SLTM; HCT116 cells do not express SAFB2). Scale bar, 10 μm. **g**, Immunoblots showing the extent of wt-L1Hs and L1-ORFeus expression using ORF1p as a reporter in cells depleted of SAFB proteins and/or the HUSH complex. SAFB KD, simultaneous SAFB1 and SLTM depletion (HCT116 cells do not express SAFB2; Extended Data Fig. 4b); HUSH KD, TASOR and MPP8 depletion. **h**, RNA blot using a DIG-labelled probe against ORF2 in untransfected or wt-L1Hs-transfected (same construct as in Fig. 2g) HCT116 cells depleted of SAFB, HUSH or both, showing highest L1 expression in cells transfected with an L1Hs-transcribing plasmid that are depleted of both HUSH and SAFB proteins.

Fig. 4c)[19–21]. We reasoned that, if suppression of L1 elements by SAFB proteins is dependent on the presence of A-biased open reading frames (ORFs), then ORFeus should not be subject to SAFB regulation. To test this hypothesis we transfected cells with either a plasmid encoding wt-L1Hs or another plasmid encoding ORFeus, which is identical to the wt-L1Hs plasmid except for the encoding of its ORFs. To gain additional insight into transcriptional versus post-transcriptional control we also targeted the HUSH complex, which is a well-characterized, RNA-based transcriptional regulator of L1 elements[22]. In wt-L1Hs-transfected cells, individual depletion of either SAFB (SAFB1 + SLTM, because no SAFB2 expression was detected in this cell line; Extended Data Fig. 4b) or the HUSH complex (TASOR + MPP8) increased the expression of ORF1p to a similar extent (Fig. 2g, lanes 2 and 3, compared with lane 1; also

see Extended Data Fig. 4d). Codepletion of SAFB and HUSH resulted in higher ORF1p levels than either depletion alone (Fig. 2g, compare lanes 2 and 3 with 4).

These results suggest that transcriptional derepression through HUSH depletion, coupled with post-transcriptional release following SAFB depletion, culminates in a strong total derepression of wt-L1Hs, which we further verified at the RNA level by RNA blotting (Fig. 2h). As expected, ORFeus was expressed at a much higher level compared with wt-L1Hs in control siRNA-treated cells (Fig. 2g, lane 5 versus 1). Depletion of SAFB proteins had no effect on ORFeus expression (Fig. 2g, compare lanes 6 and 5) whereas depletion of the HUSH complex upregulated ORFeus (Fig. 2g, compare lanes 7 and 5). Consistent with these results, dual knockdown of the SAFB + HUSH complex showed identical

expression of ORF1p compared with HUSH depletion alone (Fig. 2g, compare lanes 8 and 7), underscoring the observation that the removal of A-bias in L1 eliminated a key sequence feature recognized by SAFB proteins but not by the HUSH complex. These results suggest that, by maintaining A-biased coding sequences, mammalian L1 elements remain subject to SAFB repression.

## SAFB intronizes L1 and Tigger TEs

Given that SAFB proteins can bind to and affect the life cycle of L1 RNA, we reasoned that their depletion might also affect the expression of genes harbouring these sequences. We therefore depleted SAFB1, SAFB2 and SLTM, either individually or all three simultaneously (SAFB KD), in HEK293 cells and carried out poly(A+) RNA sequencing (RNA-seq). Consistent with the retrotransposition assay, depletion of SAFB1 led to the most pronounced gene expression changes among the three SAFB proteins (Extended Data Fig. 5a) followed by SLTM and SAFB2 (Extended Data Fig. 5b,c). Cells simultaneously depleted of all three proteins (SAFB KD), however, showed the most marked alterations in gene expression (Extended Data Fig. 5d). Similar changes were also observed in HeLa and HCT116 cells following SAFB KD (Extended Data Fig. 5e–h and Supplementary Fig. 5a,b), showing that the observed effects are overall independent of the cell model used. Combined with the results of the retrotransposition assay, RNA-seq results support the idea that SAFB proteins are at least partially redundant with each other. To test this idea further we transfected HEK293 cells with all siRNA combinations (SAFB1, SAFB2, SLTM, SAFB1 + SAFB2, SAFB1 + SLTM, SAFB2 + SLTM, SAFB1 + SAFB2 + SLTM) and used immunoblotting to monitor both siRNA efficiency and the response of SAFB proteins to each other's depletion (Extended Data Fig. 4c). These blots showed that downregulation of SAFB1 upregulates SAFB2, and vice versa, but SLTM expression appeared independent of SAFB1 or SAFB2 (Extended Data Fig. 4c). Interestingly, in HCT116 cells, in which SAFB2 is not expressed (Extended Data Fig. 4b), depletion of SAFB1 upregulated SLTM (Extended Data Fig. 4d), suggesting that a complicated ternary feedback loop regulates the expression of SAFB proteins and that depletion of all SAFB proteins is necessary to show the full extent of SAFB regulation in that system. Consistently, quantitative PCR analysis of targets showed that splicing inclusion events (discussed below) in SAFB KD cells are most severe when all three SAFB proteins are depleted (Extended Data Fig. 4e,f).

We further analysed changes in repeat-element expression using TEtranscripts[23], which showed a modest upregulation of L1 elements in SAFB1-depleted cells and SVA elements in SLTM-depleted cells (Extended Data Fig. 6b,c) and a clear upregulation of several L1 elements, as well as Tigger1 and Tigger2 DNA transposons in SAFB KD (Fig. 3a and Extended Data Fig. 6a,d). Although upregulation of L1 elements could be expected, because SAFB proteins bind to and suppress their retrotransposition (Fig. 2d,e), upregulation of extinct Tigger elements was unexpected but consistent with FLASH data showing specific enrichment of these elements (Fig. 2b,c and Supplementary Fig. 4b). To understand the nature of these upregulated transposons, we first compared SAFB-bound regions with genes misregulated in SAFB KD, which showed that genes bound by SAFB proteins at the RNA level are more likely to be misregulated (HEK293: $\chi^2 = 12.6$, $P < 0.001$, odds ratio 1.79; other cell lines and further details in Supplementary Table 15). This strongly suggests that the observed gene expression changes are the result of an altered post-transcriptional process in which SAFB proteins directly participate.

Autonomous TEs, including the main SAFB targets L1 and Tigger elements, encode at least one polypeptide for their life cycle, which requires that the transposon maintain a polyadenylation site (PAS) to produce a translation-competent mRNA. Because these PASs are efficiently skipped by the host RNA-processing machinery under normal conditions, we wondered whether depletion of SAFB proteins could activate these cryptic PASs, generate chimeric transcripts and cause the

observed changes in gene expression (Fig. 3b). Because their abundance increases in both the nucleus and cytoplasm, these chimeric transcripts probably result from de novo splicing events following SAFB depletion and are not due to enhanced nuclear export of steady-state products (Extended Data Fig. 5m). The main hallmarks of such a de novo event would be the apparent upregulation of the SAFB-bound transposon, because at least part of the transposon upstream of the PAS would now be part of the new terminal exon and the apparent downregulation of exons downstream of the SAFB peak, because transcription will be terminated before RNA polymerase II can reach these exons (Fig. 3d). Analysis of genes with or without upregulated SAFB peaks with respect to the expression of their exons upstream (pre-peak) or downstream (post-peak) of the SAFB peak showed that the presence of a SAFB peak within a host gene is indeed associated with downregulation of its downstream exons (Fig. 3c,d and Supplementary Fig. 5a,b). Our analysis of alternative polyadenylation in Oxford Nanopore Technology (ONT) direct RNA-seq data confirmed the activation of cryptic polyA sites as a cause of decoupled differential expression of pre- and post-peak gene segments. We identified 247 genes with alternative polyadenylation site usage following SAFB loss—that is, 1.7% of all genes, or around 5% of genes with long transcripts were disrupted by SAFB loss. Taken together, we provide evidence that SAFB proteins bind to transposable elements that have the potential to act as gene traps and keep them intronic by preventing the use of their PAS, and explain why we saw a significant upregulation of both Tigger elements and L1s in the RNA-seq data.

## SAFB prevents nested genes from becoming gene traps

Even strong, canonical PASs do not lead to premature termination of transcription when placed within an intron[24], consistent with the inclusion of a strong splice-acceptor site upstream of the PAS in gene traps used in genetic screens to ensure gene disruption[25]. During visual inspection of genes showing signs of early termination in SAFB KD, we noticed that the usage of the cryptic PAS found in TEs commonly coincided with the activation of an upstream cryptic splice site (Fig. 3d and Extended Data Fig. 5i). Unbiased transcriptome-wide analysis of new splice sites in our SAFB KD indicated significant enrichment in L1 and Tigger elements (Extended Data Figs. 5i and 6g,h), showing that loss of SAFB binding over these transposons activates cryptic splice sites, which brings the TE-encoded, intronic PAS into an exonic context and terminating host gene expression.

We then used SpliceAI[26] to evaluate the strength of new splice sites detected in SAFB KD. This analysis showed that new splice sites are indeed predicted to be weaker than annotated splice sites but stronger than random AG or GT dinucleotides in a 500 nt window around the novel site (Extended Data Fig. 5j; random AG|GT). Importantly, although weaker than annotated splice sites, new splice sites are the strongest within the 500 nt window (Extended Data Fig. 5j; best AG|GT). If SAFB depletion converts TE sequences into splice enhancers, we would expect to find new splice sites not just within SAFB-bound TEs (Extended Data Fig. 5i) but also in the local vicinity of these elements. Indeed, we found that new, upregulated splice acceptors that do not overlap with a TE are significantly closer to a downstream L1 element (Extended Data Fig. 5k). These are unlikely to be misannotated transposon boundaries—for example, a 6.3 kb L1PA7 fragment in the middle of gene *RB1* is exonized through the activation of a splice-acceptor sequence more than 400 base pairs (bp) away from the 5′ end of the element (Supplementary Fig. 5c) or, in the case of gene *RAD54L2*, a 6.1 kb L1PA6 fragment is exonized through a splice–splice acceptor site created by an antisense AluJb insertion more than 600 bp away (Supplementary Fig. 5d).

L1 and Tigger elements are intronless, single-exon genes that are neither spliced nor require splicing for their reproduction. It is therefore surprising that both elements are enriched with sequences that can act as splicing enhancers in SAFB-depleted cells. Intriguingly, analysis of

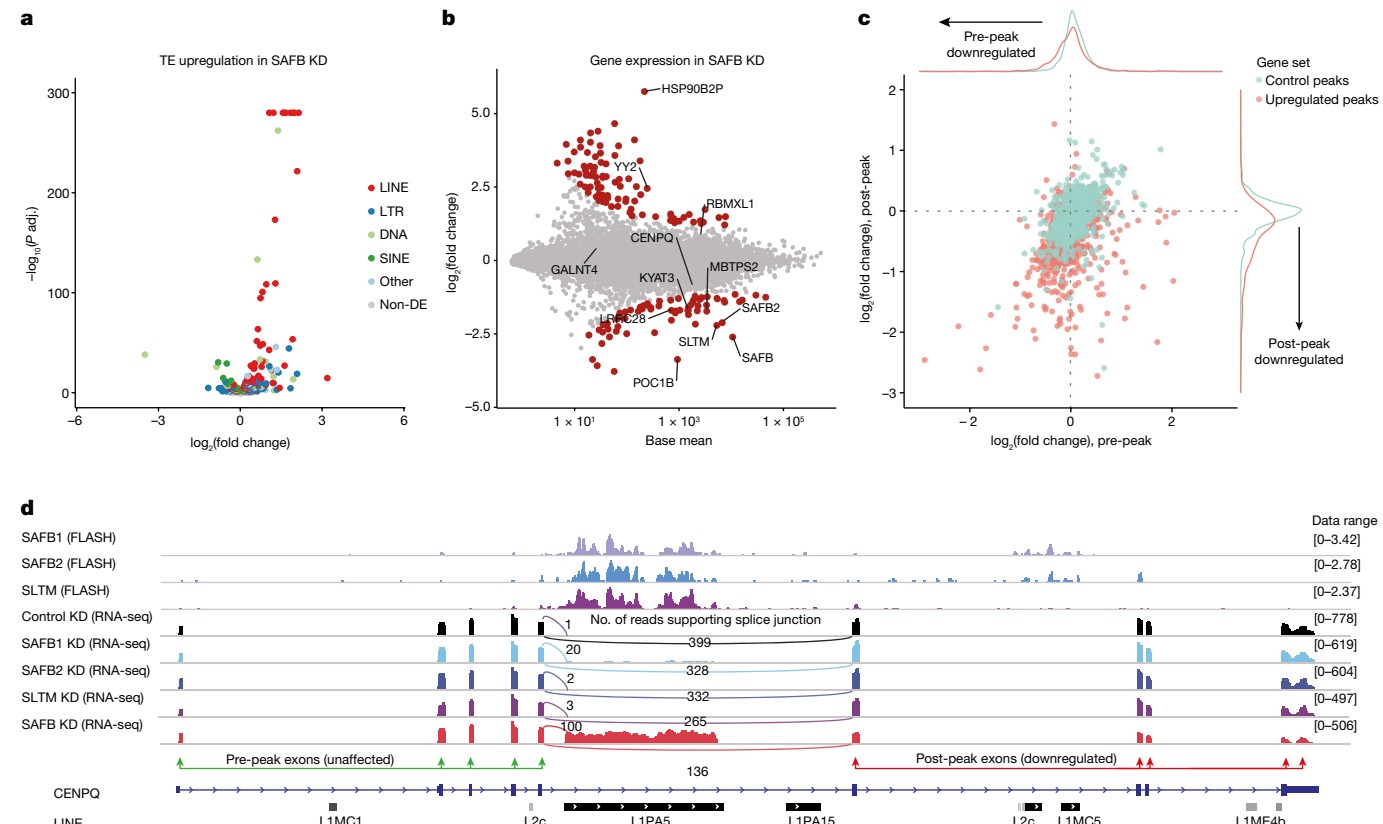

**Fig. 3 | SAFBs intronize L1 and Tigger transposons to prevent them from acting as gene traps. a,b**, Differential expression of transposable elements (**a**) and genes (**b**) following concurrent loss of the three SAFB proteins in HEK293 cells. Non-DE, non-differentially expressed. **c**, Comparison of expression change in pre- and post-peak gene fragments in genes with exonized (red points, *n* = 878) and control SAFB peaks (green points, *n* = 1457). **d**, IGV snapshot of the gene *CENPQ* showing extensive binding of SAFB1, SAFB2 and SLTM to a 4,165-bp-long L1PA5 retrotransposon inserted on the same strand as the host gene (top three tracks). Bottom five tracks show RNA-seq coverage of HEK293 cells transfected with control siRNAs or siRNAs against SAFB1, SAFB2, SLTM or all three together (SAFB KD).

FLASH data showed that SAFB proteins recognize an adenosine-biased, purine-rich sequence GAAGAA (Extended Data Fig. 5l), a prototypical exonic splicing enhancer (ESE) motif that strongly promotes splicing in natural contexts[27] and high-throughput screens[28,29] and that was recently identified in an in silico *k*-mer search for motifs that boost both acceptor and donor probabilities[26]. Furthermore, the sequences encoding ORFs of both L1 and Tigger elements are adenosine biased (over 33% adenosine content for Tigger1, over 40% for L1Hs) and enriched with purine-rich *k*-mers, which is evolutionarily conserved[20] (Supplementary Table 1). Interestingly, purine-rich ESE motifs were shown to promote nuclear retention of intronless RNA but not spliced RNA[30,31], consistent with L1 RNA leaving the nucleus in SAFB-depleted cells (Fig. 2f).

Intriguingly, intronless complementary DNA constructs of multi-exonic genes tend to be retained in the nucleus[32], suggesting that naturally intronless genes have a specific sequence composition that allows their efficient export. Indeed, intronless genes in humans have a high GC content throughout their bodies and increasing the GC content of reporter constructs leads to higher expression levels[33]. We consistently found that the GC-rich ORFeus is not suppressed by SAFB proteins (Fig. 2g), suggesting that high concentrations of purine-rich ESEs within an intronless construct—such as the cDNA of a multi-exonic gene or L1/Tigger RNA—may impede nuclear export. Our results show that these sequences do not lose their ESE potential, but rather are masked by SAFB proteins. Consistently, when we look at severely misregulated genes in our SAFB KD that cannot be traced to a TE exonization event, we find that these are sense pseudogenes that are cDNA copies of multi-exonic genes reverse transcribed into the genome by the L1 machinery (Supplementary Fig. 6).

SR proteins, some of which bind to GAA-rich ESE motifs, are also involved in nuclear export of mRNAs via their interactions with the mRNA exporter NXF1 (ref. 34). We thus wondered whether depletion of SAFB proteins would result in higher SR protein occupancy at SAFB targets, which could explain both the activation of splice sites in intronic TEs and the cytoplasmic accumulation of intronless, full-length L1 RNA in SAFB-depleted cells. To this end we depleted SAFB proteins in HEK293 cells and carried out FLASH experiments using a monoclonal antibody against phosphorylated SR proteins (1H4 (ref. 35); Extended Data Fig. 7c). In addition we used DHX9 as a control that interacts primarily with inverted *Alu* element pairs[15]. When genes were analysed together with SAFB-bound regions, around 64% of differentially expressed regions showed higher phospho-SR (p-SR) occupancy (716 of 1,123), roughly 90% of which were SAFB-bound regions (645 of 716). Segmentation of genes into exons, introns and repeat elements agnostic of SAFB binding showed that the category most affected was sense-L1 elements or their fragments, with approximately 90% (328 of 369) showing increased SR binding following SAFB depletion (Fig. 4a). These results suggest that SAFB proteins are an important component of the nascent L1 RNP, and a reduction in SAFB levels tends to increase SR binding at SAFB targets.

We then carried out affinity-purification mass spectrometry experiments (Extended Data Fig. 7a,b) with the cell lines used for FLASH that express tagged SAFB1, SAFB2 or SLTM (with green fluorescent protein (GFP) as control) to determine whether the redundancy of SAFB proteins and their potential competition against SR proteins has a biochemical basis (Supplementary Table 3). We found a common set of hnRNP and hnRNP-like proteins that interact with all three

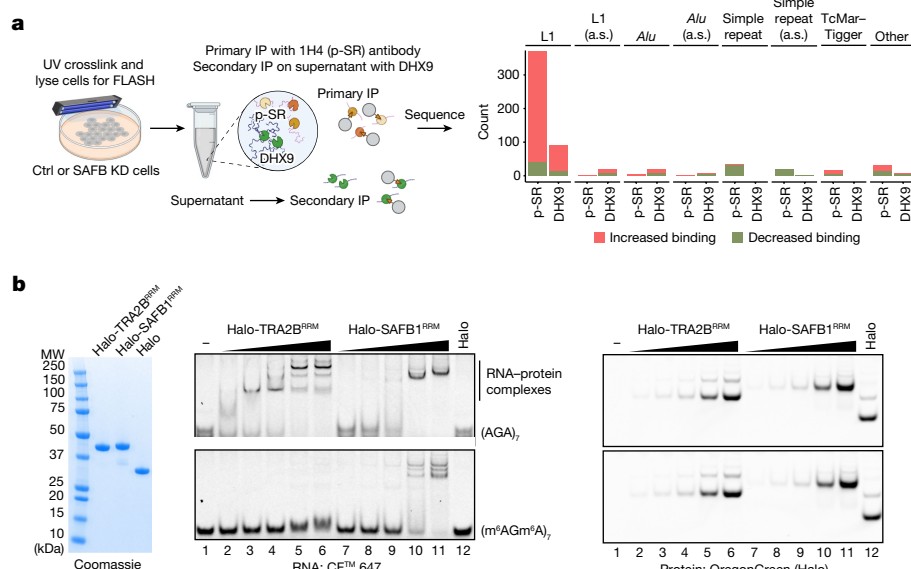

**Fig. 4 | Competition between SAFB and SR proteins is steered by m6A modification. a**, Left, experimental setup of the FLASH experiment. HEK293 cells were transfected with either control (ctrl) siRNA or siRNAs against SAFB1, SAFB2 and SLTM and ultraviolet (UV) crosslinked. Lysates from these samples were then used to carry out FLASH experiments with an antibody against either p-SR proteins (mAb 1H4) or DHX9, which primarily interacts with inverted *Alu* repeats and therefore serves as a control. Right, analysis of p-SR FLASH data using intronic repetitive elements, showing robust enrichment of sense-L1 elements (L1) compared to antisense-L1 (a.s.) in SAFB-depleted cells, whereas

DHX9 shows minimal changes on its preferred substrates (sense or antisense (a.s.) *Alu* elements). **b**, Left, Coomassie-stained acrylamide gel showing the purity of Halo-TRA2B^RRM^, Halo-SAFB1^RRM^ and Halo used for EMSA. Middle, EMSA using Halo-TRA2B^RRM^ (lanes 2–6), Halo-SAFB^RRM^ (lanes 7–11) and Halo (lane 12) and short RNA probes (24 nt long) that are either completely methylated (bottom) or unmethylated (top). Right, visualization of proteins via OregonGreen that was covalently attached to the Halo moiety. Gel images shown here are representative of two replicates.

SAFB proteins, such as RBM12B and RBMX/XL1, but could not identify copurifying proteins unique to a single SAFB protein in our assays (Extended Data Fig. 7c). Verification of interactions by immunoblotting showed a conspicuous lack of interactions with SR proteins and other splicing-associated factors (Extended Data Fig. 7c). We further verified these observations by immunoprecipitation (IP) of SAFB1 using a specific antibody with or without RNase treatment and immunoblotting, which showed RNA-independent interactions with NCOA5, RBM12B and ZNF638 (Extended Data Fig. 7d). Finally, using domain deletions, we show that these interactions depend on the presence of the ER-rich C terminus of SAFB1 (Extended Data Fig. 7e,f), which is also the only construct that completely failed to rescue splicing defects (Extended Data Fig. 4e,f) in complementation experiments. Although we cannot totally exclude the possibility that SAFB proteins directly block spliceosomal assembly on their targets, biochemical evidence is more consistent with SAFB proteins and associated hnRNPs dynamically competing with SR proteins to suppress exonization of their target RNAs.

## SAFB regulates giant protein-coding cassette exon splicing

Our results thus far suggest that SAFB proteins compete with SR proteins binding to GAAGAA and similar purine-rich motifs to prevent splicing interference by same-strand insertions of L1s, Tiggers and pseudogenes without interfering with the splicing of average-sized exons containing similar ESE motifs. Intriguingly, the preference of SAFB for long stretches of RNA was observed in all aspects of its RNA biology, whereas the median human exon is 136 nt in length, the median length of new exons that appear in SAFB-depleted cells is more than one order of magnitude longer (1,902 nt; Extended Data Fig. 8b) and that of all significantly upregulated exons in SAFB-depleted cells is 1,273 nt (Extended Data Fig. 8c). Among all the proteins profiled in our screen,

SAFB proteins are also unique in their enrichment for long coding exons (Extended Data Fig. 8a). During analysis of unusually long exons upregulated in SAFB KD we noticed a new category of exons that, to our knowledge, have not previously been described: giant protein-coding cassette exons. The most striking example in this category is ANK3, the master organizer of the axon-initial segment, a specialized subcellular region in neurons in which action potentials are generated. ANK3 anchors voltage-gated ion channels and cell-adhesion molecules to the cytoskeleton through interactions with α,β-spectrins[36]. For this role, a neurospecific, 480 kDa ANK3 isoform called giant ankyrin-G must be expressed and depends on the splicing of the giant 7.8 kb coding exon at the 3' end of gene *ANK3*. SAFB depletion in HEK293 cells increases the expression of this exon, from undetectable levels to those observed in Genotype-Tissue Expression Project (GTEx) brain RNA-seq data (Supplementary Fig. 7a). SAFB proteins also bind to and suppress splicing of the giant protein-coding exon of ANK2, which is another ankyrin that couples plasma membrane to the cytoskeleton within neuronal axons[37]. Among other giant coding exons that are bound by SAFB and spliced more efficiently in SAFB-depleted cells are the following: MAP2, which is mostly expressed in the somatodendritic compartment of neurons[38]; MAP4, which is expressed more broadly in somatic tissues, with splicing of its giant exon restricted to muscles and heart tissue (Supplementary Fig. 7c); MAPT, also known as tau, which is also expressed mainly in neurons although the larger version (Big tau) is restricted to the peripheral nervous system[39]; NIN, which is involved in anchoring the centrosome to microtubules during cortical neuron development[40]; MPRIP, an F-actin-binding protein involved in actin stress fibre regulation[41]; NASP a histone-interacting protein, the giant isoform of which is expressed preferentially in the testis[42]; and CLIP1, a microtubule plus-end-tracking protein, the giant version of which is expressed during spermiogenesis (Supplementary Fig. 7b) and involved in manchette formation[43]. All giant coding exons bound and suppressed by SAFB, with the exception of MAPT, are adenosine

biased (over 33% A) and enriched in purine-rich ESEs (Supplementary Table 1), mirroring the compositional bias of L1 and Tigger elements.

SAFB protein competition with SR proteins (Fig. 4a) could explain how these large exons are generally suppressed. However, it is not immediately clear why long exons (over 1 kb) would be more prone to SAFB-mediated repression than average-sized (around 150 nt) exons. Previous work with SAFB1 suggests that N6A-methylated adenosine (m6A) containing RNA might be involved in binding of SAFB1 to its targets[44] and that it is endogenously enriched over long exons, as well as L1 RNA and other autonomous TEs[45]. We therefore tested whether m6A could affect the interaction of SAFB1 with a short RNA (AGAx7) in vitro using electrophoretic mobility shift assay (EMSA). To focus on interactions mediated by globular domains we expressed the RNA-binding (RRM) domain of SAFB1 fused to a Halo-tag. As a control we used the RRM domain of SR-like protein TRA2B, which binds to AAGA RNA with micromolar affinity and can activate splicing of nearby splice sites[46] (Fig. 4b). Both TRA2B$^{RRM}$ and SAFB1$^{RRM}$ interact with the unmethylated (AGA)x7 RNA, TRA2B, with clearly higher affinity than SAFB1 (Fig. 4b, left, RNA; right, proteins visualized via Halo-tag on the same gel). Strikingly, TRA2B$^{RRM}$ lost all interactions with m6A-modified (m6AGm6A)x7 RNA whereas the interaction of SAFB1$^{RRM}$ was barely affected (Fig. 4b). Similar results were obtained using a 949 nt fragment of L1 ORF2 and native nuclear lysates prepared from HCT116 cells (Extended Data Fig. 7g). Taken together, these results indicate that TRA2B would have a clear advantage over SAFB1 in interacting with unmethylated nascent RNA released from RNA polymerase II; however, this advantage would shift towards SAFB1 as RNA becomes progressively methylated in the nucleus. Both intronless A-rich L1 transcripts and giant exons are therefore more likely to be subject to SAFB regulation via progressive m6A modification compared with average-sized exons.

## Ancestral function of SAFB proteins

SAFB targets in humans share several key features, including a long, contiguous stretch of purine-rich motifs and a propensity to switch to an exonic context from an intronic one following SAFB depletion. Nonetheless, the diversity of the targets, including L1 TEs, Tigger TEs, pseudogenes and giant cassette exons, is highly unusual, which prompted us to investigate their conservation in other species.

To this end we carried out FLASH experiments with endogenously tagged Safb1 in mouse 3T3 fibroblast cells to determine SAFB targets in mice, and carried out poly(A$^+$) RNA-seq experiments in Safb1-, Safb2- and Sltm-depleted cells (SAFB KD) to determine the fate of these targets. Similar to human cells, we detected a significant increase in L1 expression in 3T3 SAFB KD cells (Extended Data Figs. 9d and 6e). Unexpectedly, we also detected a significant increase in the expression of the LTR/ERV family of TEs (Extended Data Fig. 6e), which are active in mice but no longer active in the human genome. In accordance, analysis of Safb1 FLASH peaks showed enrichment of both L1 and LTR TEs and the adenosine-biased, purine-rich motif GAAGAAGA (Extended Data Fig. 5l). Analysis of splice sites activated in SAFB KD showed significant enrichment for L1 and LTR/ERV elements (Extended Data Fig. 6i), resulting in chimeric transcripts similar to human cells (Supplementary Fig. 8a). Interestingly, the ORFs of the two most upregulated ERV elements, MMERVK10C and MERVL (Supplementary Table 13), are A biased—roughly 32 and 29% A content, respectively—and enriched with purine-rich $k$-mers similar to human L1 and Tigger elements (Supplementary Table 1). Notably, we also detected splicing of the neuro-specific giant exons of Ank3 and Ank2, as well as the spermatid-specific giant exon of Clip1 (Extended Data Fig. 9b) following SAFB KD in mouse 3T3 fibroblast cells, indicating that the giant coding exon suppression function of SAFB proteins is conserved in mice. Furthermore, expression of Mbtps2 and Poc1b, which are downregulated in human cells due to splicing interference and early termination induced by same-strand retrogenes *Yy2* and *Galnt4*, was similarly downregulated

in 3T3 cells following SAFB KD by the same retrogenes. We also found mouse-specific host–pseudogene pairs including *Acsl3/Utp14b*, *H13/Mcts2*, *Cdk5rap2/retro-Ywhaq*, *Mipol1/Prps1l3*, *Bfar/retro-Pphln1* and *Smo/retro-Rpl35*, in which the host is downregulated at the expense of pseudogene upregulation. Supporting the notion that SAFB proteins compete with SR proteins on GAA-rich target sequences to prevent them from acting as splicing enhancers, we found four genes in mice (*Cp*, *Tmx3*, *Vps13b* and *Uggt2*) in which Safb1 binds intronic, GA-rich simple repeats and, following depletion of SAFB proteins, proximal cryptic splice sites are activated, leading to downregulation of host genes (Supplementary Fig. 8b). These results demonstrate that SAFB proteins in humans and mice are functionally analogous.

Unlike mammals, which have three SAFB proteins, invertebrates have a single SAFB orthologue in their genome, prompting us to investigate whether the function of SAFB extends also to invertebrates. *Drosophila melanogaster* is an interesting model for this purpose: although its genome contains up to about 20% transposons, most of these elements are found within heterochromatin (Extended Data Fig. 9e) such that fly genes have introns that are virtually transposon free (median transposon content is 0%, mean 0.1%)[47]. Next we performed FLASH in *Drosophila* S2 cells to determine Saf-B targets and depleted the single Saf-B protein by RNA interference to measure ensuing gene expression changes by RNA-seq. Similar to human and mouse SAFB proteins, *Drosophila* Saf-B also enriches a purine-rich motif, AGGAGAAG (Extended Data Fig. 5l). Similar to mice, the *Drosophila* genome contains active L1 and LTR elements enriched in Saf-B FLASH data, and these were upregulated following Saf-B depletion (Extended Data Figs. 9d and 6f). Strikingly, the most significantly upregulated gene, *dlt*, and the most significantly downregulated gene, *alpha-Spectrin*, are expressed from the same promoter and share the first non-coding exon, mirroring the pseudogene–host gene architecture seen in mammalian cells (Extended Data Fig. 9c). We saw the same pattern in other severely misrelated gene pairs, including *vimar/CG30156*, *Act/us*p, *CG8176/JHDM2*, *grp/squ* and *eIF2gamma/Su(var)3–9*. Even though we noted an increase in TE expression following Saf-B depletion similar to that in mammalian cells, we did not detect splicing or early termination events that could be directly traced back to an intronic TE, consistent with the scarcity of intronic TE insertions in flies. However, when we sorted differentially expressed genes following Saf-B depletion by their TE content we found three exceptionally long genes with extremely high TE content that were severely downregulated following Saf-B depletion: *Gprk1*, *Dbp80* and *Parp1* (Extended Data Fig. 9f). All three genes are located in the pericentromeric regions on three arms of chromosomes 2 and 3 (Extended Data Fig. 9e). Interestingly, the early exons of all three genes showed normal expression following Saf-B depletion (Extended Data Fig. 9g, green boxes), reminiscent of mammalian genes spliced into intronic TEs (Extended Data Fig. 9a), the exons downstream being markedly downregulated (Extended Data Fig. 9g, red boxes), indicating that these genes were prematurely terminated following Saf-B depletion.

## SAFB regulation in natural contexts

Our results show that the role of SAFB proteins in suppression of TE activity has deep evolutionary roots and is probably exapted from the need to regulate nested gene expression, prevention of pseudogene exonization and enabling the regulation of giant cassette exon splicing (Extended Data Fig. 9a–c). The adenosine bias in ORFs of L1 is well conserved in vertebrates[20] and our results show that this sequence bias is shared by unrelated autonomous transposon families, suggesting the existence of a hitherto unknown evolutionary pressure to maintain it. Transposons have been linked to adaptation to stress since their discovery[48], and SAFB is a prominent member of nuclear stress bodies (Extended Data Fig. 10a,b), which are membraneless organelles that are nucleated by the long, non-coding RNA HSATIII expressed from pericentromeric HSATIII repeats following exposure to stress[49]. Reanalysis

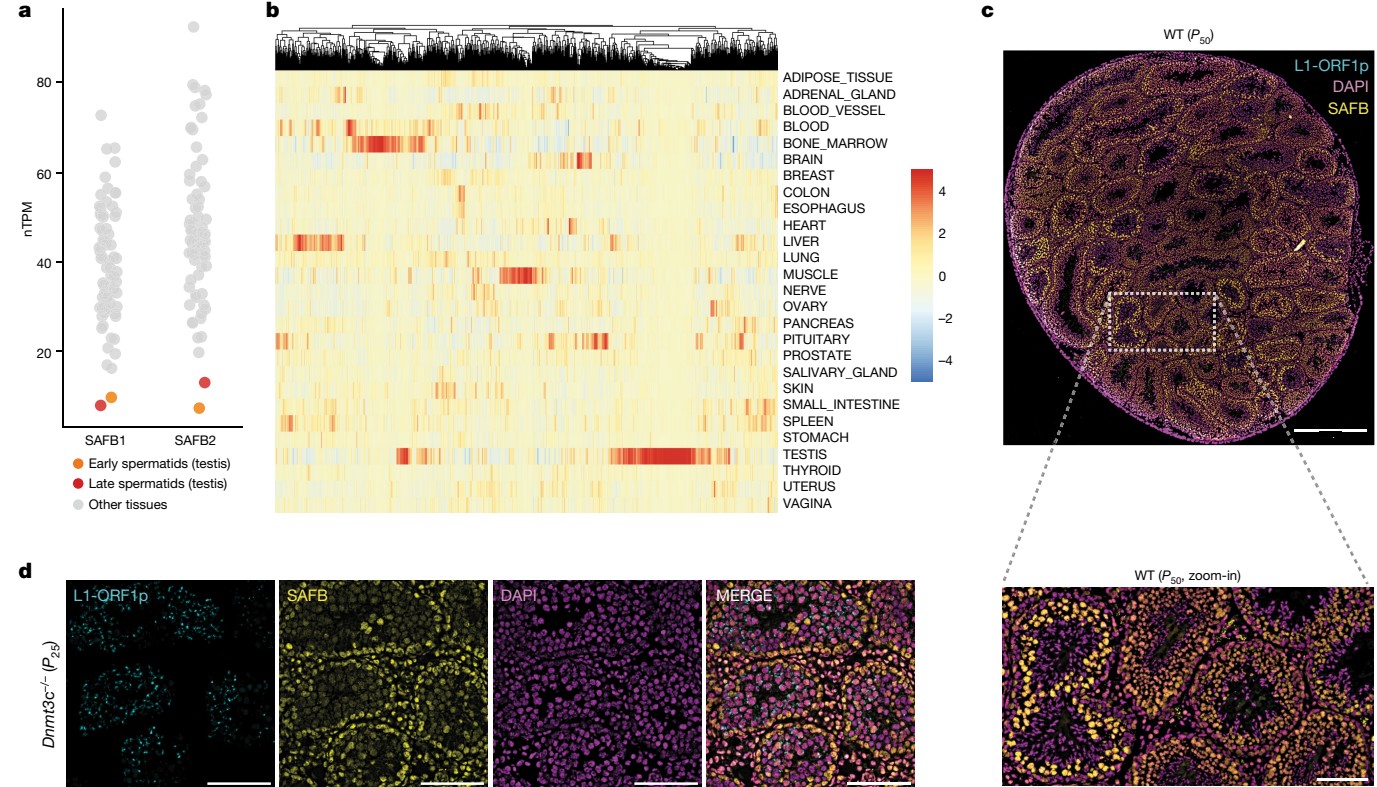

**Fig. 5 | Evolutionarily conservation of SAFB function. a**, Expression of SAFB1 and SAFB2 in various human tissues (single-cell RNA-seq data from the Human Protein Atlas) shown as normalized transcripts per million (nTPM). **b**, Enrichment of splice junctions between annotated splice donors and intronic SAFB peaks in human tissues catalogued by the GTEx consortium ($n = 1,104$). **c**, Cryosection of a WT mouse testis ($P_{50}$), costained with antibodies against Safb1 (yellow) and ORF1p (cyan) to show the differential expression of Safb1 at different stages of spermatogenesis. No specific signal for ORF1p was detected; see also Extended Data Fig. 10f, in which the channels are represented separately. Main image scale bar, 500 µm; magnified image scale bar, 100 µm. **d**, Cryosection of *Dnmt3c$^{-/-}$* mouse testis costained with antibodies against Safb1 (yellow) and ORF1p (cyan), showing intense staining of ORF1p towards the lumen where Safb1 expression is low. Scale bars, 100 µm.

of nascent RNA-seq data following 90 min of heat shock[50] showed that new splice sites detected in triple KD cells tend to be upregulated during heat shock (Extended Data Fig. 10d,e) together with a broad upregulation of TEs, including L1 elements (Extended Data Fig. 10c). These results suggest that sequestration of SAFB proteins (Extended Data Fig. 10a–e) could limit their availability and relax TE suppression under stress conditions, which can be evolutionarily advantageous. SAFB proteins are broadly expressed in the human body, supporting this role (Fig. 5a). We noticed, however, that, although broad, SAFB1 and SAFB2 expression drops precipitously in postmeiotic spermatids (Fig. 5a). Immunofluorescence staining of SAFB1 on a cryosection of a testis from a 7-week-old, wild-type (WT) mouse confirmed these observations, with peak expression detected in spermatocytes (Fig. 5c). To validate the inverse relationship between SAFB activity and L1 expression (Fig. 2g,h) we examined a new *Dnmt3c* knockout model, because this gene was recently shown to regulate L1 transcription and activity purely within the male germline[51]. Notably, *Dnmt3c* knockout testis resulted in a robust ORF1p signal, but almost entirely within differentiating cells showing low SAFB1 intensity (Fig. 5d and Extended Data Fig. 10f). This germline-specific relationship may present an interesting window of opportunity for new TE insertions, because they could be inherited by the next generation without creating a mutational burden on the host, in contrast to somatic TE insertions which can be lethal but cannot be inherited. Furthermore, unlike early embryogenesis, which is a one-time event per organism, spermatogenesis in adult males is a continuous process, creating billions of opportunities for new TE insertions without compromising host fitness. Because somatic TE insertions are not heritable and the Piwi-interacting RNA system is

active in the germline, confinement of the TE–host conflict to spermatogenesis would be a mutually beneficial strategy for both host and TEs. In support of this model, splice sites derepressed following SAFB depletion are most significantly upregulated in the testis, which represents a natural low-SAFB-expression regime (Fig. 5b). Moreover, ectopic overexpression of human SAFB1 in mouse N2A cells suppresses the splicing of Clip1's giant exon (Extended Data Fig. 10g,h), suggesting that SAFB levels must remain low during spermiogenesis.

## Discussion

Transposable elements and their host genomes must maintain a precarious balance for their continued co-existence but strikingly, on average, roughly 45% of mammalian genomes consist of TEs[52] and, in humans, about 65% of TEs are intronic[53]. The evolutionary histories of introns and TEs are deeply intertwined: spliceosomal introns and L1 elements share a common evolutionary origin with bacterial transposons, called self-splicing group II introns[54], and DNA transposons are shown to be responsible for rapid intron gains in eukaryotes[1]. Extant intronic TEs are therefore the product of a continuous evolutionary process shaped by the splicing machinery that can distinguish TEs as 'non-self' through recognition by RBPs of specific patterns[55]. Our results show that one such pattern is long, A-biased RNA, which remains intronic by attracting SAFB proteins, a process probably also influenced by m6A modification. Reduced SAFB expression, naturally or induced, leads to exonization of these RNAs, such as retrogenes, pseudogenes, nested genes, giant coding exons and autonomous transposons but, most notably, L1 elements (Extended Data Fig. 10i). SAFB proteins oppose

retrotransposition of L1 elements and maintain them in an intronic context if they do retrotranpose into genes; however, although SAFB proteins are broadly expressed in somatic tissues and the germline, their levels drop during spermatogenesis. Moreover, engineered removal of the adenosine bias from L1 RNA liberates it from SAFB suppression.

In sum, we show that autonomous TEs, which must express at least one protein for their reproduction, maintain an adenosine-biased coding sequence and thus subjecting themselves to SAFB-mediated suppression in somatic cells where TE activity is futile; this leaves open a highly controlled window of opportunity during spermatogenesis to avoid extinction, and thereby TEs continue to contribute to the evolution of their host genome

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

## Methods

### Cell culture and generation of stable cell lines

Flp-In T-REx HEK293 (Thermo Fisher Scientific, catalogue no. R78007) cells were maintained according to the manufacturer's recommendations. Cells were cultured in DMEM with glutamax supplemented by Na-Pyruvate and High Glucose (Thermo Fisher Scientific, catalogue no. 31966-021) in the presence of 10% fetal bovine serum (FBS; Thermo Fisher Scientific, catalogue no. 10270106) and penicillin/streptomycin (Thermo Fisher Scientific, catalogue no. 15140-122). Before their introduction, transgene cells were cultured at a final concentration of 100 μg ml$^{-1}$ zeocin (Thermo Fisher Scientific, catalogue no. R250-01) and 15 μg ml$^{-1}$ blasticidin (Thermo Fisher Scientific, catalogue no. A1113903). For generation of stable cell lines, pOG44 (Thermo Fisher Scientific, catalogue no. V600520) was cotransfected with pcDNA5/FRT/TO (Thermo Fisher Scientific, catalogue no. V652020) containing the gene of interest at a 9:1 ratio. Cells were transfected with Lipofectamine 2000 (Thermo Fisher Scientific, catalogue no. 11668019) on a six-well-plate format with 1 μg of DNA (that is, 900 ng of pOG44 and 100 ng of pcDNA5/FRT/TO + GOI) according to the transfection protocol provided by the manufacturer. All transgenes were cloned with an N-terminal His$_6$-biotinylation sequence-His$_6$ tandem (HBH) tag that allows rapid and ultraclean purification without the use of antibodies. We also added a 3× FLAG tag immediately before the HBH tag to increase the versatility of the construct, which we refer to as the 3FHBH tag. Twenty-four hours following transfection, cells were split among three wells of a six-well plate at dilution ratios of 1:6, 2:6 and 3:6 to allow efficient selection of hygromycin B (Thermo Fisher Scientific, catalogue no. 10687010). Hygromycin selection was started 48 h following the transfection time point, with a final concentration of 150 μg ml$^{-1}$, and refreshed every 3–4 days until control, non-transfected cells on a separate plate were totally dead. Induction of the transgene was performed overnight at a final concentration of 0.1 μg ml$^{-1}$ doxycycline (DOX). Cells were validated by immunoblotting of whole-cell lysates.

An endogenous biotin acceptor peptide affinity tag and a FLAG tag were inserted into the *Safb* gene locus for mouse and fly cell lines using CRISPaint. The mouse Flp-In 3T3 cell line was purchased from Thermo Fisher Scientific (catalogue no. R76107) and cultured according to the manufacturer's instructions. Vells were cultured in DMEM (Thermo Fisher Scientific, catalogue no. 31966-021) in the presence of 10% FBS (Thermo Fisher Scientific, catalogue no. 10270106) and penicillin/streptomycin (Thermo Fisher Scientific, catalogue no. 15140-122). The *Drosophila* S2R+ -MT::Cas9 cell line was purchased from DGRC (DGRC stock no. 268) and cultured in S2 medium (Thermo Fisher Scientific, catalogue no. 21720024) in the presence of 10% FBS (Thermo Fisher Scientific, catalogue no. 10270106). For CRISPaint[56] constructs (see Supplementary Table 2 for a list of single-guide RNAs), cells were cotransfected with three plasmids according to the CRISPaint protocol on the six-well-plate format using FuGene HD (Promega, catalogue no. E2311). Twenty-four hours following transfection, cells were expanded on 10 cm culture plates to facilitate efficient selection of puromycin (Thermo Fisher Scientific, catalogue no. A1113803). Puromycin selection is provided in the tag construct and is driven by expression from the gene locus (in this case, either the mouse or fly *Safb1* gene locus). Puromycin selection was started 48 h following transfection, at 1 μg ml$^{-1}$ final concentration, and was refreshed every 2 days and, in total, was maintained until all untransfected 3T3 or S2 cells were dead. Cells were validated by immunoblotting of whole-cell lysates.

The HeLa cell line (ACC57) was purchased from Deutsche Sammlung von Mikroorganismen und Zellkulturen and maintained in the same medium as the Flp-In 3T3 cell line, but with the addition of non-essential amino acids (Thermo Fisher Scientific, catalogue no. 11140050).

Mouse N2A cells were maintained in DMEM, and stably expressing 3× FLAG-Cas9 or 3× FLAG-SAFB1 (Extended Data Fig. 10g) was created by cotransfection of cells with plasmids expressing the protein of interest (Cas9, SAFB1 or control) under the EF1alpha promoter flanked by PiggyBack inverted repeats, together with a plasmid expressing PiggyBac transposase. In this design, because neomycin resistance was coupled to transgene expression via an IRES element, cells were selected with 1 mg ml$^{-1}$ geneticin until none remained in control transfected cells.

Cell lines (human Flp-In T-REx HEK293, human HeLa, human HCT116, mouse Flp-In 3T3, mouse N2A and fly S2R+) were all purchased from vendors or repositories or provided by colleagues (as described above), and no further authentication of cell lines was performed following purchase. Routine mycoplasma contamination tests were performed on all cell lines using the Jena Biosciences Mycoplasma (PCR-based) detection kit (Jena Biosciences, no. PP-401).

### FLASH

Cells on 15 cm dishes were washed with 6 ml of ice-cold PBS and UV-crosslinked with 0.199 mJ cm$^{-2}$ UV-C light, after which they were pelleted, snap-frozen in liquid nitrogen and stored at −80 °C until use. Pellets were resuspended in 600 μl of 1× native lysis buffer (NLB) with protease inhibitors and briefly sonicated in a Bioruptor water bath sonicator (30 s on, 30 s off, five cycles at 4 °C). Lysates were then centrifuged at 20,000 relative centrifugal force (rcf) for 10 min at 4 °C to remove insoluble material. Supernatant was transferred to a fresh tube with 25 μl of MyONE C1 streptavidin beads (Thermo) and incubated in a cold room with end-to-end rotation for 1 h. Beads were washed once with high-salt buffer (HSB), once with non-denaturing buffer (NDB), treated with 0.02 U μl$^{-1}$ RNase I (Thermo) in 100 μl of NDB for 3 min at 37 °C and immediately placed on ice to stop the reaction. Beads were then washed once each with HSB and NDB. RNA ends were repaired with T4 polynucleotide kinase, after which barcoded s-oligos were ligated with T4 RNA ligase 1 for 90 min at 25 °C. The 3′ phosphate at the 3′ end of each s-oligo was removed with recombinant shrimp alkaline phosphatase (NEB, M0371) and beads were washed once each with lithium dodecyl sulfate buffer, protein lysis buffer and HSB, and finally with NDB. RNA was released by treatment with proteinase K and purified using Oligo Clean and Concentrator columns (Zymo). Reverse transcription was carried out with SuperScript III and samples then treated with RNase H (NEB) to phosphorylate the 5′-end of the cDNA molecule. Following a final round of purification with Oligo Clean and Concentrator columns (Zymo), cDNA was circularized with CircLigaseII (Lucigen) and amplified with Q5 polymerase (NEB). PCR products were purified with solid-phase reversible immobilization beads, quality controlled with Bioanalyzer and subjected to high-throughput sequencing.

### FLASH data processing

Paired-end reads were merged with bbmerge.sh v. 38.72 using the following command: bbmerge.sh in1 = {R1.fastq.gz} in2 = {R2.fastq.gz} out = {merged.fastq.gz} outu1 = {unmerged.R1.fastq.gz} outu2 = {unmerged.R2.fastq.gz} ihist = {histogram.txt} adapter1=AGATCGGAAGAGCACACGTCTGAACTCCAGTCACCCAACAATCTC adapter2=AGATCGGAAGAGCGTCGTGTAGGGAAAGAGTGTAGATCTCGGTGG --mininsert=1. Short inserts (below 20 nt, following removal of the unique molecular identifier (UMI) and internal index) were removed with bbduk.sh v. 38.72 bbduk.sh in = {infile} out = {out} minlen=34. The UMI was removed from reads and written to the header with UMI_tools v.1.0.0: umi_tools extract --bc-pattern=NNNXXXXXXNNNNN -I {IN.fastq.gz} -–3prime --stdout = {OUT.fastq.gz}, followed by separation of replicates with flexbar v.3.5.0: flexbar -r INPUT.fastq.gz -b barcodes.fa --barcode-trim-end RTAIL --barcode-error-rate 0.2 --zip-output GZ. Reads were aligned first to abundant RNAs such as transfer RNA, small nuclear RNA, small nucleolar RNA and ribonuclear RNA, then to the genome with bowtie2 v.2.3.5: bowtie2 --no-unal --un-gz -L 16 --very-sensitive-local -x bt2_index -U fastq_in.fastq.gz -o bam_out.bam. Unaligned reads were remapped to the genome with bbmap.sh v.38.72 to capture spliced reads: bbmap.sh -Xmx50G in = {fastq_in} out = {bam_out} outu = {unmapped_out} ref = {reference.fa} sam=1.3 mappedonly=t

mdtag=t trimreaddescriptions=t nodisk. Finally, PCR duplicates were removed using UMI-tools: umi_tools dedup -I in_bam -S out_bam --spliced-is-unique --soft-clip-threshold 3 --output-stats = {stats}. Coverage files were generated with bamCoverage v.3.3.1: bamCoverage -b bam --filterRNAstrand [forward | reverse] --binSize 1 --normalizeUsing CPM --exactScaling -o out_file.

## UMAP of FLASH data

For construction of the UMAP, peak calling was carried out on all profiles using HOMER: findPeaks {tag_directory} -style factor -strand separate -o {peaks.txt} -i {background_tag_directory}. Peaks from all profiles were then merged with: mergePeaks -strand -d given -matrix {peaks1.txt peaks2.txt ...} > merged.peaks.txt. A count matrix, using all alignments from all profiles against merged peaks, was then created with featureCounts v.2.0.1: featureCounts -F SAF -Q 10 --primary -s 1 -T 12 -a {merged_peaks} -o {merged_peaks.counts.txt} {all_bam_files}. The count matrix was imported into a Jupyter notebook with pandas: peaks = pd.read_csv("merged_peaks.counts.txt", sep = "\t", index_col = "Geneid"), scaled with sklearn.preprocessing.StardardScaler: peaks_scaled = StandardScaler().fit_transform(peaks), which was then used to create the UMAP: peaks_scaled_mapper = umap.UMAP(n_neighbors=15, random_state=42).fit(peaks_scaled), and plotted using umap.plot.points function. Clusters were called with HDBSCAN: clusterable_embedding = umap.UMAP(n_neighbors=30, min_dist=0.0, n_components=14, random_state=42).fit_transform(peaks_scaled), then hdbscan_labels = hdbscan.HDBSCAN(min_samples=100, min_cluster_size=600, core_dist_n_jobs=1).fit_predict(clusterable_embedding).

## Sample and library preparation for RNA-seq

Flp-In T-REx HEK293 and HeLa ACC57 cells were transfected at a final concentration of 5 nM each (in the case of triple knockdown, total siRNA concentration became 15 nM and hence single-knockdown transfections were increased to 15 nM with the addition of 10 nM negative control siRNA) using Silencer Select siRNAs (Thermo Fisher Scientific, catalogue no. 4427037 for 1 nM scale) and RNAiMAX (Thermo Fisher Scientific, catalogue no. 13778030) on six-well plates (around 200,000 were used per replicate). Silencer Select siRNAs are 21 nt long, chemically modified (the exact modification is proprietary; Thermo Fisher) and reduce overall off-target effects by up to 90% without compromising potency. This modification also exaggerates strand bias, which correlates with better knockdown, and therefore they are 5- to 100-fold more potent than other siRNAs. The siRNA ID for human SAFB1 is s12452, for SAFB2 is s18599 and for SLTM is s36384. Cells were harvested on the second day of knockdown.

The Silencer Select siRNAs used were s29362 for MPP8 was s23449 for TASOR.

Flp-In 3T3 cells were first reverse transfected (roughly 100,000 per replicate) with 5 nM siRNA, boosted with the same amount 24 h following knockdown (forward transfected) and harvested on the third day following initial transfection. The siRNA ID for mouse Safb1 is s104978, for Safb2 is s104977 and, because the human SLTM siRNA also targets mouse mRNA, the same siRNA was used.

*Drosophila* S2R+ cells (DGRC no. 150) were transfected with control dsRNA against GFP or Saf-B using FuGENE HD (Promega) for 3 days, after which cells were harvested for RNA isolation.

Total RNA from human, mouse or *Drosophila* cells was extracted with the Quick-RNA MicroPrep kit (Zymo). Polyadenylated RNA was isolated from total RNA with the Dynabeads mRNA Purification Kit (Thermo). Purification was carried out twice to enrich poly(A)+ RNA. Sequencing libraries were generated using the KAPA Stranded RNA-Seq Library Preparation Kit (Roche).

## Isolation of nuclear and cytoplasmic RNA for RNA-seq

Forty-eight hours following siRNA transfection (control or SAFB1 + SAFB2 + SLTM, 5 nM each), approximately 1 million Flp-In T-REx HEK293 cells per replicate were trypsinized and either used directly for RNA isolation (total sample) or resuspended with a buffer containing 0.5% Igepal CA-630 to separate nuclear and cytoplasmic fractions, as described in ref. 57. Nuclear and cytoplasmic RNAs were isolated with the Quick-RNA MicroPrep kit (Zymo). Ribo-depleted RNA-seq samples were prepared using the KAPA RNA HyperPrep Kit with RiboErase (HMR) (no. KK8560, Roche).

## Transient transfections in rescue experiments and sample preparation for qPCR detection

SAFB triple knockdown was performed on Flp-In T-REx HEK293 cells as described above, and then FuGENE HD forward transfected with WT or truncation mutants as shown in Extended Data Fig. 7f while at the same time refreshing the medium 6 h following transfection of siRNAs. Transgenes were induced on day 1 of knockdown with 0.1 µg ml$^{-1}$ DOX for 24 h. On day 2 of knockdown, total RNA extracts were prepared with the Zymo Quick-RNA Kit and first-strand cDNA synthesis was carried out with PrimerScript RT Master Mix (TaKaRa, no. RR036A). Quantitative real-time PCR was performed using the oligos listed in Supplementary Table 1 with the Blue S'Green qPCR Kit (Biozym, no. 331416).

## ONT direct RNA-seq

Isolation of polyA-enriched mRNA from Flp-In T-REx HEK293 cells treated with either control siRNA or siRNAs against SAFB1, SAFB2 and SLTM (5 nM each) for 2 days was carried out using the Dynabeads mRNA DIRECT purification kit (Thermo Fisher Scientific) following the manufacturer's instructions, with minor modifications. In brief, approximately $4 \times 10^6$ cells were subjected to the standard protocol and hybridization of the beads/mRNA complex was carried out for 10 min on a Mini Rotator (Grant-bio). DNA containing supernatant was removed and the beads were resuspended with 2 × 2 ml of buffer A following a second wash step with 2 × 1 ml of buffer B. Purified RNA was eluted with 10 µl of preheated elution buffer (10 mM Tris-HCl pH 7.5) for 5 min at 80 °C. Quantification of isolated mRNA was performed using a Qubit Fluorometer together with the RNA HS Assay kit (Thermo Fisher Scientific). For direct RNA-seq, 700 ng of freshly isolated polyA-enriched mRNA was processed according to the manufacturer's protocol (no. SQK-RNA002). Final sequencing libraries were then loaded on R9.4 flow cells and sequenced on MinION and PromethION sequencers.

## Retrotransposition assay

The transfection and experimental timeline for the retrotransposition assay was followed as described in ref. 18. Initially around 200,000 HeLa cells were transfected, with the same siRNAs and under the conditions listed above, on a six-well plate with 5 nM final concentration each of negative control, SAFB1, SAFB2 and SLTM siRNAs. The following day, knockdown HeLa cells were transfected with 200 ng of plasmids pYX015 (based on JM111, which has a point mutation in ORF1p) for background control and pYX017 (pCAG-driven L1RP) for L1 activity in triplicates, using Lipofectamine 2000 on a 48-well plate in triplicate. Twenty-four hours following reporter construct transfection, 2.5 µg ml$^{-1}$ puromycin selection was started and maintained for 3 days (that is, day 5 of knockdown). Cells were washed with PBS before lysing with 40 µl of passive lysis buffer from the Dual-Luciferase Reporter Assay System (Promega, catalogue no. E1960). Half of the lysate was transferred to a 96-well, reading-compatible plate and measured using an Omega Lumistar machine.

## RNA–FISH

FISH was carried out in HCT116 cells transfected with control versus siRNAs against SAFB1 and SLTM (no SAFB2 expression was detected in HCT116 cells) for 48 h using the Stellaris RNA–FISH kit (https://www.biosearchtech.com/assets/bti_stellaris_protocol_adherent_cell.pdf). Probes against L1Hs were synthesized by LGC Biosearch Technologies (see Supplementary Table 2 for sequences). Probes against GAPDH were

sourced from LGC Biosearch Technologies (SMF-2026-1), provided by M. Bothe. Probes were used at a concentration of 125 nM and hybridized for 16 h at 37 °C. Samples were imaged using a Leica Stellaris 8 confocal microscope.

### EMSA with recombinant Halo, Halo-SAFB1[RRM] and Halo-TRA2B[RRM]

The RNA-binding domain of TRA2B (residues 111:201) and SAFB1 (residues 386:485) were cloned into a plasmid encoding 10× His-TEV-Halo. Three constructs (Halo only, Halo-TRA2B[RRM] and Halo-SAFB1[RRM]) were then expressed using BL21-CodonPlus(DE3)-RIL bacteria, which were induced when an optical density of roughly 0.6 was reached, with 0.2 mM isopropyl-ß-D-thiogalactopyranoside for 4 h at 37 °C, then collected by centrifugation. Bacteria were resuspended with lysis buffer (50 mM HEPES pH 8.0, 300 mM NaCl, 5 mM imidazole and 0.05% Igepal CA-630) and disrupted with a Branson sonifier, clarified by centrifugation and filtered through a 0.45 μm membrane. Cleared lysates were incubated with cOmplete His-Tag Purification Resin (Roche), washed extensively with lysis buffer and incubated with 0.5 μM OregonGreen (Promega) on beads in lysis buffer at room temperature for 30 min for fluorescent labelling of proteins. Beads were first washed extensively with lysis buffer, then with high-salt wash buffer (50 mM Tris.Cl pH 8.0, 1 M NaCl, 5 mM imidazole) and lastly with lysis buffer. Proteins were eluted with elution buffer (50 mM Tris.Cl pH 8.0, 100 mM KCl, 200 mM imidazole). Eluates were pooled, dithiothreitol (DTT, 1 mM final concentration) and TEV protease (home-made, 6× His-tagged, approximately 1:100) were added and samples dialysed against 25 mM Tris.Cl pH 7.4, 50 mM KCl, 5% glycerol and 1 mM DTT overnight in a cold room (about 8 °C). Dialysed eluates were then incubated with cOmplete His-Tag Purification Resin (Roche) for removal of TEV protease and undigested proteins, and flowthrough was centrifuged at 23,000 rcf for 30 min and filtered through a 0.22 μm membrane to remove particulate matter. The UV spectra showed no significant absorption at 260 nm and were used to quantify purified proteins, which were then normalized and their quality checked with PAGE and Coomassie staining (Fig. 4a). Concentrations used in EMSAs were: Halo-TRA2B[RRM] (lanes 2–6: 3.6, 7.2, 14.4, 57.6 and 102.4 μM, respectively); Halo-SAFB[RRM] (lanes 7–11: 3.6, 7.2, 14.4, 57.6 and 102.4 μM, respectively); and Halo (lane 12: 102.4 μM). Lane 1 contained only those probes with no added protein.

The RNA probes were prepared by in vitro transcription. Briefly, a plasmid containing the relevant sequence TAATACGACTCACT ATAGGGAAGAAGAAGAAGAAGAAGAAGAT^ATC, in which the T7 promoter sequence is underlined, was digested with EcoRV (site of digestion, indicating that the last nucleotide of the final RNA is marked− indicated by ^), purified and in vitro transcribed using a HighYield T7 RNA Synthesis Kit (Jena Biosciences, no. RNT-101) with either 1 mM (final) CTP/UTP/GTP/ATP or 1 mM CTP/UTP/GTP and 1 mM N6-Methyl-ATP (Jena Biosciences, no. RNT-112-S), completely replacing ATP. RNA was cleaned up using SPRI beads to remove the plasmid and other potential high-molecular-weight products, then with the OCC-5 kit (Zymo). RNA was then oxidized using freshly prepared sodium periodate (250 mM in water, final concentration 10 mM; Sigma, no. 311448) in 60 mM NaOAc pH 5.5 for 1 h on ice, with tubes kept in the dark. After a further clean-up with OCC-5, RNA was then labelled with CF 647 Hydrazide (Sigma, no. SCJ4600046; 10 mM in water, 0.8 mM final concentration in approximately 120 mM NaOAc, pH 5.5) at room temperature overnight. RNA was purified with OCC-5, eluted in water and normalized to 5 μM. EMSAs were carried out in 25 mM Tris.Cl pH 7.4, 50 mM KCl, 5% glycerol and 1 mM DTT with an RNA probe of around 100 nM and the indicated concentration of the protein of interest. Following incubation of RNA and proteins on ice for 30 min, mixtures were loaded directly on a Nature 8% polyacrylamide gel cast with 0.5× Tris-borate-EDTA (final) and run in 0.5× Tris-borate-EDTA in a cold room for 45 min at 100 V (gels were pre-run at 100 V for 15 min). Proteins and RNA were sequentially visualized

on the same gel using a Typhoon Scanner with appropriate excitation lasers and emission filters.

### In vitro unmethylated and methylated RNA-binding assay

Nuclei were isolated from wt-HCT116 cells using a buffer containing 0.5% Igepal CA-630, following Lubelsky and Ulitsky[57], and snap-frozen in liquid nitrogen until use. Nuclei were resuspended with 500 μl of 25 mM Tris.Cl pH 7.4, 150 mM KCl, 2 mM MgCl$_2$, 0.5% Igepal CA-630, 5% glycerol, 5 mM β-mercaptoethanol, 1× protease inhibitors and 1× PhosS-TOP and sonicated with a Branson sonifier. Next, 15 μl of TURBO-DNase was added followed by incubation at 25 °C for 20 min and then by the slow addition to the lysate of 1.5 m of base buffer (25 mM Tris.Cl pH 7.4, 50 mM KCl, 5% glycerol) to bring the KCl concentration to 75 mM and Igepal CA-630 concentration to 0.125% (final). Lysate was incubated with 50 μl of Pierce Control Agarose Resin (no. 26150) for 20 min, with rotation in a cold room, and spun down at full speed for 10 min at 4 °C to remove insoluble material. A 949 bp fragment of L1 ORF2 was amplified from pYX017 using primers AATAATACGACTCACTATAGCGT ATCACCACCGATCCCACAG (T7 promoter underlined) and GGCTGAG ACGATGGGGTTTT and in vitro transcribed using a HighYield T7 RNA Synthesis Kit (Jena Biosciences, no. RNT-101) with either 1 mM (final) CTP/UTP/GTP/ATP or 1 mM CTP/UTP/GTP and 1 mM N6-methyl-ATP (Jena Biosciences, no. RNT-112-S), completely replacing ATP. RQ1 DNase (Promega) was added to each reaction with incubation for for 20 min at 37 °C, after which RNA was cleaned up using RCC-25 (Zymo) and oxidized with freshly prepared sodium periodate (250 mM in water, final concentration 10 mM; Sigma, no. 311448) in 60 mM NaOAc pH 5.5 for 1 h on ice, with tubes kept in the dark. After a further clean-up with RCC-25, RNA was then labelled with biotin Hydrazide (Sigma, no. 87639; 50 mM in DMSO, 2 mM final concentration in approximately 120 mM NaOAc, pH 5.5) at room temperature overnight. RNA was purified with RCC-25, eluted in water and quantified with Nanodrop, then 5 μg of each RNA or buffer was incubated with 25 μl of MyONE C1 streptavidin beads in base buffer + 0.1% Igepal CA-630 for 1 h at room temperature and washed twice with base buffer + 0.1% Igepal CA-630. The nuclear lysate was incubated with these beads for 1 h at 16 °C, with shaking at 1,100 rpm. Beads were washed and transferred from fresh tubes with base buffer + 0.1% Igepal CA-630. Proteins bound to the beads were eluted with base buffer + 0.1% Igepal CA-630 + 2 μl of RNaseA + T1 (no. EN0551, Thermo Fisher Scientific) for 30 min at 30 °C and demonstrated by immunoblotting.

### RNA blotting

HCT116 cells were transfected with 5 nM siRNA (as indicated in Fig. 2h) then, 48 h later, were either transfected with a plasmid encoding L1Hs and driven by a minimal EF1alpha (without an intron) promoter or mock transfected. Twenty-four hours later (72 h post siRNA transfection), cells were trypsinized and resuspended with a buffer containing 0.5% Igepal CA-630, essentially as described in ref. 57. The cytoplasmic fraction was purified with RNA Clean & Concentrator columns (Zymo), 2 μg of which was loaded onto 1.2% agarose gel and electroblotted to a nylon membrane. DIG-labelled probes against ORF2 were prepared with in vitro transcription (see Supplementary Table 2 for primers) and probe hybridization, washes and immunodetection were carried out as described in the manual of the DIG Northern Starter Kit (Roche, no. 12 039 672 910).

### p-SR (1H4) and DHX9 FLASH in SAFB-depleted cells

Flp-In T-REx HEK293 cells were transfected with either control siRNA or siRNAs against SAFB1, SAFB2 and SLTM 48 h following transfection, then washed with PBS and UV-crosslinked with 0.2 mJ cm$^{-2}$ UV-C light on ice. Nuclei were isolated as described in ref. 57, resuspended in 1× NLB + 5 mM MgCl$_2$ with protease and phosphatase inhibitors and sonicated using a Branson sonifier. Following centrifugation, to remove insoluble material the supernatant was incubated with an agarose resin

(Pierce, no. 26150) for 20 min in a cold room followed by further incubation with Dynabeads Protein G beads prebound to p-SR antibody (10 μl per IP; 1H4, Santa Cruz, no. sc-13509) for 90 min in a cold room. The supernatant from 1H4 IP was used for DHX9 IP (2.5 μl per IP; abcam, no. ab26271). The FLASH protocol was identical to that described above, except that all HSB washes were replaced with NLB and s-oligos were pre-dephosphorylated to skip the recombinant shrimp alkaline phosphatase treatment that could dephosphorylate SR proteins on the beads, potentially leading to their elution.

## RIP–qPCR
Flp-In T-REx HEK293 cells were crosslinked with 0.2% formaldeyhde for 10 min at room temperature, extensively washed with PBS, resuspended with 1× NLB and sonicated using a Branson sonifier. The lysate was centrifuged at 23,000 rcf for 10 min at 4 °C to remove insoluble material and the supernatant then incubated with an agarose resin (Pierce, no. 26150) for 30 min in a cold room. Following brief centrifugation, the supernatant was used for IP with Dynabeads Protein G beads coupled to either an antibody against SAFB1 (10 μl per IP; Santa Cruz, no. sc-393403) or control IgG (Santa Cruz, no. sc-2025) overnight in a cold room. Beads were washed with 1× NLB and bead-bound RNA was eluted with proteinase K, as described above, purified using RCC-5 (Zymo) and utilized for RT–qPCR.

## Generation of the *Dnmt3c*-null allele
*Dnmt3C* knockout animals were generated as described in ref. 58. For specific abolition of enzymatic activity we designed a sgRNA against the methyltransferase domain of *Dnmt3C* targeted to exon 15 with the following protospacer sequence: 5′-GGACATCTCACGATTCCTGG-3′. P0 animals were genotyped using Sanger sequencing following PCR with primers 5′-CTGGCCGGCTCTTCTTTGAG-3′ and 5′-GGAAATCATTCCCACCTGTCAGC-3′. The founding animal was chosen based on a 31 bp deletion, which resulted not only in a frameshift mutation beginning at codon 598 but simultaneous removal of a *Pfo*I restriction enzyme digestion site for straightforward genotyping. The founder mutation was subsequently backcrossed into the C57BL/6 J background. Homozygous knockout males were validated as infertile, with significantly smaller and disordered testes by P42, as reported previously[51]. The generation of these experimental animals was regulated following ethical review by Yale University Institutional Animal Care and Use Committee (protocol no. 2020-20357) and was performed according to governmental and public health service requirements. No sample size selection, randomization or blinding was performed.

## Direct antibody labelling
The Mix-n-Stain CF488 A Antibody Labelling Kit (Biotium, no. 92253) and Mix-n-Stain CF555 Antibody Labelling Kit (Biotium, no. 92254) were used to label rabbit antihuman SAFB1/SAFB antibody (LSBio, LS-C286411) and rabbit anti-LINE-1-ORF1p antibody (abcam, no. ab216324), respectively. The standard protocol listed on the product website was followed, including the ultrafiltration protocol, with minor modifications. In brief, 25–35 μg of antibody was placed in the ultrafiltration vial provided and centrifuged at 14,000*g* for 2 min to remove all liquid. Depending on the initial amount of antibody, antibodies were eluted in 1× PBS to a final concentration of 0.75 ng μl$^{-1}$ and the appropriate volume of 10X Mix-n-Stain Reaction Buffer added. The entire solution was transferred to the vial containing the dye and the labelling reaction allowed to proceed at room temperature (22–23 °C) in the dark for 30 min. Finally, 150 μl of storage buffer was added to each reaction with storage in aliquots of 50 μl at −20 °C until use.

## Testis sectioning and Immunofluorescence microscopy
Testes from P25 *Dnmt3C* homozygous and heterozygous mutant males were dissected and embedded in O.C.T. compound (Tissue-Tek). Using cryosectioning, 8 μm sections were obtained with a Leica CM3050S and spotted onto Fisherbrand Superfrost Plus Microscope Slides (Fisher Scientific, no. 12-550-15) and stored at −80 °C until use. For immunofluorescence detection, slides were thawed at room temperature for over 10 min before fixing in 4% paraformaldehyde for 8 min. Permeabilization and blocking were performed at room temperature for 1 h with blocking buffer (5% bovine serum albumin (BSA), 0.2% Triton X-100 and PBS). Sections were incubated with directly labelled antibodies overnight at 4 °C, followed by three 5 min washes in 1× PBS and mounting with VECTASHIELD PLUS Antifade Mounting Medium and DAPI (Vector Laboratories, no. H-2000). Images were acquired using a Leica THUNDER Imaging System at ×40 magnification.

## Mass spectrometry
Flp-In T-REx HEK293 cells stably expressing SAFB1, SAFB2 or SLTM (same cell lines used for FLASH) were induced with 0.1 μg ml$^{-1}$ DOX for 16 h in triplicate, lightly crosslinked with formaldehyde (0.016% final) at room temperature for 10 min, extensively washed with PBS, resuspended with HMGT-K200 buffer (25 mM HEPES-KOH pH 7.4, 10 mM MgCl$_2$, 10% glycerol, 0.2% Tween-20) and homogenized using a water bath sonicator. Following centrifugation, supernatants were then incubated with MyONE C1 streptavidin beads to pull down proteins of interest. Beads were washed with HMGT-K200, 20 mM Tris-Cl pH 7.4 and 1 M NaCl and finally with 20 mM Tris-Cl pH 7.4 and 50 mM NaCl, then submitted to the in-house MS-facility for further processing. Silver gel staining was performed using a SilverQuest Silver Staining Kit (Thermo Fisher Scientific, no. LC6070) for SAFB1 to ensure that conditions were sufficiently stringent in comparison with GFP pulldown (Extended Data Fig. 7b).

## On-beads digest and mass spectrometry analysis
Twelve samples were boiled at 95 °C and 500 rpm for 10 min, followed by tryptic digest including reduction and alkylation of cysteines. The reduction was performed by the addition of tris(2-carboxyethyl)phosphine at a final concentration of 5.5 mM at 37 °C on a rocking platform (500 rpm) for 30 min. To perform alkylation, chloroacetamide was added at a final concentration of 24 mM at room temperature on a rocking platform (500 rpm) for 30 min. Proteins were then digested with 200 ng of trypsin (Roche) per sample, shaking at 800 rpm and 37 °C for 18 h. Samples were acidified by the addition of 1.3 μl of 100% formic acid (2% final concentration), centrifuged and placed on a magnetic rack. Supernatants containing the digested peptides were transferred to a new low-protein-binding tube. Peptide desalting was performed on self-packed C18 columns in a Tip. Eluates were lyophilized and reconstituted in 19 μl of 5% acetonitrile and 2% formic acid in water, briefly vortexed and sonicated in a water bath for 30 s before injection into nano-liquid chromatography–tandem mass spectrometry (nano-LC–MS/MS).

## LC–MS/MS instrument settings for shotgun proteome profiling and data analysis
LC–MS/MS was carried out by nanoflow reverse-phase liquid chromatography (Dionex Ultimate 3000, Thermo Scientific) coupled online to a Q-Exactive HF Orbitrap mass spectrometer (Thermo Scientific), as reported previously[59]. In brief, LC separation was performed using a PicoFrit analytical column (75 μm internal diameter × 50 cm length, 15 μm Tip ID; New Objectives) and packed in house with 3 μm of C18 resin (Reprosil-AQ Pur, Dr Maisch). Peptides were eluted using a gradient from 3.8 to 38% solvent B in solvent A over 120 min at a flow rate of 266 nl min$^{-1}$. Solvent A was 0.1% formic acid and solvent B comprised 79.9% acetonitrile, 20% H$_2$O and 0.1% formic acid. Nanoelectrospray was generated by the application of 3.5 kV. A cycle of one full Fourier transformation scan mass spectrum (300–1,750 *m/z*, resolution 60,000 at *m/z* 200, automatic gain control target 1 × 10$^6$) was followed by 12 data-dependent MS/MS scans (resolution of 30,000, automatic gain control target 5 × 10$^5$) with a normalized collision energy of 25 eV.

To avoid repeated sequencing of the same peptides, a dynamic exclusion window of 30 s was used.

Raw MS data were processed with MaxQuant software (v.1.6.17.0) and searched against the human proteome database UniProtKB UP000005640 (containing 75,074 protein entries, released May 2020). The parameters of MaxQuant database searching were a false discovery rate of 0.01 for proteins and peptides, a minimum peptide length of seven amino acids, a first-search mass tolerance for peptides of 20 ppm and a main search tolerance of 4.5 ppm. A maximum of two missed cleavages was allowed for the tryptic digest. Cysteine carbamidomethylation was set as a fixed modification whereas N-terminal acetylation and methionine oxidation were set as variable modifications. The MaxQuant-processed output files can be found in Supplementary Table 3, showing peptide and protein identification, accession numbers, percentage sequence coverage of the protein and $q$-values.

## IP
Native whole-cell extracts prepared using 0.5× NLB were incubated with ProtG Dynabeads (Life Technologies, no. 10004D) coupled to 1 µg of either SAFB antibody ('Antibodies') or IgG (mouse; Santa Cruz, no. sc-2025) in a cold room for 150 min. Beads were washed twice in 0.5× NLB for 5 min then once with NDB. RNase-treated samples were resuspended in 90 µl of NDB to which 10 µl of RNaseA + T1 mix (Thermo Scientific, no EN0551) was added. Samples were then incubated at 20 °C for 15 min and washed twice with 0.5× NLB. Elution from the beads was performed in 1× protein-loading dye by incubation for 5 min at 95 °C with shaking. Interaction partners were detected using the antibodies against proteins shown in Extended Data Fig. 7 ('Antibodies').

## Immunofluorescence
Cells were crosslinked with 4% methanol-free formaldehyde in PBS at room temperature for 10 min, permeabilized with 0.5% Triton X for 10 min then blocked with 5% BSA in PBS for 30 min at room temperature. Primary antibodies (further details in 'Antibodies') were diluted in PBS with 0.1% Triton X and 1% BSA and incubated with fixed cells at 4 °C for about 16 h. Fluorescently labelled secondary antibodies with the appropriate serotype were used to demonstrate target proteins. Hoechst 33342 was used to stain DNA.

## Antibodies
The following antibodies were used: AFB1 (Santa Cruz, no. sc-393403), SAFB2 (Santa Cruz, no. sc-514963), SAFB1/2 (HET) (human: Merck/Sigma-Aldrich, no. sc05-588; mouse: LSBio, no. LS-C2886411), SLTM (Invitrogen, no. PA5-59154), ORF1p (human: abcam, no. ab230966; mouse: abcam, no. ab216324), TASOR (Sigma-Aldrich, no. HPA006735), 1H4 (p-SR) (Merck/Sigma-Aldrich, no. MABE50), RBM12B (Bethyl, no. A305-871A-T), RBMX (Cell Signaling Technology, no. 14794 S), NCOA5 (Bethyl, no. A300-790A-T), ZNF638 (Sigma-Aldrich, no. ZRB1186), ZNF326 (Santa Cruz, no. sc-390606), TRA2B (Bethyl, no. A305-011A-M), U2AF2 (U2AF65; Santa Cruz, no. sc-53942), TUBULIN (Santa Cruz, no. sc-32293), SRRM1 (abcam, no. ab221061), SRRM2 (SC35) (Sigma-Aldrich, no. S4045), SON (Sigma-Aldrich, no. HPA023535), DHX9 (abcam, no. ab183731), U1-70K (SySy, no. 203011), PRP8 (Santa Cruz, no. sc-55533), RNAPII (Creative Biolabs, no. CBMAB-XB0938-YC), IgG normal mouse (Santa Cruz, no. sc-2025), SRSF1 (Santa Cruz, no. sc-33652), SRSF2 (abcam, no. ab204916), SRSF3 (Elabscience, no. E-AB-32966), SRSF7 (MBL, no. RN079PW), RB1 (Cell Signaling Technology, no. 9309 S), TRA2B (Santa Cruz, no. sc-166829) and YTHDC1 (Proteintech, no. 14392-1-AP).

## TE expression analysis
RNA-seq data from human (HEK293, HeLa, HCT116), mouse (3T3) and *Drosophila* (S2) cells were mapped to their respective genome (hg38, mm10 and dm6, respectively) using the snakePipes non-coding-RNA-seq pipeline[60]. Internally this pipeline uses TEtranscripts[23], which estimates both gene and TE transcript abundance in RNA-seq data and conducts differential expression analysis on the resultant count tables, which is carried out by DESeq2 (ref. 61). The outputs of this analysis can be found in Supplementary Tables 4–11.

## SAFB peak annotation and TE enrichment
Overlapping SAFB1, SAFB2 and SLTM regions called by HOMER on FLASH data were merged using the function IRanges::reduce(), resulting in a single set of 29,806 SAFB-bound genomic intervals (SAFB peaks), 23,136 of which were located inside GENCODE-annotated genes (within-gene SAFB peaks). All GENCODE v.29 genes located on standard chromosomes were used as a control set ($n$ = 58,721). repeatMasker annotation was downloaded from the UCSC genome browser, and the fraction of total length contributed by different transposable elements was calculated for 23,136 SAFB peaks and 58,721 GENCODE-annotated genes, separately for TEs inserted in sense and antisense orientation. Enrichment was calculated for a subset of sense and antisense TEs by dividing the TE fraction in peaks (that is, observed TE fraction) by that in whole genes (that is, fraction expected if SAFB peaks were distributed randomly on transcripts), followed by $\log_2$-transformation of values.

## Short-read RNA-seq data analysis
Raw RNA-seq reads were subject to adaptor and quality trimming using cutadapt 4.1. Default options were used, except for -q 16 --trim-n -m 25 -a AGATCGGAAGAGC -A AGATCGGAAGAGC.

Trimmed reads from human and mouse cell lines were mapped to human GRCh38 (HEK293, HeLa and HCT116 cell lines) and mouse GRCm38 (3T3 cell line) genomes using the STAR 2.7.9a aligner[62]. To improve the sensitivity of spliced read detection and quantification, mapping was done in two passes. In the first pass, all reads were mapped simultaneously to the STAR genome index built with GENCODE gene models (v.29 for human, v.19 for mouse) using default options, with the exception of --outFilterMismatchNoverReadLmax 0.05 --outSAMtype None. In the second pass, each sequenced library was mapped to a genome index with GENCODE gene models extended with new splice junctions detected in the first pass (--sjdbFileChrStartEnd pass1.SJ.out.tab). Other non-default STAR options used included --outFilterMismatchNoverReadLmax 0.05 --quantMode Gene-Counts --alignIntronMax 1000000 --alignMatesGapMax 2000000 --sjdbOverhang 100 --limitSjdbInsertNsj 2000000.

Trimmed reads from the fruitfly S2 cell line were mapped to the dm6 genome assembly using STAR 2.7.4a, and reads were counted using featureCounts (subread package v.2.0.0).

## Differential gene expression
Differential gene expression analysis was performed using the DESeq2 package[61] on reverse-stranded gene counts from the STAR alignment step. Genes with fewer than ten mapped reads were discarded; lfcThreshold = 1 and alpha = 0.05 were used for calling of differentially expressed genes, and results were shrunk using lfcShrink(…, type = "ashr").

## Differential exon usage
To avoid assignment of exonic reads to SAFB peaks, within-gene SAFB peak fragments or entire peaks overlapping GENCODE v.29-annotated exons were masked and ignored in exon usage analysis. The 22,129 peaks remaining (intronic SAFB peaks) were assigned to their host genes and RNA-seq reads were counted on both annotated exons and intronic SAFB peaks using the function Rsubread::featureCounts() with default arguments, except for countMultiMappingReads = FALSE, strandSpecific = 2, juncCounts = TRUE, and isPairedEnd = TRUE. Differentially expressed SAFB peaks were identified using the DEXSeq R package[63] and, for each gene, the peak with the lowest DEXSeq $P$ value was used as a reference for gene fragmentation. In total, 5,394 affected genes

were fragmented into pre- and post-peak parts. Exonic read counts were aggregated separately for pre- and post-peak fragments and their differential expression measured using DESeq. Genes hosting SAFB peaks with DEXSeq $P_{adjusted} < 0.05$ and log-fold change above 2 were classified as (genes with) upregulated peaks ($n = 878$) whereas those hosting peaks with DEXSeq $P_{adjusted} > 0.05$ and log-fold change between $-0.5$ and $0.5$ were used as the control set ($n = 1,457$).

## Differential splice junction usage

The number of RNA-seq reads supporting each splice junction was counted in the second STAR alignment pass (SJ.out.tab file). Splice junctions that could not be unambiguously assigned a host gene, or that were supported by fewer than ten reads in total across all treatments and replicates in a given cell line, were ignored. Differentially used splice junctions were identified using DEXSeq, with default settings; splice junctions were treated as feature IDs and host genes as group IDs.

## Splice site strength quantification

For each gene in the human genome, the probability of each nucleotide acting as a splice donor or acceptor was estimated using SpliceAI[26], with default options. SpliceAI scores were matched to splice junctions detected and quantified by STAR.

## Splice site to TE distance measurement

Distances between splice sites and nearest upstream or downstream TEs were calculated for a set of ten repeat families (L1, L2, Alu, SVA, ERVL, ERV1, TcMar-Tigger, MIR, Simple_repeat, hAT_Charlie) as follows: (1) all GENCODE genes were flattened using the function IRanges::reduce() in R; (2) STAR-detected splice junctions and repetitive elements outside annotated genes were dropped; and (3) for each remaining splice donor and acceptor, the distance (in nucleotides) to the nearest sense or antisense TE within the same flattened gene was measured separately for each of the ten repeat families. Donors and acceptors within TEs were assigned the distance of 0 nt.

## New splice acceptors within SAFB peaks in human tissues

The number of reads supporting splice junctions in the GTEx consortium tissue data was extracted using the recount3 R package[64]. Tissues with fewer than 1 billion spliced reads were excluded from further analysis. Alternative splicing was quantified in an intron-centric manner—that is, splicing index was calculated separately for each splice donor and acceptor. We extracted all splice junctions located within an annotated human gene, with splice donor annotated in GENCODE v.29 and splice acceptor sited within a fully intronic SAFB peak ($n_{peaks} = 16,929$). A further 21,693 such splice junctions were filtered for junction where the donor participated in multiple events, had a splicing index above 1% in at least one tissue and was supported by at least 500 reads in all 27 tissues (that is, used ubiquitously), resulting in a highly stringent set of of 1,104 splice junctions.

## p-SR and DHX9 FLASH analysis

FLASH reads uniquely mapping to the hg38 genome were counted using featureCounts on two custom gene annotation reference sets. The first of these contained exons and SAFB peaks, with exons prioritized over SAFB peaks in the case of overlaps. SAFB peaks were assigned to their host genes and treated as exons for read counting. The second reference contained genes fully fragmented into exons, repetitive elements and introns, with exons prioritized over repeats and introns, and repeats prioritized over introns where their genomic coordinates were overlapping. Whereas the first reference allows for increased sensitivity when quantifying FLASH signal on known SAFB-binding regions, the latter sacrifices sensitivity (because it contains many short genomic fragments) for the power of recognizing regions of increased binding outside of SAFB peaks, or in SAFB peaks not called by the peak-calling software. DEXSeq analysis was performed separately on exon/peak and exon/repeat/intron counts. Regions with adjusted $P < 0.05$ were considered differentially bound.

## Alternative polyadenylation sites

Aligned ONT direct RNA-seq performed on control and triple KD samples was screened for their end coordinates, under the assumption that these are derived from the close proximity of a polyadenylation site. Genomic coordinates of this collection of almost 1.5 million single-nucleotide-resolution read end sites were extended by 50 nt upstream and downstream, and overlapping intervals were collapsed into a total of 274,330 putative polyadenylation regions. The number of control and triple KD reads ending in each of these regions was counted and, for each gene, the fraction of ONT reads ending in each of its polyadenylation regions was calculated separately for control and triple KD libraries. Genes supported by at least 20 reads in which the contribution of at least one polyA isoform was changed by at least 20 percentage points between triple KD and control were considered differentially polyadenylated. In total, 14,148 genes (4,433 of genomic length over 50 kb) were supported by 20 or more reads, and 247 (231 longer than 50 kb) showed differential polyA site usage.

## Locus-specific L1 quantification

Raw reads from HEK293 fractionation RNA-seq libraries were aligned to the hg38 genome using bwa aln, and alignments further processed with L1EM[65], both with default options. L1EM counts from categories 'only', '3prunon' and 'passive_sense' were summed. These total read counts were combined with read counts on individual genes (GENCODE v.29 annotation), and DESeq2 differential gene expression analysis was performed together on gene and L1 counts, treating L1 elements as independent genes.

## Reporting summary

Further information on research design is available in the Nature Portfolio Reporting Summary linked to this article.

## Data availability

RNA-seq and FLASH data are available under GSE223263. Source data are provided with this paper.

## Code availability

The code is available at https://github.com/aktas-lab/safb_paper.

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

**Acknowledgements** We thank all Aktaş group members for their input on the manuscript. We thank W. An for sharing the retrotransposition luciferase reporter constructs with us. We also thank the Sequencing Facility at MPI-MG and members of the Microscopy

and Cryo-electron Microscopy facility at MPI-MG, especially T. Mielke, B. Fauler and R. Buschow. N2A cells were a gift from M. Kraushar. Figure 4a and Extended Data Figs. 7a and 10i were created with BioRender.com. Z.D.S. is funded by an NIH New Innovator Award (no. DP2HD108774), the Mathers Foundation and a Chen Innovation Award. K.T. is funded by a Gruber Science Fellowship. B.B. and F.-J.M. were supported by Deutsche Forschungsgemeinschaft (DFG, German Research Foundation) under Germany's Excellence Strategy (no. EXC 22167-390884018) and by the German Federal Ministry for Research and Education (BMBF IntraEpiGliom, no. FKZ 13GW0347C). Research in the laboratory of T.A. is funded by the Max-Planck Society. P.G. is paid from project no. 456668871 of DFG (given to T.A.). Computational analysis pipelines generated for the DFG project were utilized in this study.

**Author contributions** I.A.I. and T.A. conceived the study and designed the research. I.A.I carried out all experiments except for long-read direct RNA-seq, which was performed by B.B. and supervised by F.-J.M. K.T. and Z.D.S. generated the *Dnmt3C* mouse model and performed testis sectioning and staining experiments. D.M. performed mass spectrometry experiments (with assistance from B. Lukaszewska-McGreal) and analysed the data. I.A.I. and P.G. performed the computational analysis of data under the supervision of T.A. I.A.I., P.G. and T.A. wrote the manuscript.

**Funding** Open access funding provided by Max Planck Society.

**Competing interests** I.A.I. and T.A. are inventors on a patent application (no. EP3325621B1, European Patent Office) regarding the s-oligo design used in FLASH experiments. Z.D.S. is a cofounder and scientific advisor of Harbinger Health.

### Additional information
**Correspondence and requests for materials** should be addressed to Tuğçe Aktaş.

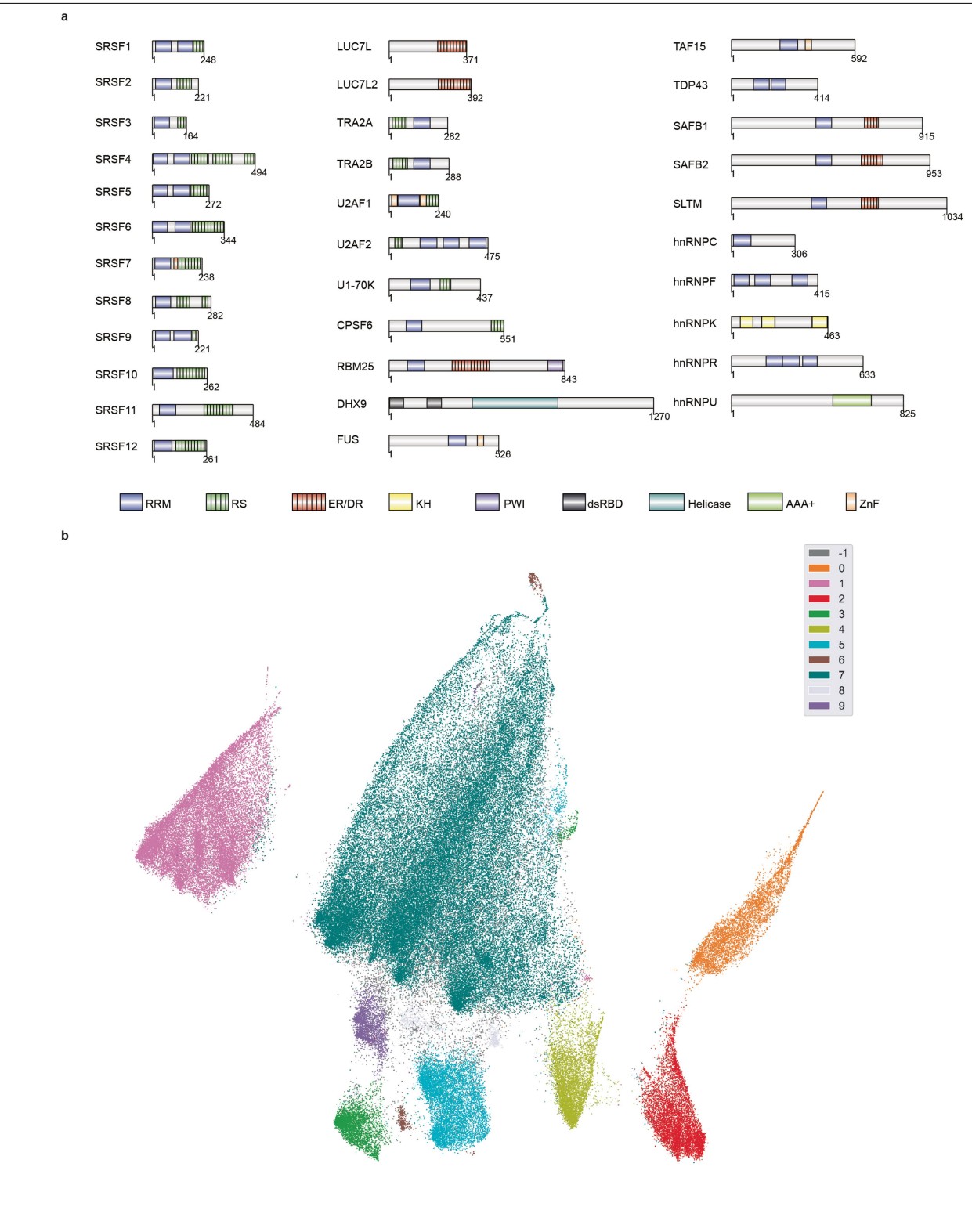

**Extended Data Fig. 1 | RNA-binding proteins profiled using FLASH.**
**a**, Schematics showing the prominent domains of the 33 RBPs profiled using FLASH. Some RRM domains are likely to be quasi-RRM domains, for example the second RRM domain of SRSF1, and all three RRM domains of hnRNPF[66]. **b**, Clusters identified on the UMAP projection of all FLASH data.

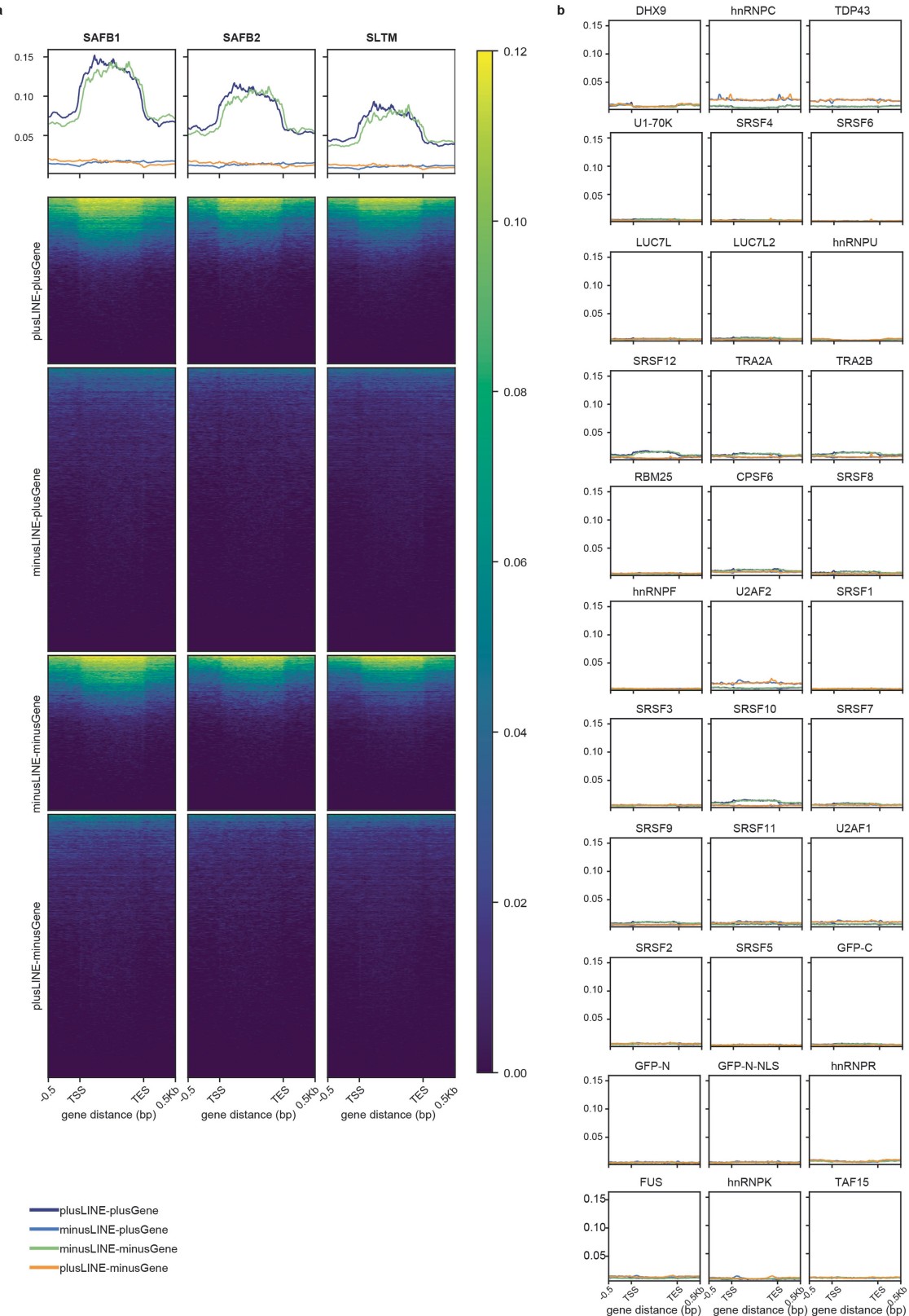

**Extended Data Fig. 2 | SAFB proteins bind to L1 elements only when they are inserted on the same strand as the host gene.** Cumulative coverage was calculated for all RBPs profiled for FLASH on all intronic L1 insertions +/− 500 bp on each side. Targets are split into four: i) genes on the plus strand, plus strand L1 insertions ii) genes on the plus strand, minus strand L1 insertions iii) genes on the minus strand, minus strand L1 insertions iv) genes on the minus strand, plus strand L1 insertions. **a**, SAFB1, SAFB2 and SLTM. **b**, other RBPs, only profiles are shown.

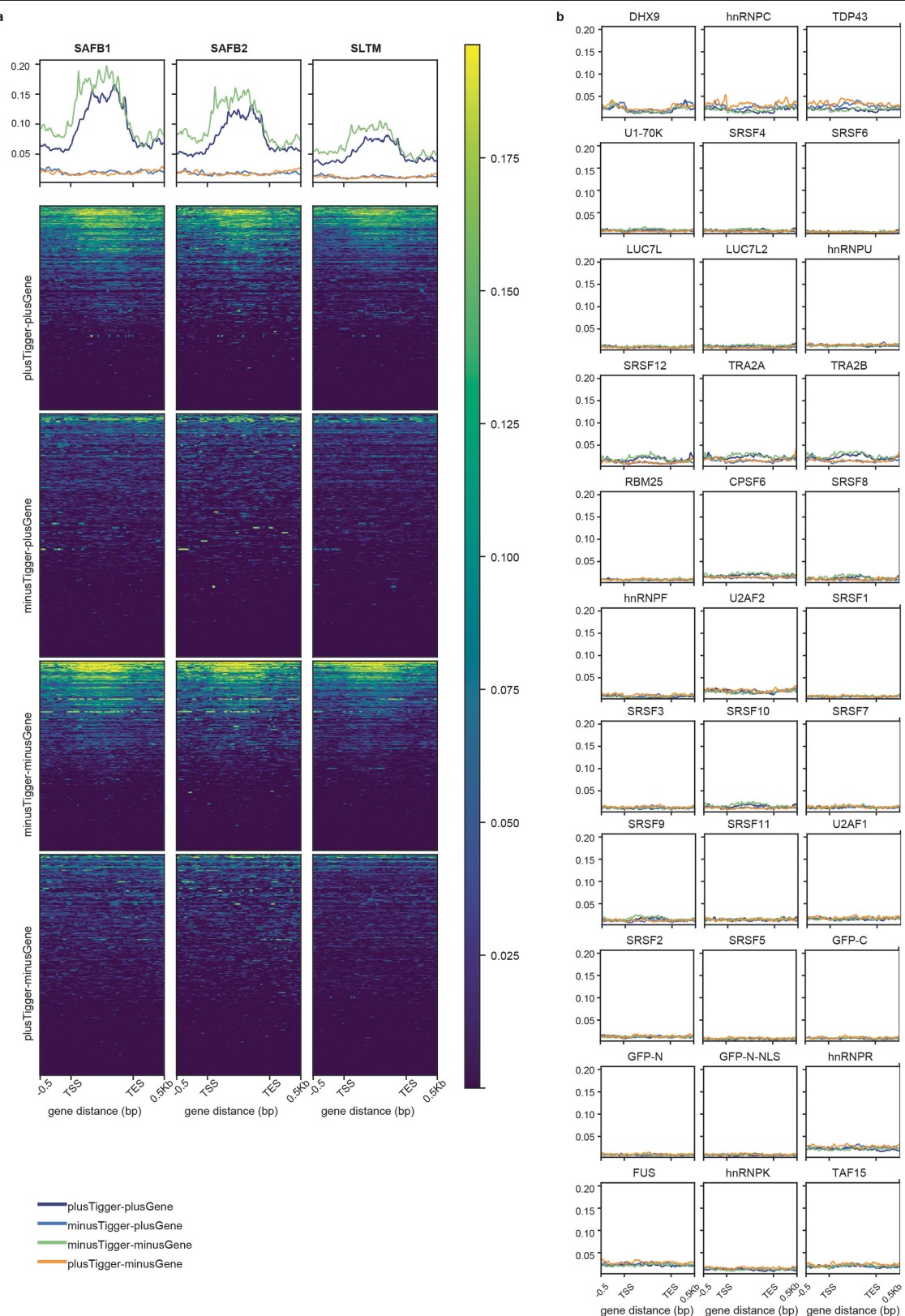

**Extended Data Fig. 3 | SAFB proteins bind to Tigger elements only when they are inserted on the same strand as the host gene.** Cumulative coverage was calculated for all RBPs profiled for FLASH on all intronic Tigger insertions +/− 500 bp on each side. Targets are split into four: i) genes on the plus strand, plus strand Tigger insertions ii) genes on the plus strand, minus strand Tigger insertions iii) genes on the minus strand, minus strand Tigger insertions iv) genes on the minus strand, plus strand Tigger insertions. **a**, SAFB1, SAFB2 and SLTM. **b**, other RBPs, only profiles are shown.

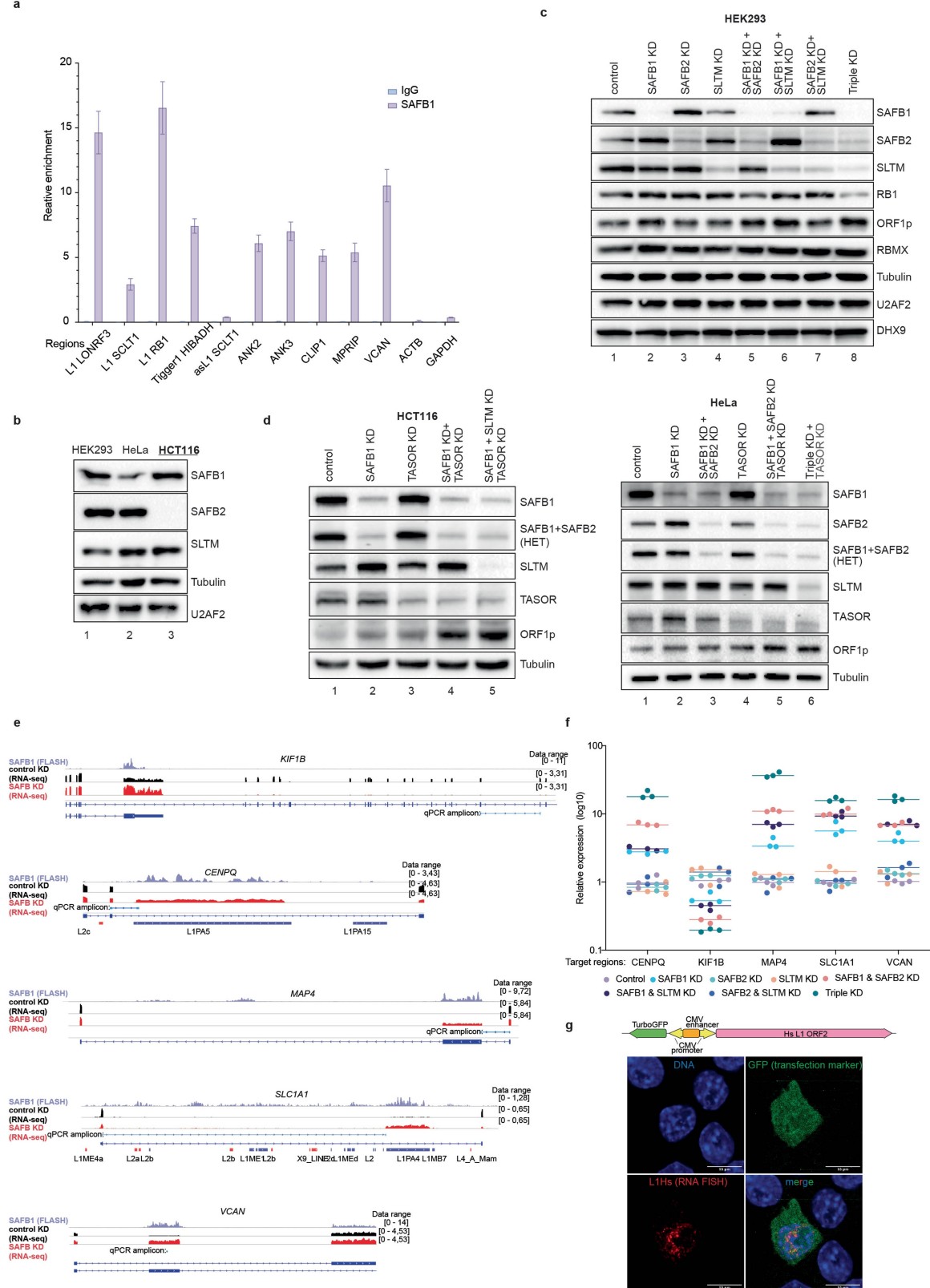

**Extended Data Fig. 4** | See next page for caption.

**Extended Data Fig. 4 | Validation of SAFB transcriptomic data. a**, RT-qPCR of RNA isolated from either SAFB1 immunoprecipitation of IgG control using 0.2% formaldehyde crosslinked HEK293 lysates. The plot shows the average of three replicates, error bars show SD of three replicates. Primers against transposons are designed to be unique to the locus, see Supplementary Table 1 for primer sequences. **b**, Immunoblots showing expression of SAFB1, SAFB2 and SLTM in HEK293, HeLa and HCT116 cells. **c**, Immunoblots showing protein levels of SAFB1, SAFB2 and SLTM in single, double and triple siRNA transfections. Note the increase in SAFB2 expression in SAFB1 siRNA transfected cells (lane 2 vs 1) and the increase in SAFB1 expression in SAFB2 siRNA transfected cells (lane 3 vs 1) Also see Supplementary Fig. 5c for the L1PA7 inclusion event that likely attenuates RB1 expression. **d**, Immunoblots showing increase in SLTM expression in SAFB1 depleted HCT116 cells which do not express SAFB2, and ORF1p expression is highest when HUSH complex and SAFB proteins are co-depleted (also see Fig. 2g,h). **e**, The amplicons used to interrogate the hierarchy of SAFB proteins on suppressing splicing events that are detected from RNA-seq data. **f**, Hierarchy of SAFB proteins in regulating splicing of select targets using depletion conditions shown in (c) and amplicons shown in (e). **g**, Specificity of the L1 RNA probe is shown with RNA FISH in mouse N2A cells transfected with an L1Hs sequence containing plasmid that also harbors a GFP as a transfection marker. Scale bar: 10 μm.

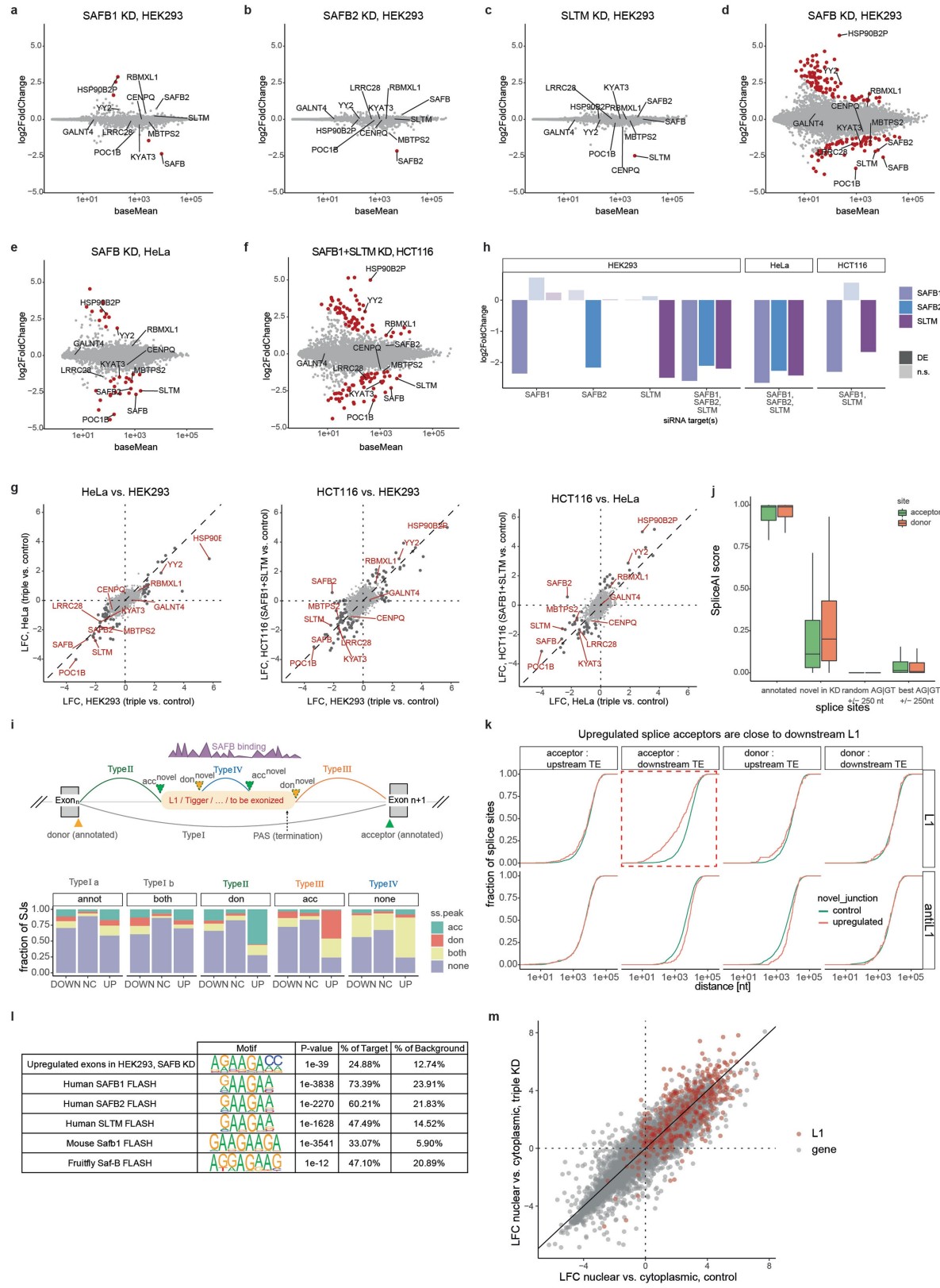

**Extended Data Fig. 5 |** See next page for caption.

**Extended Data Fig. 5 | Gene expression changes and differential splicing in SAFB1, SAFB2, SLTM depleted cells.** MA plots showing gene expression changes upon. **a**, SAFB1 KD in HEK293 cells, **b**, SAFB2 KD in HEK293 cells, **c**, SLTM KD in HEK293 cells, **d**, SAFB1 + SAFB2 + SLTM (SAFB) KD in HEK293 cells, **e**, SAFB1 + SAFB2 + SLTM (SAFB) KD in HeLa cells. **f**, SAFB1 + SLTM (SAFB) KD in HCT116 cells. **g**, SAFB KD, HeLa vs HEK293 cells, HCT116 vs HEK293 and HCT116 vs HeLa cells. Only genes supported by on average 50 reads per library were plotted to avoid LFC-shrinking artefacts. **h**, Levels of each SAFB protein in each depletion for each cell line as detected in RNA-seq data. **i**, Top, schematic describing different categories of splice-junctions in SAFB depleted cells. Type Ia: both splice sites, as well as the splice-junction is annotated; Type Ib: both splice sites are annotated, but junction is novel, Type II: donor splice site is annotated, acceptor is novel; Type III: acceptor splice site is annotated, donor novel; Type IV: both splice acceptor and novel sites are novel. Bottom, fraction of downregulated (DOWN), unchanged (NC) and upregulated (UP) splice-junctions in SAFB depleted cells, split by categories described above. Green: acceptor site is within a SAFB peak, orange: donor site is within a SAFB peak, yellow: both acceptor and donor contained within a SAFB peak, purple: neither acceptor nor donor splice site overlaps with a SAFB peak. **j**, SpliceAI scores for splice donors and acceptors in annotated splice junctions (random sample, n = 1000), novel donors and acceptors in splice junctions upregulated in triple SAFB KD (DEXSeq, p < 0.05, LFC > 1; 295 acceptors and 142 donor), and control sets of random and best-scoring donor and acceptor dinucleotides in 500 nt windows around the novel sites. **k**, Empirical cumulative distribution function (ECDF) plot of the distance between splice site (acceptor or donor) to the nearest (upstream or downstream) L1 or anti-sense L1 transposon. Upregulated splice acceptor sites in SAFB depleted cells (orange) are closer to downstream L1 elements compared to control splice sites (green). **l**, The most significantly enriched sequence motif within upregulated exons in SAFB depleted cells, or in SAFB1, SAFB2, SLTM peaks obtained from data in humans as well as Safb1 and Saf-B FLASH data in mouse and fly cells. Data obtained using HOMER. **m**, Nuclear-to-cytoplasmic ratio of individual L1 elements (L1) and genes (gene) in control and SAFB-depleted HEK293 cells.

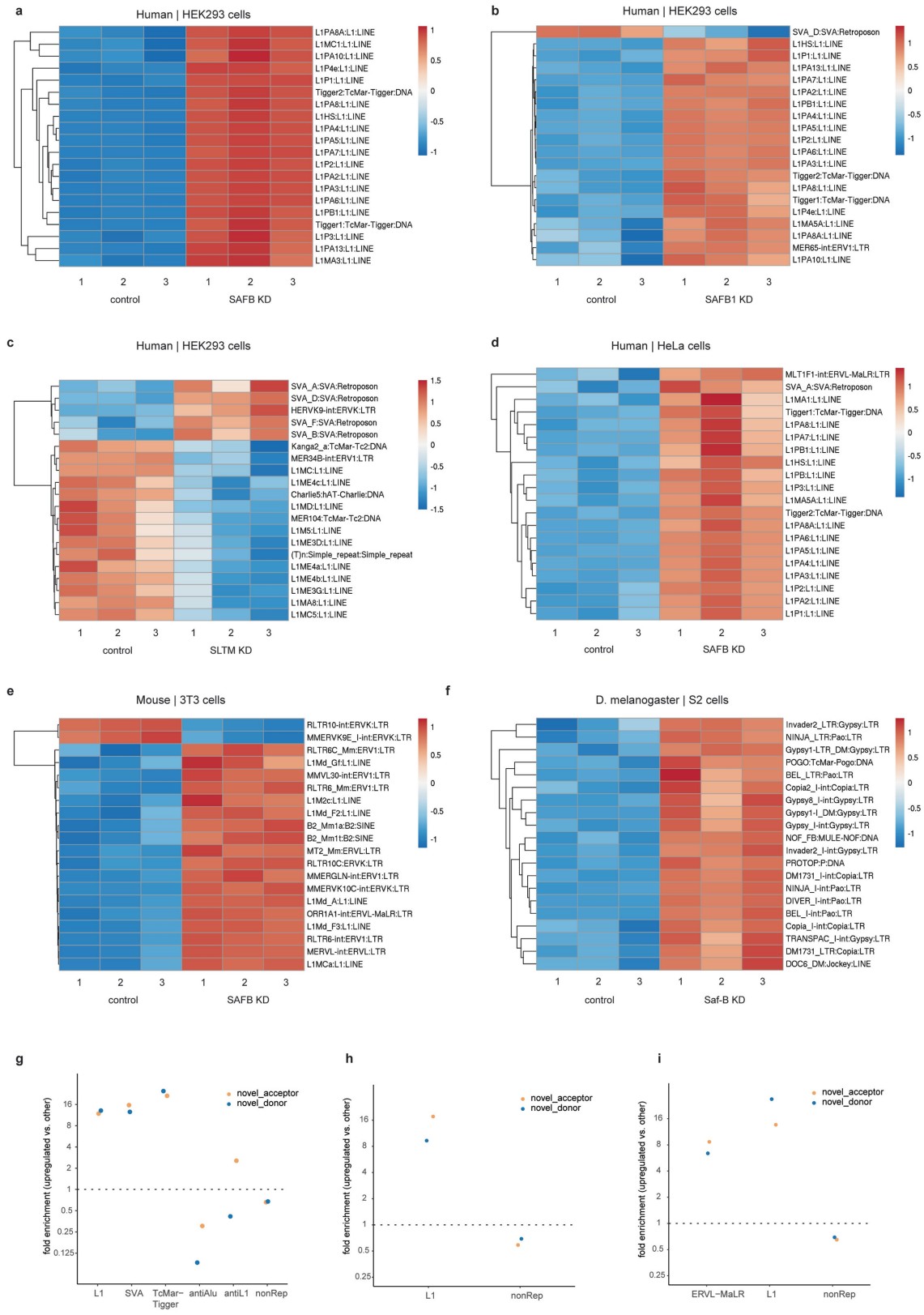

**Extended Data Fig. 6 | Differential Transposon expression in SAFB1, SAFB2, SLTM depleted cells.** Top 20 most significantly changing repetitive elements, as quantified by the snakePipes non-coding-RNA pipeline (no significant changes were seen in SAFB2-depleted cells). Complete DESeq2 output can be found in Supplementary Data Tables 4–14. Z-scores (calculated between replicates for each sample to show agreement between replicates) are shown

for: **a**, SAFB1 + SAFB2 + SLTM (SAFB) KD in HEK293 cells, **b**, SAFB1 KD in HEK293 cells, **c**, SLTM KD in HEK293 cells, **d**, SAFB1 + SAFB2 + SLTM (SAFB) KD in HeLa cells, **e**, Safb1 + Safb2 + Sltm (SAFB) KD in 3T3 cells, **f**, Saf-B KD in S2 cells. Fold enrichment of novel splice sites detected in: **g**, SAFB-depleted HEK293 cells, **h**, SAFB-depleted HeLa cells. **i**, SAFB-depleted 3T3 cells.

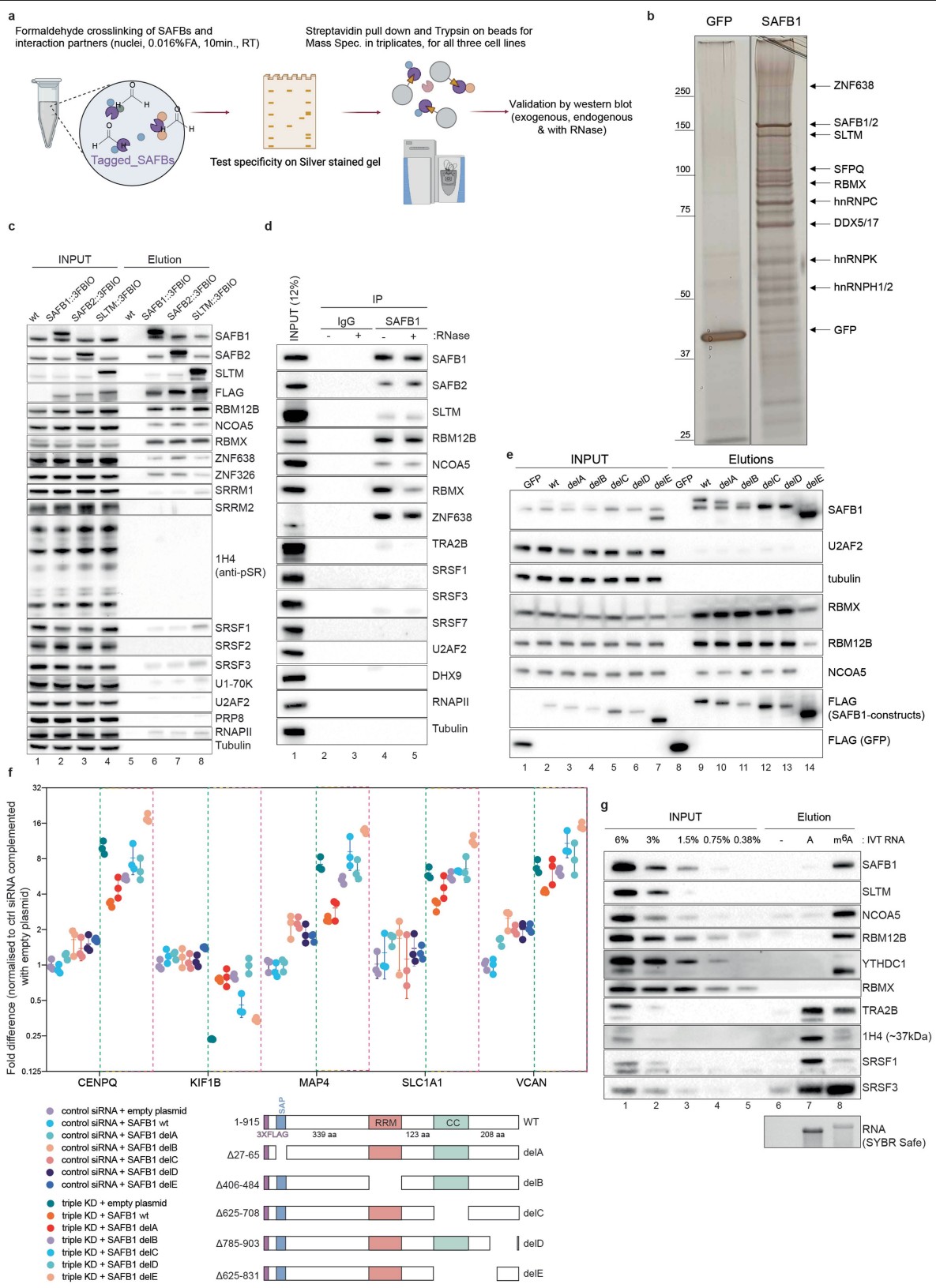

**Extended Data Fig. 7** | See next page for caption.

**Extended Data Fig. 7 | Biochemical characterization of SAFB interacting proteins. a**, Scheme of biochemical purifications involving tagged (b,c,e) or endogenous SAFB (d) proteins. **b**, Silver-stained polyacrylamide gel showing the specificity of the purification for 3xFLAG-Bio-SAFB1. Proteins indicated on the left were determined by mass-spectrometry. **c**, Verification of the candidate co-interectors determined AP-mass-spectrometry experiments with 3xFLAG-Bio-SAFB1, 3xFLAG-Bio-SAFB2 and 3xFLAG-Bio-SLTM with immunoblotting. These are the same cell lines used for FLASH. Also see Supplementary Table 3 for the results of the MS analysis. **d**, Verification of the candidate co-interectors determined AP-mass-spectrometry using an antibody against endogenous SAFB1, with or without RNAse treatment to determine whether the interactions are RNA-bridged. **e**, Interaction of SAFB co-interactors with SAFB1 truncation mutants. See panel f, of this Extended Data Figure for the description of the deletions. **f**, (left) RT-qPCR results interrogating targets as described in Extended Data Fig. 4e with the description of the deletions (right). **g**, Interaction specificity of in-vitro transcribed m6A RNA towards SAFB1 and its interaction partners; NCOA5, RBM12B is shown with respect to known m6A reader YTHDC1 or SR/SR-like proteins; TRA2B, SRSF1, SRSF3 by using a nuclear lysate.

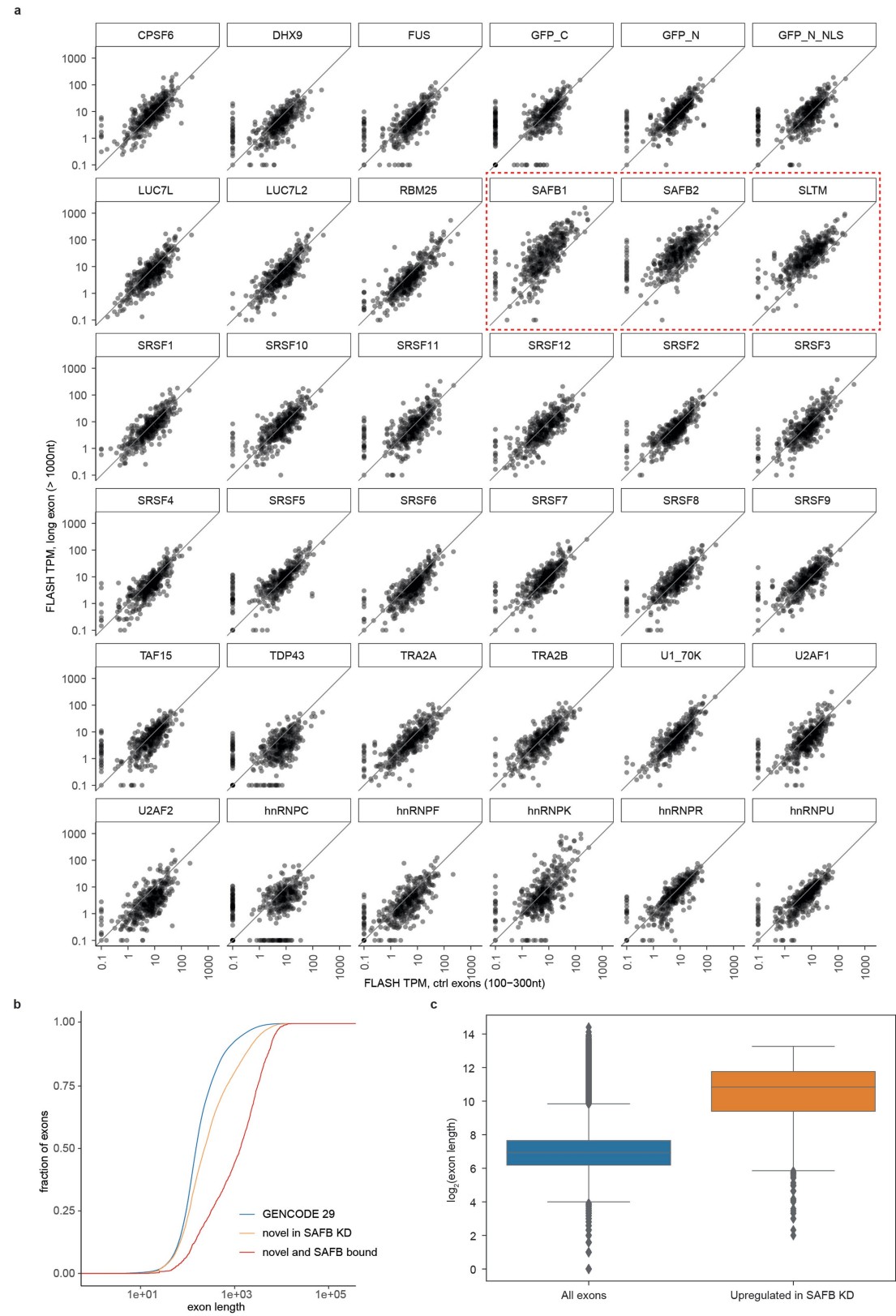

**Extended Data Fig. 8 | SAFB proteins bind to and suppress long exon splicing. a**, Scatterplots comparing FLASH coverage of all RBPs on average sized exons (100–300nt) on the x-axis, versus long exons (>1000nt) on the y-axis. SAFB1, SAFB2 and SLTM are highlighted. **b**, Empirical cumulative distribution function (ECDF) plot showing length of exons in GENCODE v.29, compared to novel exons detected in SAFB-depleted HEK293 cells. **c**, Boxplot showing length distribution of all exons, compared to exons that are upregulated in SAFB-depleted HEK293 cells, which includes novel and previously characterised exons.

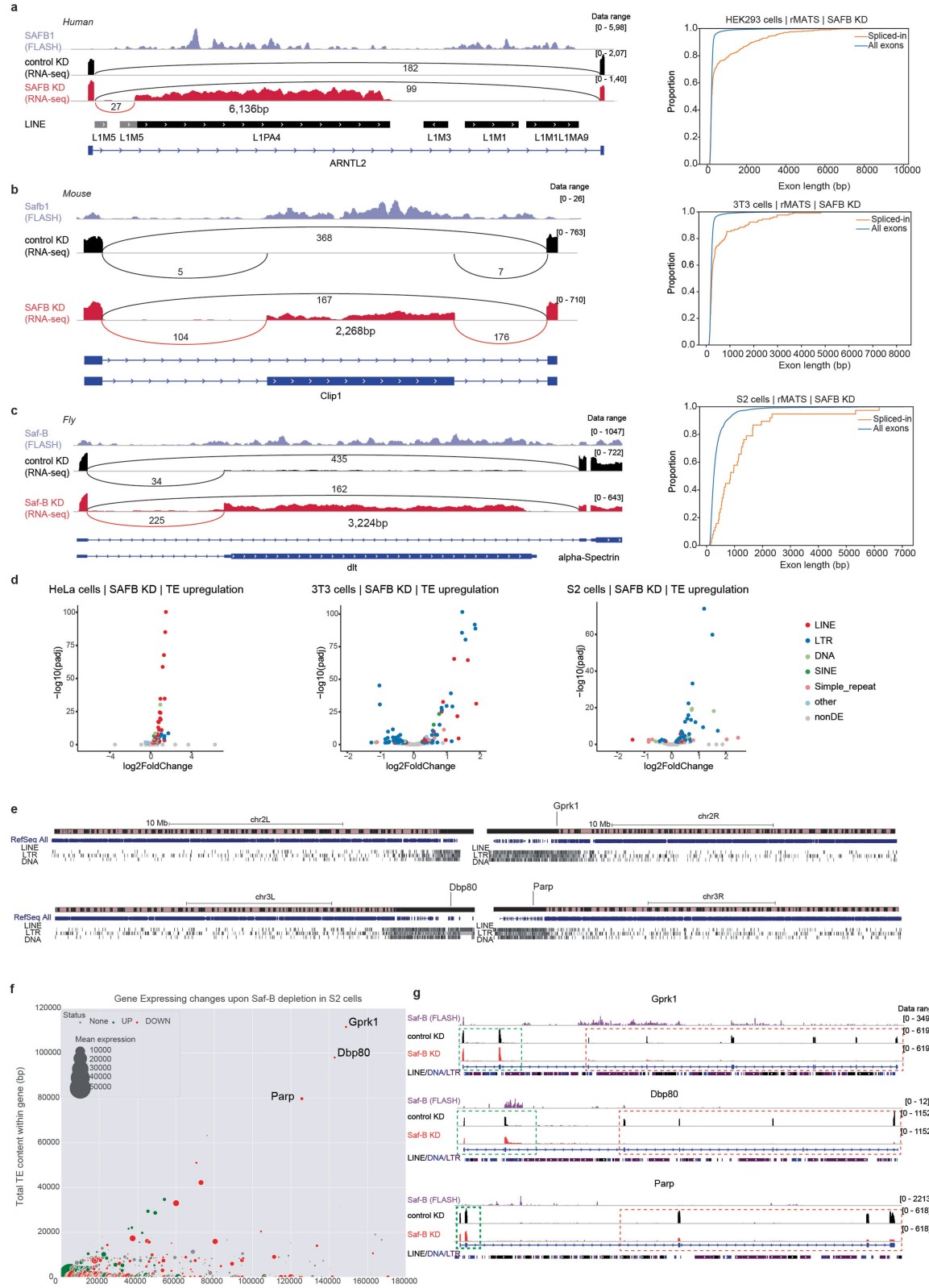

**Extended Data Fig. 9** | See next page for caption.

**Extended Data Fig. 9 | Genes at the pericentromeric heterochromatin with high TE content are vulnerable to Saf-B depletion in flies. a**, Left, IGV snapshot of the human gene ARNTL2, with SAFB1 binding and RNA-seq data in control vs SAFB-depleted cells. Lines connecting adjacent exons depict splice-junctions, provided together with the number of reads supporting a given junction. Right, ECDF plot of exon lengths, orange line: exons that spliced-in upon SAFB-depletion, blue line: all exons. **b**, Left, IGV snapshot of the mouse gene Clip1, with Safb1 binding and RNA-seq data in control vs SAFB-depleted cells. Lines connecting adjacent exons depict splice-junctions, provided together with the number of reads supporting a given junction. Right, ECDF plot of exon lengths, orange line: exons that spliced-in upon SAFB-depletion, blue line: all exons. **c**, Left, IGV snapshot of the fly same-strand nested pair dlt and alpha-Spectrin, with Saf-B binding and RNA-seq data in control vs Saf-B-depleted cells. Lines connecting adjacent exons depict splice-junctions, provided together with the number of reads supporting a given junction. Right, ECDF plot of exon lengths, orange line: exons that spliced-in upon SAFB-depletion, blue line: all exons. **d**, Differential expression of transposable elements in human (HeLa, also see Fig. 3a for HEK293 cells), mouse (3T3) and fly (S2) cells. **e**, Four chromosome arms, 2 L, 2 R, 3 L and 3 R, depicted with genes (blue boxes) as well as transposons (black boxes, separated by class), showing enrichment of the latter at pericentromeric heterochromatin where left and right arms of the chromosome are physically connected. Positions of the genes-of-interest, Gprk1, Parp and Dpb80 are highlighted. **f**, Scatter plot showing the size (x-axis) vs total transposon contents (y-axis) of all genes in D. melanogaster in base-pairs (the plot is restricted to 180.000 bp on the x-axis). The size of the dots indicates relative gene expression in S2 cells, while the color show if the genes are differentially expressed or not upon Saf-B depletion. **g**, IGV snapshot showing FLASH coverage of Saf-B, as well as RNA-seq coverage in control vs Saf-B dsRNA treated S2 cells. Green boxes highlight the initial 2 exons of each gene with little to no changes in expression, while red boxes highlight downstream exons which are significantly downregulated, reminiscent of phenotypes observed in mammalian cells (see Fig. 3).

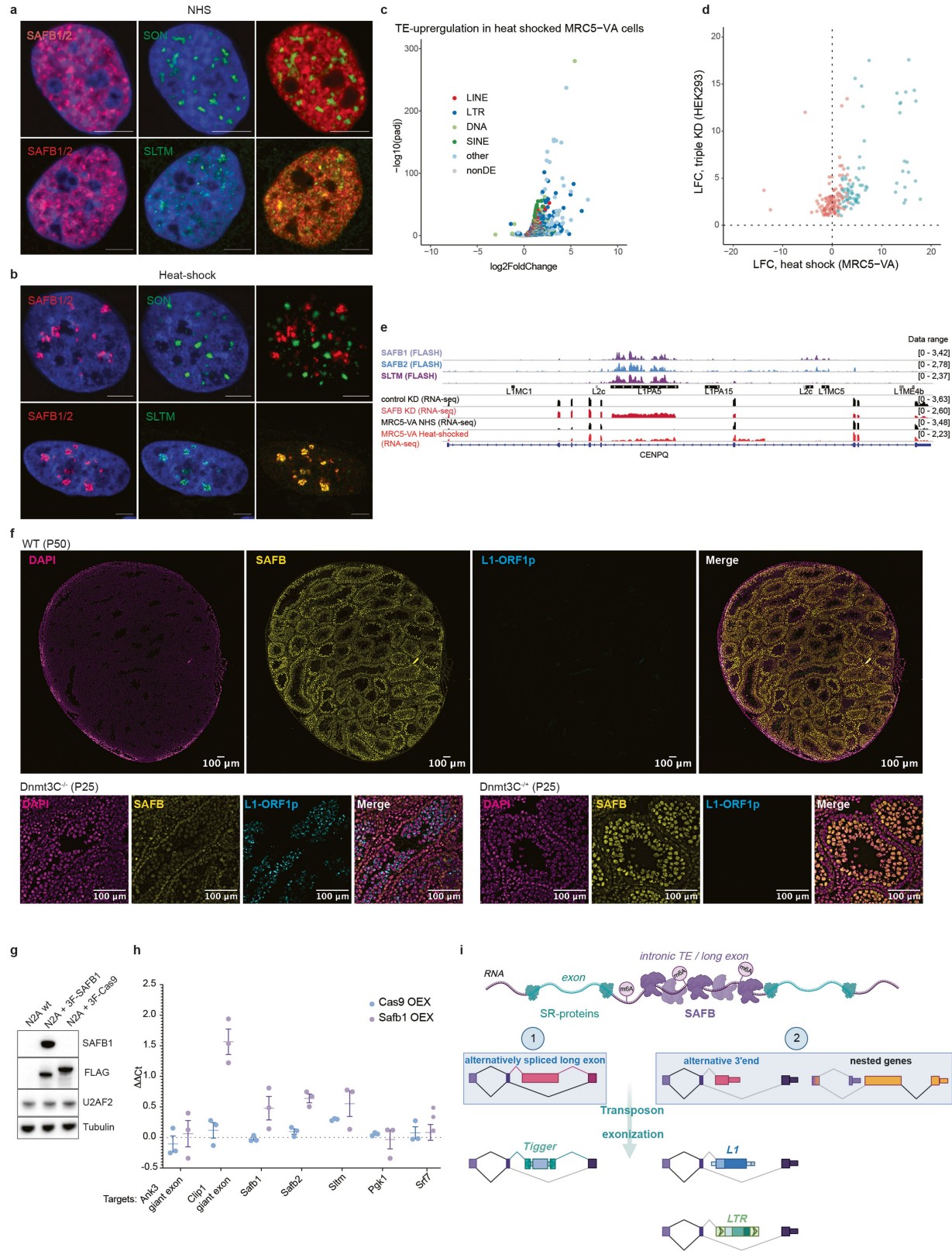

**Extended Data Fig. 10** | See next page for caption.

**Extended Data Fig. 10 | Heat-shock sequesters SAFB proteins at nuclear stress bodies (nSB) and SAFB expression correlates with L1-ORF1p in testis tissue or giant exon exclusion in N2A cells.** SAFB1/2 (HET), SON and SLTM stainings in controls (NHS) vs cells incubated at 42 °C for 90 min (Heat-shock) depicting co-localization of SAFB1/2 and SLTM before, and more clearly after heat-shock at nSBs. SON, a core component of nuclear speckles where splicing factors accumulate, does not overlap with SAFB1/2 under normal conditions (**a**), and forms nuclear bodies that are completely separate from nSBs after heat-shock (**b**). **a**, Stainings under normal conditions (HEK293 cells). **b**, Stainings after heat-shock conditions (HEK293 cells). **c**, Volcano plot showing changes in repetitive element expression in heat-shocked MRC5-VA cells. **d**, Scatter plot showing splice sites that are upregulated in HEK293 cells upon SAFB-depletion (log-fold change against control treatment on the y-axis), compared to changes in heat-shocked MRC5-VA cells (log-fold change against NHS on the y-axis). **e**, IGV snapshots showing FLASH coverage of SAFB1, SAFB2 and SLTM, as well as RNA-seq coverage in control or SAFB1 + SAFB2 + SLTM (SAFB) siRNA treated HEK293 cells, together with normal (NHS) and heat-shocked MRC5-VA cells on CENPQ gene (also see Fig. 3). **f**, (left) Cryo-section of a wild-type (P50) testis, co-stained with antibodies against Safb1 (yellow) and ORF1p (cyan) to reveal the differential expression of Safb1 in different stages of spermatogenesis. No specific signal for ORF1p is detected, Scale bar=500 μm. This figure is the same as in Fig. 5g, channels are separated for better visibility. (right) Cryo-section of the Dnmt3c-/- and Dnmt3c-/+ mice co-stained with antibodies against Safb1 (yellow) and ORF1p (cyan) showing intense staining of ORF1p towards the lumen where Safb1 expression is low. Scale bar=100 μm. **g**, Immunoblot showing expression of 3xFLAG-SAFB1 or 3xFLAG-Cas9 in mouse N2A cells. Wild-type N2A cells are used as a control. **h**, RT-qPCR experiment interrogating the effect of 3xFLAG-SAFB1 overexpression to the splicing of Ank3 and Clip1's giant cassette exons. Error bars depict the SD of three replicates. **i**, Model summarising the findings of this work. SAFB proteins bind to long, adenine-rich RNAs that are likely enriched with m6A modification (top). These characteristics are enriched in autonomous transposons such as L1 elements in humans and mice, but also in other diverse TEs such as Tigger DNA transposons and LTR elements. Similar molecular patterns apparently allow for regulation of giant cassette exons as well as nested genes, pseudogenes and retro-genes. In this model, we categorise the splicing changes upon SAFB depletion into two: (1) cassette exons, where either a coding exon, such as ANK3/Ank3, CLIP1/Clip1 or a TE fragment utilises both an splice-acceptor and and a splice-donor site for exonization or (2) where a splice-acceptor and a polyadenylation site is utilised to generate alternatively spliced 3'-ends, such as KIF1B, KIF16B, which is molecularly similar to nested-genes in Drosophila, as well as L1 and LTR elements that act as gene-traps and form chimeric transcripts with the host mRNA, causing early termination.

# Reporting Summary

## Statistics

For all statistical analyses, confirm that the following items are present in the figure legend, table legend, main text, or Methods section.

| n/a | Confirmed | |
|---|---|---|
| ☐ | ☒ | The exact sample size (*n*) for each experimental group/condition, given as a discrete number and unit of measurement |
| ☐ | ☒ | A statement on whether measurements were taken from distinct samples or whether the same sample was measured repeatedly |
| ☐ | ☒ | The statistical test(s) used AND whether they are one- or two-sided *Only common tests should be described solely by name; describe more complex techniques in the Methods section.* |
| ☒ | ☐ | A description of all covariates tested |
| ☐ | ☒ | A description of any assumptions or corrections, such as tests of normality and adjustment for multiple comparisons |
| ☐ | ☒ | A full description of the statistical parameters including central tendency (e.g. means) or other basic estimates (e.g. regression coefficient) AND variation (e.g. standard deviation) or associated estimates of uncertainty (e.g. confidence intervals) |
| ☐ | ☒ | For null hypothesis testing, the test statistic (e.g. *F*, *t*, *r*) with confidence intervals, effect sizes, degrees of freedom and *P* value noted *Give P values as exact values whenever suitable.* |
| ☒ | ☐ | For Bayesian analysis, information on the choice of priors and Markov chain Monte Carlo settings |
| ☒ | ☐ | For hierarchical and complex designs, identification of the appropriate level for tests and full reporting of outcomes |
| ☒ | ☐ | Estimates of effect sizes (e.g. Cohen's *d*, Pearson's *r*), indicating how they were calculated |

*Our web collection on statistics for biologists contains articles on many of the points above.*

## Software and code

Policy information about availability of computer code

| Data collection | Deep sequencing was performed with NovaSeq (Illumina). Long read direct RNA sequencing was performed with ONT. Luciferase activity was read on Omega Lumistar machine. Confocal microscopy images are acquired with Leica Stellaris 8. |
|---|---|
| Data analysis | High-throughput sequencing data were analyzed with: bowtie2 (v. 2.3.5), cutadapt 4.1, STAR 2.7.9a aligner, UCSC repeatMasker annotation, HOMER, rMATs, DESeq2, DEXSeq , Splice AI, recount3. The data analysis is described in the "Methods" section and code is available at: https://github.com/aktas-lab/safb_paper |

For manuscripts utilizing custom algorithms or software that are central to the research but not yet described in published literature, software must be made available to editors and reviewers. We strongly encourage code deposition in a community repository (e.g. GitHub). See the Nature Portfolio guidelines for submitting code & software for further information.

## Data

Policy information about availability of data

All manuscripts must include a data availability statement. This statement should provide the following information, where applicable:

- Accession codes, unique identifiers, or web links for publicly available datasets
- A description of any restrictions on data availability
- For clinical datasets or third party data, please ensure that the statement adheres to our policy

The FLASH and RNA-seq data were deposited in the Gene Expression Omnibus under accession code: GSE223263

## Human research participants

Policy information about studies involving human research participants and Sex and Gender in Research.

| Reporting on sex and gender | n/a |
|---|---|
| Population characteristics | n/a |
| Recruitment | n/a |
| Ethics oversight | n/a |

Note that full information on the approval of the study protocol must also be provided in the manuscript.

# Field-specific reporting

Please select the one below that is the best fit for your research. If you are not sure, read the appropriate sections before making your selection.

☒ Life sciences          ☐ Behavioural & social sciences          ☐ Ecological, evolutionary & environmental sciences

For a reference copy of the document with all sections, see nature.com/documents/nr-reporting-summary-flat.pdf

# Life sciences study design

All studies must disclose on these points even when the disclosure is negative.

| Sample size | No statistical methods were used to predetermine sample size. |
|---|---|
| Data exclusions | No data were excluded form the analysis. |
| Replication | Experiments were repeated as described in the methods section. All experiments shown could be reproduced as described. |
| Randomization | This study does not involve animal experiments (only tissues sections were taken from the male wild-type or Dnmt3C KO animals) |
| Blinding | The experiments presented in this article were not blinded (consistent with what is published in the field). |

# Reporting for specific materials, systems and methods

We require information from authors about some types of materials, experimental systems and methods used in many studies. Here, indicate whether each material, system or method listed is relevant to your study. If you are not sure if a list item applies to your research, read the appropriate section before selecting a response.

## Materials & experimental systems

| n/a | Involved in the study |
|---|---|
| ☐ | ☒ Antibodies |
| ☐ | ☒ Eukaryotic cell lines |
| ☒ | ☐ Palaeontology and archaeology |
| ☐ | ☒ Animals and other organisms |
| ☒ | ☐ Clinical data |
| ☒ | ☐ Dual use research of concern |

## Methods

| n/a | Involved in the study |
|---|---|
| ☒ | ☐ ChIP-seq |
| ☒ | ☐ Flow cytometry |
| ☒ | ☐ MRI-based neuroimaging |

## Antibodies

| Antibodies used | In imaging: SAFB1/2 het (Human; Millipore, 05-588 (clone 6F7), Mouse; LSBio LS-C2886411), SLTM (Invitrogen PA5-59154), SON (Sigma, HPA023535), Orf1p (Mouse; Abcam, ab216324)<br>In Western Blots: SAFB1 (Santa Cruz, sc-393403), SAFB2 (Santa Cruz, sc-514963), SAFB1/2 (HET) (Human; Merck/Sigma-Aldrich, sc05-588), SLTM (Invitrogen, PA5-59154), ORF1p (Human; Abcam, ab230966), TASOR (Sigma-Aldrich, HPA006735), 1H4 (p-SR) (Merck/Sigma-Aldrich, MABE50), RBM12B (Bethyl, A305-871A-T), RBMX (Cell Signalling Technology, 14794S), NCOA5 (Bethyl, A300-790A-T), ZNF638 (Sigma-Aldrich, ZRB1186), ZNF326 (Santa Cruz, sc-390606), TRA2B (Bethyl, A305-011A-M), U2AF2 (U2AF65) (Santa Cruz, sc-53942), TUBULIN (Santa Cruz, sc-32293), SRRM1 (Abcam, ab221061), SRRM2 (SC35) (Sigma-Aldrich, S4045), SON (Sigma-Aldrich, HPA023535), DHX9 (Abcam, ab 183731), U1-70K (SySy, 203011), PRP8 (Santa Cruz, sc-55533), RNAPII (Creative |
|---|---|

Biolabs, CBMAB-XB0938-YC), IgG normal mouse (Santa Cruz, sc-2025), SRSF1 (Santa Cruz, sc-33652), SRSF2 (Abcam, ab204916), SRSF3 (Elabscience, E-AB-32966), SRSF7 (MBL, RN079PW), RB1 (Cell Signalling Technology, 9309S), TRA2B (Santa Cruz, sc-166829) YTHDC1 (Proteintech, 14392-1-AP).

| Validation | anti-FLAG-M2 (Sigma) was used for the validation of mouse and fly cell lines that contain endogenously tagged SafB allele. Specificity for SAFB antibodies were validated by siRNA based knock-downs. |

# Eukaryotic cell lines

Policy information about cell lines and Sex and Gender in Research

| Cell line source(s) | Cell lines (human FlpIn Trex HEK293, human HeLa, human HCT116, mouse FlpIn 3T3, mouse N2A, fly S2R+) were all purchased from vendors, repositories or provided by colleagues (as described in the methods section). |

| Authentication | No further authentication of the cell lines was done after purchasing. |

| Mycoplasma contamination | Routine mycoplasma contamination tests were performed with Jena Biosciences Mycoplasma (PCR-based) detection kit (PP-401) |

| Commonly misidentified lines (See ICLAC register) | The list of cell lines used in this study are not amongst the misidentified cell lines. |

# Animals and other research organisms

Policy information about studies involving animals; ARRIVE guidelines recommended for reporting animal research, and Sex and Gender in Research

| Laboratory animals | Dnmt3C knockout animals were generated as described in (Wang et al., 2013).The founder mutation was subsequently backcrossed into the C57BL/6J background. Homozygous knockout males were validated as infertile, with significantly smaller and disordered testes by P42 as reported previously (Barau et al., 2016). |

| Wild animals | n/a |

| Reporting on sex | The experiments needed to be performed in a tissue where the SAFB expression levels are dynamically changing, the testis tissue was selected. Therefore, only male mice at p25 or p50 stages were used in this study. |

| Field-collected samples | n/a |

| Ethics oversight | The generation of these experimental animals is regulated following ethical review by the Yale University Institutional Animal Care and Use Committee (IACUC, Protocol #2020-20357) and was performed according to governmental and PHS requirements. |

Note that full information on the approval of the study protocol must also be provided in the manuscript.

