## [Peer Review File · Nature]

Manuscript Title: Autonomous transposons tune their sequences to ensure somatic suppression

Reviewer Comments & Author Rebuttals

Reviewer Reports on the Initial Version:

Referees' comments:

Referee #1:

Ilik, Glazar and co-workers present an exciting and generally well-executed study identifying SAFB family proteins as ancient repressors of transposable elements (TEs) and certain cryptic splice sites. I found the results to be very interesting and important, and worth publishing in Nature. However, I have a few criticisms and suggestions that the authors may want to consider. I detailed them here following their order in the manuscript:

1) In the Abstract, it is stated that "Splicing events suppressed by SAFB in somatic cells are activated in the testis, coinciding with low SAFB expression in post-meiotic spermatids". This claim is based on results presented in Fig. 4e,f and mentioned only in the Discussion. I think this statement is at least partially misleading. It is true that SAFB expression is low in spermatids, fitting the conclusion. However, even though testis-specific splice junctions are the largest group of tissue-specific junctions, they are only a minority in Fig. 4f when all junctions are considered. For instance, those in bone marrow are nearly as prevalent, based on the heatmap. Moreover, many of the giant exons they discussed are neuronal-specific. Therefore, things seem to be more complex than claimed by the authors, and such a crystal-clear conclusion in the Abstract seems misleading.

2) Fig. 3c: It is claimed that cryptic polyA sites from TEs are used upon SAFB KD. This is clear and convincing in the few examples presented (e.g. Fig. 3d), but not so strongly supported by the genome-wide analysis of Fig. 3c comparing up and downstream parts of affected genes. While the effect is observed and significant, this does not directly inform about the use of the cryptic polyA sites. This can be properly demonstrated generating 3' seq data for the same conditions and comparing the differential use of cryptic vs. canonical polyA sites.

3) If I understood correctly, it is suggested that activation of cryptic splice sites in the TEs upon SAFB KD leads to exonization of the TE, which is what then leads to the activation of the cryptic polyA site (lines 159-163). I have several doubts here. First, how "common" is the coupling between activation of cryptic splice sites and the use of cryptic polyA sites (line 158)? This is important information to know, since "common" may refer to 20% or 95%, and this will have different mechanistic implications. Second, it is also possible that recruitment of the transcription termination machinery enhances somehow spliceosomal recruitment (see also next point). A relatively simple way to assess this could be to knock-down the spliceosomal or the termination machinery, and see if the usage of cryptic polyA sites or cryptic splice sites is affected, respectively. In other words, the mechanistic link between the two events could be better elucidated. However, I acknowledge that this does not cast doubts on the validity of the overall observations.

4) Perhaps the least convincing mechanistic claim is that related to the role of SAFB in relation to the GAAGAA motifs. This motif is mentioned to be an ESE, and it is suggested that SAFB proteins mask it. I do not think this is properly demonstrated and, in fact, I would tend to think it is incorrect. From the work presented, it seems to me SAFB may repress spliceosomal assembly directly, not by avoiding the binding of other "promoting" proteins (SR factors). First, the authors say in lines 209-210 that "Our results show that SAFB proteins directly compete against SR-proteins by binding to GAAGAA". This claim was surprising, since (unless I missed it) this has not been shown anywhere. Perhaps the authors have evidence for it, but it is not presented. A nice and obvious way to show this would be to perform FLASH of SR proteins upon SAFB KD (they have done this for WT cells only and do not comment on the overlap of binding, as far as I can tell, only briefly for SRSF12). Second, the authors say that "If SAFB depletion turns TE sequences into splice enhancers, then we would expect to find novel splice sites not just inside SAFB-bound TEs, but also in the local vicinity of these elements". I am not sure this is the case. ESEs should act on the exonized sequence, not in the "intronic" vicinity. However, if SAFB motifs act as splicing silencers, with SAFB proteins directly repressing spliceosomal recruitment, then the vicinity effect may be expected. Also, stronger splice sites may be tolerated in the surroundings, precisely because SAFB represses them. Third, why does SAFB not outcompete SR proteins binding to GAAGAA motifs in regular exons, but only in TEs and seemingly long exons? And could SRSF3 be involved, as it is associated with long exons? Certainly, figuring out the mechanistic details on the splicing roles of SAFB is complex and likely beyond the scope of this study. But the authors should be careful with their claims and wording and discuss these issues more explicitly.

More minor points:

- Much of the Discussion is in fact results. Perhaps these should be placed in the results section, and the Discussion should probably elaborate more on the functional relationship with Piwi and the mechanistic insights with respect to splicing/polyA site selection/repression.
- Even though redundancy is expected based on homology, SLTM seems to be a bit of an outlier with respect to SAFB1 and SAFB2 in the different analyses. Moreover, in Fig. 2g it seems that the single SLTM KD *significantly* and substantially reduces retrotransposition efficiency. Therefore, I wonder, have the authors done double KDs? Is it really possible that SLTM is not playing the same role? I acknowledge, however, that this is not a central point for the study.
- The joint UMAP-based analysis of the FLASH data is spectacular. However, its description in lines 51-66 seems suboptimal, and not very easy to follow. I just the authors give this some more thinking.
- Line 42: Mention the cell type in which FLASH was performed.
- Ext. Data Figs. 4 and 5 are too large and not very clear and useful.
- Ext. Data Fig. 6: As presented, the similarity in the effects of the KD in HEK and HeLa is only anecdotal (a few examples and genes). I have no reasons to doubt it, but it is not properly shown (e.g., by plotting the fold change in both conditions of affected genes).

Referee #2:

Ilik et al. present data exploring the relationship between SAFB proteins, transposable element (TE) activity, splicing, and gene expression. Strong claims are made around SAFB proteins repressing L1 retrotransposons, acting as an RNA-based defence system against TEs (like the HUSH complex) and the resultant "tuning" of TE sequences in somatic cells. The evidence presented leaves open too many alternative explanations. It seems likely the SAFB triple KD produces a generalised increase of nuclear export, which would be less exciting given the known roles of the complex, and go against a TE-specific mechanism. I realise the authors will likely find this a disappointingly unenthusiastic viewpoint, but I have explained my reasons in detail below and signed my review.

1) Lack of evidence for specific L1HS repression by SAFB.

Fig. 2g shows the results of an L1 retrotransposition assay conducted using a luciferase reporter system from Wenfeng An's lab. It's not said, but here the positive control is a pCAG-driven L1RP element, which is extremely mobile in vitro, and boosting L1RP transcript abundance beyond this pCAG-driven level is known to negligibly impact retrotransposition rate. At a certain point, making more L1 mRNA has no impact on jumping. If SAFB is acting as an L1HS transcriptional repressor, it is therefore very unexpected that SAFB (triple) KD would boost L1RP mobility a further 3-fold. Technical points: data for the L1 RT- control were produced but not shown. The replicates are said to be biological (which is normally considered experiments conducted on different days), but appear to instead be technical (multiple wells on the same day) and so the SD would likely be substantially larger if replicated twice more.

The L1 probe in Fig. 2h here is designed against L1HS ORF2, which is highly conserved amongst L1s and therefore this probe is very unlikely to be specific to L1HS, which may explain why the signal is so strong. It would be prudent to test these probes in mouse cells, and in mouse cells expressing the L1RP plasmid, as controls.

Fig. 2d: How was this plot generated? The FLASH reads would not align to L1HS loci uniquely and so this must be generated by aligning them directly to an L1HS sequence? It is conspicuous that no L1HS loci are presented in IGV-style figure panels; they're all older L1s, and so presumably FLASH reads rarely appear on L1HS sequences displayed in this way. In this figure, the use of sense/antisense as a comparison is then hard to follow, and one would imagine most TEs have stronger FLASH signal on the sense strand.

Fig. 2: The siRNA experiments lack detail. Was one siRNA designed per protein? Immunoblots are not shown to indicate efficiency of protein KD. It is surprising that the triple KD gives such an effect on retrotransposition in Fig. 2g when KD against each of the SAFB proteins singly gives opposing results.

Fig. 3a and Ext. Data Fig. 7 show that the impact of SAFB KD on endogenous L1HS elements, as measured by a non-standard TE analysis of RNA-seq, is not significant. More to the point, the use of Z-score for colouring in Ext. Data Fig. 7 is likely to mislead as to the strength of these effects. What justification is there for strongly colouring a Z-score of 1? The claim that SAFB KD upregulates L1HS is not based on robust data. It is also interesting that YY2 is upregulated in this KD as YY2 perturbation

may contribute to the mild change in L1 expression observed as YY2 (and YY1) can both bind the L1HS promoter.

Bottom line: There is a lack of support that SAFB is a repressor of L1HS retrotransposition, or L1 transcription in general, and that is a major problem for the main claims around host defence in somatic cells.

2) SAFB is a multifunctional protein beyond splicing.

The data in Figs. 2h and 3d give very good reason to entertain an alternative, perhaps more mundane, explanation for the TE part of this manuscript, namely that SAFB impacts nuclear export and that this phenomenon is not specific to TEs. The examples of SAFB KD in mouse and fly (Fig. 4b,c) support this view, as they don't describe TEs, and for some of the examples (e.g. Ext. Data Fig. 6) there is appreciable signal on the L1 sequences highlighted in the control RNAi experiments.

Crucially, the polyA+ RNA-seq was prepared from total RNA, which for various reasons tends to be dominated by the cytoplasmic fraction. Therefore, if SAFB KD increases the export of previously nuclear-enriched transcripts, such as those containing intronic L1s, they would appear much more abundant in the RNA-seq data. However, this would not be the result of altered splicing, it actually would be the result of altered nuclear export. That would also be consistent with the retrotransposition assay data from Fig. 2g, where mobility increases much more than one would expect, because SAFB KD enhances export of the L1 mRNA, which is more rate-limiting than transcription.

3) Much reliance is placed on the use of FLASH to assess RNA-binding by SAFB proteins. It would be useful to provide data from an orthogonal method showing SAFB binding to L1 sequences, to exclude any technical artefacts relating perhaps to the A-rich SAFB motif and their preponderance in L1 sequences.

4) It's a more minor point, and yet it is striking and surprising that no mention is made of HUSH. HUSH is a very robust and specific repressor of L1 transcription and retrotransposition (PMID: 34794168), in a way that arguably SAFB does not appear to be.

5) Another minor point: Many of the IGV examples of splicing into ancient L1s are spectacular. Even if I disagree as to their significance, with regards to tuning the expression of host genes via altered splicing, this does show the potential wealth of nuclear RNAs that we tend to miss with RNA-seq made from total RNA preps.

Geoff Faulkner (University of Queensland)

Referee #3:

This manuscript by Ilik et al. describes how autonomous transposable elements (TEs) use their sequences to be recognized and masked by SAFB proteins and thus avoid their unnecessary splicing and exonization in target genes within somatic cells. They suggest the mechanism is in turn active in non-

somatic cells such that TEs can rearrange the genome effectively and propagate their own distribution. The FLASH- and RNA-seq-based data are presented with a bunch of derivable information in them. Almost 30 RBPs had been tested in FLASH to identify clustering and reveal SAFBs as specific binders to TEs, likely counteracting (alternative) splicing enhancing by SRs. The authors also identify that especially giant exons are targeted by TEs (which has in a way been suggested before, see below). As a novel finding they prove that SAFB depletion affects the splicing of TEs, in particular L1, also taking into account the TE's strand location.

Altogether, this study provides a convincing mechanistic insight into how TEs are kept suppressed by SAFBs. Although it remains open what makes SAFBs different from previously described splicing silencers, it well showcases the possible co-evolution of TEs and a suitable set of host RBPs.

The abstract is written clearly and is accessible to readers beyond the ultimate expert level. The same can be said about the introduction in principle. As authors try to put a focus on the role of SAFBs as "immune system component" they might like to introduce SAFBs a bit more in light of their unclear dual nucleic binding capabilities and role on the transcriptional level.

The data, as provided, are sound and trustworthy with sufficient control experiments carried out in most of the cases. I do almost unlimitedly support the conclusions from the presented data, and I would definitely underline the strong value of mechanistic assumptions on the SAFB-TE interactions regarding the comparison between somatic cells and e.g. reproductive tissues with respect to the role of SAFBs as immune system-like regulators.

The identified target motifs of SAFBs are supported by previous studies based at least on CLIP data, while the general question remains as how exactly SAFBs interact with their RNA target sequences.

As far as I can judge it (see my comments where I may not do so sufficiently) the treatment of statistics seems appropriate. The references are very focused on the TE side, which I may not oversee in total. For SAFBs authors have not put much focus on its manifold involvements in RNA processing, also connecting to diseases and underlined by a growing number of strong literature, but this may be related to a somewhat different scope here. Still it can teach us a lot about RNA-binding by SAFB, which is little understood on the structural level.

What appears a bit puzzling to me is that the manuscript misses to link to a previously published study (Xiong et al., Cell Res., 2021) in which the concept of TE suppression by SAFB proteins has been introduced, albeit in a m6A-related context. While the two studies would not need to overlap and in fact do not mechanistically (seeing the final model provided by Xiong et al.: the role of m6A, the place and timepoint of physical engagement with RNA, no mentioning of exonization etc.; in other words: a somewhat different focus), it shows that SAFBs have been identified as regulators of TEs in maintaining host gene expression before. Also, the SAFB-related target motifs are well in line and e.g. give good value to the still non-understood RNA target preferences for SAFBs (esp. supported by structural data). Still, I assume the herein presented study by Ilik et al. definitely adds a very strong value, a second, expanded perspective on SAFB-TE and its impact on gene regulation and will impact its research field, and on the longer run potentially beyond (seeing SAFBs as biomarkers, etc).

I, however, personally suggest to place the manuscript in its current form in a slightly more specific journal with a cell-biological and/or genetic focus seeing the above-mentioned.

Independently, I see a clear need of addressing the following issues:

- Connecting to the above mentioned: have the authors (meanwhile) looked at the role of methylation for SAFB localizing to TEs and the influence on TE activity in their setup? SAFBs are hardly canonical readers of m6A, but methylation may steer interactions indirectly. Please comment or add findings if available.

- The domain architecture of SAFB proteins is very similar. How are differences in effects, especially in k.d., between SAFBs explained? Is this based on expression levels or non-identical RNA affinities?

- Speaking of that: seeing the apparently clear but short target RNA consensus, how would SAFBs interact with/mask binding sites? Just through their RRM domains, which would require a pretty strong Kd, also to compete out other RBPs with similar preferences e.g. hnRNPG and Tra2-b1 (Moursy et al., NAR, 2014)?

- How does masking work? The suggested motifs from FLASH do not fit e.g. the ENCODE-deposited RBnS data (SAFB2) and at least may not be explainable with RRM(only)-mediated binding. The authors should add data that give more insight into the roles of RNP complex formation from the protein side, e.g. using truncations, depletions, etc. The role of homo- and heterooligomeric SAFB for RNA processing e.g. has been discussed recently and may be relevant here. Else, SAFBs could also be recruited to their final binding sites by another RBP.

- As critical also in other dual nucleic acid-binding proteins: the authors should provide a control experiment that undoubtedly excludes a potential influence on the transcriptional level, i.e. DNA-binding. For example, one could test effects of a SAFB with a knocked-out SAP domain. I do not doubt the post-transcriptional data presented here, but the proteins have originally been identified as chromatin regulators. The interplay between DNA- and RNA-binding is not clear and should not be ignored, e.g. in the statement of nuclear retention of TEs by SAFB.

- Please comment (or even show) if in general mRNA export is affected in SAFB k.o. systems!

- Related to the suggested concept of TE somatic suppression vs. activity in testes: can the authors comment on the apparent activity in bone marrow (Fig. 4f)?

- Finally, I do not fully understand the possibility of SAFBs to selectively regulate giant protein-coding cassette exons. How exactly could they themselves recognize the size of exons and what differentiates them from canonical SR-antagonists like hnRNPs? Please comment or add additional data as this would valuably contribute to answering a central question of selective splicing suppression organized by RBPs. Do SAFBs interact with the splicing machinery in some way?

- I strongly suggest to include a final model figure/panel that summarizes the findings.

Regarding methods, their description seems comprehensive, but some essentials are missing.

- As a non-expert in this, I miss data on the control over SAFB expression levels: are the transfected mammalian cells expected to express protein on the endogenous-like level? Is there SAFB endogenous background (which is to get rid of in the siRNA treatments)? Please provide experimental data, i.e. WB!

- For Flash: I do not fully understand: are all RBPs equipped with Biotin acceptor peptide affinity tag to parallelize the procedure? This is not explained, or at least is not sufficiently clear to me.

- It may make sense to comment on possible relayed effects of SAFB-k.d.-related changes in gene expression, i.e. indirectly through the lack of affected proteins to regulate others' expression, stabilities, etc (also on the transcript level).

Finally, some minor things:

- In many panels of the current figure versions (esp. the supplement) the axis labelling is very small and hard to read, which is complicated for the non-expert reader less familiar with the type of data presentation.

- Statements of correlation in lines 130-132 are to my understanding not shown as a figure panel.

- Line 129: ...compared SAFB-bound regions (remove "of").

- Line 142: new terminal exon? (not "now").

- Domain color coding in Ext. Data Fig. 1a is not very helpful using very similar types of blue and violet. As a side note, SR proteins usually contain a pseudo RRM as the second one if in tandem.

Referee #4:

Transposable elements (TEs) form a major component of the human genome that, when active, can be expressed and inserted within protein-coding genes, thereby in principle affecting host gene expression. However, it is well known that most intronic transposable elements (TEs) are never exonized and are simply spliced out, suggesting that cellular factors can a) discriminate TE- from "host"-derived splice sites, and b) suppress activity of the former. However, this is not well understood on the mechanistic level.

In their article "Autonomous transposons tune their sequences to ensure somatic suppression", Ilik et al. follow up on this observation and set out to identify novel factors that regulate splicing of intronic TEs in the human genome. Exploiting their own CLIP-like FLASH protocol, they generated and thoroughly analyzed RNA-protein interaction datasets for in total 33 splicing regulators, such as SR/SR-like and HnRNP proteins. This led to the identification of the SAFB family, comprising SAFB1, SAFB2 and SLTM, as specifically enriched at sense L1 and Tigger DNA transposon transcripts. Intriguingly, knockdown of all SAFB members enhanced L1 retrotransposition in a luciferase-based reporter assay,

suggesting that this binding is functionally relevant. To investigate the role of SAFB proteins on TE (and general gene) expression on a global level, the authors compared RNA-seq profiles in two SAFB-proficient and -knockdown cell lines and observed upregulation of L1 and Tigger elements upon SAFB loss. Mechanistically, this upregulation was mainly due to the activation of cryptic splice sites embedded within or in close proximity to the 5' TE region, which resulted in the exonization of intronic TEs and activated cryptic polyadenylation sites at the 3' end of the TE. Consequently, this led to reduced expression/coverage of coding exons downstream of this novel splice site. Interestingly, this exonization phenomenon was not restricted to TEs, but was also found for sense pseudogenes and giant protein-coding cassette exons, all of which turned out to be relatively long compared to conventional exons and were enriched for adenosine-based, purine-rich GAAGAA sequences, the prototypic SAFB binding motif. Moreover, this mechanism appeared to be conserved in mice and even in flies, possibly indicating that suppression of TEs and other purine-rich motifs constitutes the evolutionary original function of SAFB proteins.

Hypothesizing about the cellular and even organismal role of this SAFB-mediated suppression, the authors in the end speculate that TE activation may be some type of stress response to allow for better adaptations under strenuous conditions. In fact, 90 min of heat-shock – which sequesters SAFB proteins into stress bodies and thus potentially depletes them for other functions – activates alternative splicing events and TE upregulation that largely recapitulate the SAFB knockdown phenotype. In addition, the authors provide evidence that SAFB expression seems to be rather low in post-meiotic spermatids, and this likewise leads to upregulation of similar splicing events as upon SAFB KD. From an evolutionary point of view, this SAFB low condition may allow TE activity in a short window that allows inheritance of all changes to the next generation, but at the same time does not affect the host.

Together, this description of SAFB proteins as evolutionary conserved suppressors of TEs and other TE-like elements such as pseudogenes and giant protein-coding exons is very interesting and overall well supported by the presented data. However, I would like to ask for a few additional experiments to further support this novel role of SAFB proteins, and to also provide more mechanistic insight into how SAFB proteins perform this task.

1. What is the individual role of the different SAFB protein domains the observed effects? Is TE binding and suppression (both of exonization and retrotransposition) mediated only by the RNA binding domain, or are other functionally relevant proteins recruited into the complex via additional domains? Can the SAFB KD effect be rescued by reconstitution of the RNA domain alone?
2. How is expression of giant protein-coding exons regulated in normal tissues? Are the respective cells, in case of ANK2 and -3, MAP2 and MAPT for example of the nervous system, specifically depleted of SAFB proteins? Can giant protein-coding exon expression therein be suppressed by exogenous SAFB protein?
3. Is there any evidence for high activity of TE retrotransposition in post-meiotic spermatids? If so, can this be counteracted by SAFB proteins?
4. For the majority of the presented data, the authors show mapped IGV snapshots for SAFB binding and RNA-seq coverage to demonstrate changes in downstream exon expression and splicing. I think it

would be beneficial to confirm these findings by qPCR assays, also in order to get a better quantitative idea of the implied alterations.

5. The authors provide RNA-seq data of SAFB knockdown cells (HEK293 and HeLa) to demonstrate their TE-suppressing function in normal somatic tissues. Not surprising given the described role of SAFB proteins in diverse RNA-centric processes, SAFB KD results in prominent changes in gene expression, in particular in the triple KD. Can the authors comment on how many genes become up- and downregulated in the respective experiments (Ext. Data Fig. 6a-g), and what percentage of these genes contain intronic SAFB target elements such as TEs, pseudogenes or giant exons? Vice versa, how many genes that are expressed beyond a certain threshold in 293 and HeLa cells do not become deregulated despite being enriched for SAFB binding and TEs? Can the authors also please provide data for the extent of the knockdown, e.g. a western blot?

Response to Referees

We thank all the reviewers for their insightful comments and constructive criticism, which lead to a significantly improved manuscript. We tried to address all the major and minor points of the reviewers, wherever possible with additional experiments, new analyses, and/or modifications to the text and figures. We used blue text to indicate our responses, as well as the changes we made into the main text.

A key claim in the manuscript was that SAFB proteins prevent exonization of RNA segments they bind to, which we proposed is a result of direct competition with SR proteins. We took the criticism about this point very seriously, and acknowledge that we probably did not provide enough evidence to make such a strong statement in the original submission. Furthermore, we realised that we should have used more precise language describing what we mean by this competition, as one reviewer rightly remarked how difficult it would be to remove, for example, TRA2B bound to its target at equilibrium. What we meant was a more kinetic competition on nascent RNA, which does not necessarily involve stripping all SR-proteins, but enough to prevent the "right" RNP formation within the time-frame of splicing, which can prevent exon recognition and prevent splicing. Taking reviewers' comments into account, and in the light of the new evidence we have now re-written this part, which hopefully settles the issue.

Specifically, we addressed this point by three biochemical approaches. First, using the same cell-lines that we used for FLASH, we purified SAFB1, SAFB2 and SLTM and identified their interaction partners by mass-spectrometry, which were then verified by immunoblotting. These results show that SAFB proteins do not seem to interact with SR proteins, but with a set of hnRNP or hnRNP-like proteins, which is consistent with immunofluorescence stainings of SAFB1/2 and nuclear speckles (with marker protein SON) which appear mutually exclusive in interphase cells, and partition into separate phase-separated bodies upon heat-shock.

Encouraged by the total lack of phospho-SR co-purification in these interaction assays, we used the same antibody for FLASH to assay p-SR proteins in control vs triple-KD cells. We expected that if SAFB proteins actually compete with SR-proteins, then we should see increased binding of SR proteins on regions that were bound by SAFB proteins. We observed that among SAFB bound regions that showed differential binding in p-SR after SAFB depletion (N=807), 80 % were indeed upregulated. We also analysed the same data agnostic to SAFB binding by segmenting introns by their repeat annotation and observed a very specific increase at sense-L1 elements in pSR binding upon SAFB depletion, suggesting that there is a competition between SAFB proteins and SR proteins on sense-L1 RNA.

In the third approach, we expressed the RNA-binding domain of SAFB1 (RRM), the RNA-binding domain of TRA2B (RRM) in bacteria and in vitro transcribed (AGA)₇ RNA which was composed either of N6-methylated adenosines or unmethylated adenosines, and carried out gel-shift assays to see if m6A has any affect on either the binding of SAFB1's or TRA2B's RRM to their targets. This experiment surprisingly showed that TRA2B cannot interact with the N6-methylated probe at all, whereas SAFB1 is virtually insensitive to N6-methylation.

These results suggest that in vivo, the competition between SR-proteins and SAFB can also be steered by RNA methylation. Initially, TRA2B (SR) would have an advantage to interact with nascent RNA which is unmethylated, promoting splicing, this advantage would shift to SAFB1 as the RNA is gradually methylated, closing the window to splicing. Interestingly, this shift would hardly affect fast-spliced short exons, but would be critical for the splicing of long-exons if an upstream splice donor site is not quickly spliced onto the splice acceptor of the long exon. This experiment therefore provided an unexpected explanation to the observed sensitivity of giant exon splicing to SAFB expression, as well as an interesting biological link between transposable element RNA (TE-RNA), which is strongly methylated.

Based on the molecular phenotypes we observed upon SAFB depletion, we proposed that evolutionarily distant or even unrelated families of autonomous transposons but especially L1 RNA has evolved towards an A-rich RNA to remain silent in the soma, but that there's a possible window of opportunity for productive TE activity in post-meiotic spermatids where SAFB expression is low, providing a status quo beneficial to both the host and TEs. If this model is right, then if the A-bias of the L1 RNA could be removed, then this L1 variant should be free of SAFB regulation. Such an L1 variant was synthesised by the Boeke lab, named ORFeus, a hyperactive transposon, which we used in control vs SAFB-depleted cells. In our model, we propose SAFB to be an RNA-based suppressor of L1 activity, therefore to see the effect of SAFB depletion on L1 expression more clearly, we decided to also deplete the HUSH complex, which is a general epigenetic repressor of intronless constructs that functions through directing deposition of the repressive chromatin mark H3K9me3 on its targets. These results show that, in our experimental system, depletion of SAFB proteins or HUSH complex lead to a comparable increase in L1 expression. Combined depletion of SAFB proteins and HUSH complex, consistent with the idea that the former is a post-transcriptional regulator and the latter a transcriptional regulator, leads to a much more drastic increase in L1 expression, which we also confirmed by northern blotting.

Importantly, ORFeus, which no longer has an A-bias but otherwise identical to L1, is still repressed by the HUSH complex, but its expression is completely independent of SAFB proteins.

We also carried out an immunofluorescence staining of SAFB1 in a 7-week old mouse testis which showed consistent results with published scRNA-seq data, showing peak SAFB1 expression in spermatocytes and spermatogonia, which is then reduced in post-meiotic round spermatids, and completely disappear in elongating spermatids, which temporally coincides with the splicing of CLIP1's giant exon. We could also show that overexpression of SAFB1 in mouse N2A cells could reduce splicing of CLIP1's giant exon. In order to be able to see if SAFB1 positive cells are less prone to express L1 elements, similar to the HUSH depletion in cell culture, we used a DNMT3C KO mouse model which leads to transcriptional upregulation of young L1 elements, which are retrotranspositionally active, and co-stained SAFB1 and ORF1p. The staining pattern of ORF1p and SAFB1 in these samples appear almost mutually exclusive with each other, which suggest that when SAFB is present, it provides a "second line of defence" against L1 RNAs that escaped epigenetic suppression.

We also provide examples and analysis of ONT directRNA-seq of triple-KD HEK293 cells, which show that the chimeric transcripts formed by splicing of a host exon onto an L1 or Tigger element are indeed one single molecule. To address a reviewer comment about an alternative explanation of the chimeric transcripts without an increase in splicing, namely that

the chimeric transcripts we report are produced at the same rate as in normal cells, but are unstable due their nuclear retention and therefore mostly undetectable, but get stabilised upon SAFB depletion due to a general reduction in nuclear retention, we separately sequenced nuclear and cytoplasmic RNA from control vs. triple-KD cells and analysed the distribution of L1 signal from individual L1 loci between the fractions. We stress out that the L1 signal described here, although scored on individual L1 elements, is largely coming from chimeric transcripts and does not describe standalone L1 expression. In summary, the results are inconsistent with the alternative model, which would predict a strong reduction in L1 RNA in the nucleus, and an increased presence of L1 elements in cytoplasmic RNA. While we do score an increase in some cytoplasmic L1 chimeric transcripts, which is to be expected if they are de novo generated via splicing, we also detect up-regulation of chimeric transcripts in the nucleus, which cannot be explained by an "export only" model, but is readily compatible with enhanced splicing, increasing their concentration first in the nucleus. We also added visual guides to the figure that explains how L1/Tigger elements act as "gene traps": when we detect an increase in a SAFB-bound TE in SAFB depleted cells, the exons of the host gene upstream of this event tend not to change, but exons downstream of this event are downregulated. This effect contradicts the "export only" model since it cannot explain why exon expression downstream of the upregulated TE would be coupled to the export and/or stability of the upstream chimeric transcript. This phenotype is the expected result when the TE is spliced into the gene, forming a chimeric transcript, and causing early termination due to the presence of a pA site, which most sense-L1 are expected to have. In other words, if more of the transcriptional activity is diverted to the chimeric transcript, then fewer pA sites map at the end of the host gene, which cannot be explained with increased export of chimeric transcripts.

We additionally provide immunoblots that not only show siRNA efficiencies but also a curious feedback-loop between SAFB1 and SAFB2 in HEK293 and Hela cells, where all three SAFB proteins are expressed, and between SAFB1 and SLTM in HCT116 cells where SAFB2 expression is undetectable. Additional experiments in this revision also include RT-qPCR analysis of splicing defects observed in triple KD cells, together with all single- and double-depletions, which show that SAFB1 is the most important of the three SAFB proteins, at least in HEK293 cells, and while depletion of either SAFB2 or SLTM in addition to SAFB1 increases the severity of splicing defect, triple KD always causes the most severe phenotype, providing strong evidence for redundancy between SAFB proteins. Other experiments provided in the revision are RIP-qPCR using SAFB1 as bait verifying interactions with L1, Tigger, and giant exons, rescue of splicing defects in triple-KD by ectopic expression of siRNA-resistant SAFB1 and a domain-deletion analysis using the same RT-qPCR assay, as well as monitoring of co-interactors by mass-spectrometry and finally polyA+ RNA-seq data generated from HCT116 cells, where we depleted SAFB1 and SLTM since SAFB2 is not expressed, which show identical phenotypes observed in triple KD HEK293 and HeLa cells.

Point-by-Point Response

Referee #1:

Ilik, Glazar and co-workers present an exciting and generally well-executed study identifying SAFB family proteins as ancient repressors of transposable elements (TEs) and certain cryptic splice sites. I found the results to be very interesting and important, and worth publishing in Nature. However, I have a few criticisms and suggestions that the authors may want to consider. I detailed them here following their order in the manuscript:

1) In the Abstract, it is stated that "Splicing events suppressed by SAFB in somatic cells are activated in the testis, coinciding with low SAFB expression in post-meiotic spermatids". This claim is based on results presented in Fig. 4e,f and mentioned only in the Discussion. I think this statement is at least partially misleading. It is true that SAFB expression is low in spermatids, fitting the conclusion. However, even though testis-specific splice junctions are the largest group of tissue-specific junctions, they are only a minority in Fig. 4f when all junctions are considered. For instance, those in bone marrow are nearly as prevalent, based on the heatmap. Moreover, many of the giant exons they discussed are neuronal-specific. Therefore, things seem to be more complex than claimed by the authors, and such a crystal-clear conclusion in the Abstract seems misleading.

We modified the abstract to tone down this statement. Human tissue analysis is now presented and discussed in more detail ("SAFB regulation in natural context" results section), and we provide our own experimental insights into SAFB expression and their consequences for post-transcriptional gene regulation in testis (Fig. 5 g and h, and Extended Data Figure 18a). At this moment we cannot provide a straightforward explanation for SAFB-related isoforms specifically expressed in other tissues, but we are intrigued by and keen to share them with other researchers. Our testis analysis shows that although SAFB protein expression levels might be comparable in different tissues, studying their spatial and temporal expression patterns using system- or tissue-specific tools is needed to fully understand the effects on splicing.

2) Fig. 3c: It is claimed that cryptic polyA sites from TEs are used upon SAFB KD. This is clear and convincing in the few examples presented (e.g. Fig. 3d), but not so strongly supported by the genome-wide analysis of Fig. 3c comparing up and downstream parts of affected genes. While the effect is observed and significant, this does not directly inform about the use of the

cryptic polyA sites. This can be properly demonstrated generating 3' seq data for the same conditions and comparing the differential use of cryptic vs. canonical polyA sites.

We agree the genome-wide analysis of the effects of cryptic polyA site usage is a valuable addition to our manuscript as it allows the reader to judge the extent and severity of the post-transcriptional effects of SAFB loss from a perspective not covered by differential gene expression analysis alone. We were, however, hesitant to rely on 3'-seq, since such methods produce short reads we might not be able to map back to the genome unambiguously, as most cryptic polyA sites are inside or right downstream of repetitive elements. Another likely obstacle would be the A-richness of spliced-in repetitive elements, resulting in internal priming. Therefore, we decided to extract this information from the ONT direct-RNA seq data, which has lower throughput but provides complete transcript information, making it feasible to align the transcript with a TE at its 3'-end back to the genome.

We focused on genes supported by on average 20 or more ONT reads per treatment (control or triple KD), annotated their polyA sites from ONT reads, and calculated the fraction of transcripts using a given polyA site for each gene. Out of 14148 genes that meet the expression threshold, we found 247 genes with at least one polyA site having its usage modified by 20% or more upon SAFB loss. Of note, 14148 well-expressed genes include genes with no introns, short introns or very few introns, which are unlikely to have intronic TEs and be targeted by SAFB. In comparison, the number of 1.7% genes with polyA sites affected by SAFB loss (247/14148) increases above 5% when only genes longer than 50 kb are taken into account (231 with diff. polyA usage out of 4433 in total). We now report these genome-wide findings in the manuscript section "SAFB intronizes L1 and Tigger transposons" and provide details of our analysis in the Methods.

3) If I understood correctly, it is suggested that activation of cryptic splice sites in the TEs upon SAFB KD leads to exonization of the TE, which is what then leads to the activation of the cryptic polyA site (lines 159-163). I have several doubts here. First, how "common" is the coupling between activation of cryptic splice sites and the use of cryptic polyA sites (line 158)? This is important information to know, since "common" may refer to 20% or 95%, and this will have different mechanistic implications. Second, it is also possible that recruitment of the transcription termination machinery enhances somehow spliceosomal recruitment (see also next point). A relatively simple way to assess this could be to knock-down the spliceosomal or the termination machinery, and see if the usage of cryptic polyA sites or cryptic splice sites is affected, respectively. In other words, the mechanistic link between the two events could be better elucidated. However, I acknowledge that this does not cast doubts on the validity of the overall observations.

While our main intention was to simply describe coinciding phenomena observed in short read RNA-seq coverage tracks, we agree that the use of the word "coupled" might have mechanistic implications that cannot be confirmed using short read RNA-seq data alone. Therefore, we decided to replace "coupled to" with "coinciding with" in the manuscript.

On the other hand, direct RNA long-read sequencing data do confirm splice-in:polyA coupling, meaning that we observe reads sequenced from a cryptic PAS through the intronic TE, and spliced to an upstream exon, in line with our initial interpretation of short-read coverage tracks. In case of the example presented below, ~63% of reads (31 reads out of 49) that start at the premature PAS also support splicing in of the L1 element, while the rest support complete

fusion of the constitutive exon with the SAFB peak (or an intermediate transcription product that is polyadenylated, but not yet completely spliced). We find the idea of quantifying these outcomes and moving towards understanding their kinetics very exciting. However, unlike the PAS selection discussed in the previous point, there are less reads available for thorough quantification of splicing outcomes (as read length also becomes a limiting factor), and we were not able to perform satisfactory genome-wide quantification of splicing: PAS coupling. For the time being, our main message, that we believe we do provide enough evidence for, is that both splice acceptors and polyA signals, when not masked by SAFB proteins, do get recognized and used by the transcriptional machinery. In many individual cases, presented in the manuscript and easily observable in the short- and long-read RNA-seq data provided via GEO, the two are generally coupled.

4) Perhaps the least convincing mechanistic claim is that related to the role of SAFB in relation to the GAAGAA motifs. This motif is mentioned to be an ESE, and it is suggested that SAFB proteins mask it. I do not think this is properly demonstrated and, in fact, I would tend to think it is incorrect. From the work presented, it seems to me SAFB may repress spliceosomal assembly directly, not by avoiding the binding of other "promoting" proteins (SR factors). First, the authors say in lines 209-210 that "Our results show that SAFB proteins directly compete against SR-proteins by binding to GAAGAA". This claim was surprising, since (unless I missed it) this has not been shown anywhere. Perhaps the authors have evidence for it, but it is not presented. A nice and obvious way to show this would be to perform FLASH of SR proteins upon SAFB KD (they have done this for WT cells only and do not comment on the overlap of binding, as far as I can tell, only briefly for SRSF12). Second, the authors say that "If SAFB depletion turns TE sequences into splice enhancers, then we would expect to find novel splice sites not just inside SAFB-bound TEs, but also in the local vicinity of these elements". I am not sure this is the case. ESEs should act on the exonized sequence, not in the "intronic" vicinity. However, if SAFB motifs act as splicing silencers, with SAFB proteins directly repressing spliceosomal recruitment, then the vicinity effect may be expected. Also, stronger splice sites may be tolerated in the surroundings, precisely because SAFB represses them. Third, why does SAFB not outcompete SR proteins binding to GAAGAA motifs in regular exons, but only in TEs and seemingly long exons? And could SRSF3 be involved, as it is associated with long exons? Certainly, figuring out the mechanistic details on the splicing roles of SAFB is complex and likely beyond the scope of this study. But the authors should be careful with their claims and wording and discuss these issues more explicitly.

We thank the reviewer for this comment, as it sparked a set of interesting experiments. As suggested, we now carried out FLASH experiments using a monoclonal antibody against phospho-SR proteins (1H4) upon depletion of SAFB proteins in HEK293 cells. As a control, we carried out FLASH experiments using an antibody against DHX9, which binds almost exclusively to inverted Alu transposons that are overwhelmingly intronic. These results are summarised in the new Figure 4. Essentially, looking at the regions recovered by 1H4 agnostically, that is without restricting ourselves to SAFB peaks or any other specific target, depletion of SAFB proteins leads to an increase in p-SR signal specifically at sense-L1 elements, suggesting that indeed SAFB proteins are a major part of the L1 RNP forming upon transcription, and is likely competing with SR proteins for binding.

We cannot completely exclude reviewer's point about SAFB proteins rather repressing spliceosomal assembly, but we have carried out purifications with all three SAFB proteins and analysed their co-interactors by MS, and we do not really see a link to the spliceosome (Extended Data Fig. 11), thus at least with the evidence we have, we think it is unlikely.

We are also perhaps a bit misunderstood with the statement about how -potential- TE-derived ESEs are supposed to work in SAFB depleted cells. We certainly agree that an ESE should work from inside the exon, and this is also what we see: SAFB-bound regions appear to be completely exonized, whether it's an L1 or Tigger element, or a simple GAGA(n) repeat. In other words, SAFB peak, which we claim functions as a splice enhancer after SAFB depletion, becomes exonized as a whole. What we do not always observe is the splice-site itself directly within a SAFB peak, (two examples are shown in Extended Data Fig. 9), and when we plot the distance of upregulated splice acceptors in SAFB depletion to SAFB peaks, we can see that they are close to the SAFB peaks compared to controls (Fig. 3g). What we mean by the vicinity

argument is that additional sequences between a SAFB peak and usually a splice acceptor can also be exonized.

We still added a statement acknowledging that SAFB proteins could be directly blocking spliceosomal assembly, since we cannot completely exclude it.

As for the comment about how/why SAFB seems to be specifically repressing long exons and TEs, we also did not expect to be able to address this point, but we found an unexpected explanation as a result of an experiment we carried out to address a point raised by Reviewer#3, which we think is also more consistent with the competition model. In short, we wanted to see whether the RRM domain of SAFB would be able to bind GAA-repeats better when the adenines are replaced with N6-methylated adenines (m6A), which is suggested to be the case for L1 elements in vivo. As a control, we used the RRM of TRA2B, which is structurally characterised to bind this exact same motif (Cléry et al. 2011). We found that while SAFB seems to be a weaker binder compared to TRA2B when it comes to unmethylated RNA, it still binds to methylated RNA, while binding of TRA2B is completely abolished (Fig. 4b). Interestingly, one of the earliest observations about m6A-methylated regions is their enrichment over long exons (Ke et al. 2015; Ke et al. 2017; Murakami and Jaffrey 2022). Thus, we think that as the exon gets longer, it is more likely to be methylated before it's spliced, pushing it towards interacting with SAFB rather than TRA2B (potentially other SR and SR-like proteins as well). Interestingly, this effect would also be contingent on the mode of splice-site definition: if the acceptor of the long exon is quickly spliced to the upstream donor site, which is more in the direction of "intron definition", then it would likely be insensitive to SAFB-regulation since the sequences it can influence are yet to be transcribed then methylated. This also fits with TE-derived RNA on the sense strand, since they are essentially long, single-exon genes that are not spliced, they are by default more likely to be methylated and shift towards SAFB binding.

More minor points:

- Much of the Discussion is in fact results. Perhaps these should be placed in the results section, and the Discussion should probably elaborate more on the functional relationship with Piwi and the mechanistic insights with respect to splicing/polyA site selection/repression.

In the revised manuscript we added a new section titled "SAFB regulation in natural contexts" that now also includes immunofluorescence data from testis, and added a new Discussion section.

- Even though redundancy is expected based on homology, SLTM seems to be a bit of an outlier with respect to SAFB1 and SAFB2 in the different analyses. Moreover, in Fig. 2g it seems that the single SLTM KD *significantly* and substantially reduces retrotransposition efficiency. Therefore, I wonder, have the authors done double KDs? Is it really possible that SLTM is not playing the same role? I acknowledge, however, that this is not a central point for the study.

This is again a very good point raised by the reviewer. SLTM is indeed the odd one out also in terms of homology and evolution. We have done double-KDs, and looked at the response of different types of splicing events to address this point more explicitly (Extended Data Fig. 6c-f). This analysis revealed that SAFB1 is the dominating protein, additionally depleting SAFB2 or SLTM always increases the severity of the molecular phenotype, SAFB2 usually with a stronger effect, but triple-KD is always worse than any other double-KD (Extended Data Fig. 6f). We also carried out mass-spectrometry experiments after purifying SAFB1, SAFB2 and SLTM, to perhaps find different sets of interactors for SLTM, but the binding partners that we detected were essentially the same (Extended Data Fig. 11), although SLTM's interactions are slightly but noticeably different (Extended Data Fig. 11c). Thus, either the divergent C-terminus of SLTM simply alters its interactions with the same set of proteins, or it perhaps has unique interaction partners in a specific tissue or developmental stage that we cannot recapitulate in HEK293 cells. It still seems to play redundant functions with SAFB1 and SAFB2 though, for example we discovered that HCT116 cells do not express SAFB2 (Extended Data Fig. 6b), and in this cell line depleting SAFB1 increases SLTM at the protein level, whereas we see this between SAFB1 and SAFB2 in other cell lines (Extended Data Fig. c,d). At this point, it seems that the best way to understand the role of these proteins is to remove all three at the same time in order not to be affected by the complicated redundancy and self-regulation at play.

- The joint UMAP-based analysis of the FLASH data is spectacular. However, its description in lines 51-66 seems suboptimal, and not very easy to follow. I just the authors give this some more thinking.

We modified this section to make the description easier to follow.

- Line 42: Mention the cell type in which FLASH was performed.

Added.

- Ext. Data Figs. 4 and 5 are too large and not very clear and useful.

These figures were modified to make them more clear.

- Ext. Data Fig. 6: As presented, the similarity in the effects of the KD in HEK and HeLa is only anecdotal (a few examples and genes). I have no reasons to doubt it, but it is not properly shown (e.g., by plotting the fold change in both conditions of affected genes).

We now provide systematic LFC comparisons of KD effects in HEK293, HeLa and HCT116 cells (Extended Data Fig. 7g). Since the LFC-shrinking step in differential expression analysis shrinks the LFC values of low-expressed genes towards zero, we plotted only genes supported by, on average, 50 or more RNA-seq fragments per library.

Referee #2:

Ilik et al. present data exploring the relationship between SAFB proteins, transposable element (TE) activity, splicing, and gene expression. Strong claims are made around SAFB proteins repressing L1 retrotransposons, acting as an RNA-based defence system against TEs (like the HUSH complex) and the resultant "tuning" of TE sequences in somatic cells. The evidence presented leaves open too many alternative explanations. It seems likely the SAFB triple KD produces a generalised increase of nuclear export, which would be less exciting given the known roles of the complex, and go against a TE-specific mechanism. I realise the authors will likely find this a disappointingly unenthusiastic viewpoint, but I have explained my reasons in detail below and signed my review.

We thank Dr. Faulkner for his review, and we will refer to him as Reviewer#2 or "the reviewer" to be consistent with the rest of this letter. We think that the reviewer's "disappointingly unenthusiastic viewpoint" can be traced back to a small number of misunderstandings about the presented data, which we tried to clarify with new experiments addressing concerns about transcriptional vs post-transcriptional regulation with the use of ORFeus and the HUSH complex, and with a fractionation experiment to address the "export only" model raised by the reviewer. We also provide more clear explanations which we hope will resolve reviewer's issues with the analysis of TE expression.

There are also other new experiments that we hope can help the reviewer to contextualise the results, which we highlighted in our response where possible, for example the gel-shift experiments linking RNA methylation, which is highly relevant to the transposon field in general, and L1 field in particular, imposing differential interactions of SAFB and TRA2B on RNA (Fig. 4b), and preferential interaction of SR proteins with intronic sense-L1 elements (Fig. 4a). We also highlight the literature on the role of SR proteins in RNA export in addition to their well-known roles in splicing.

1) Lack of evidence for specific L1HS repression by SAFB.

Fig. 2g shows the results of an L1 retrotransposition assay conducted using a luciferase reporter system from Wenfeng An's lab. It's not said, but here the positive control is a pCAG-driven L1RP element, which is extremely mobile in vitro, and boosting L1RP transcript abundance beyond this pCAG-driven level is known to negligibly impact retrotransposition rate. At a certain point, making more L1 mRNA has no impact on jumping. If SAFB is acting as an L1HS transcriptional repressor, it is therefore very unexpected that SAFB (triple) KD would boost L1RP mobility a further 3-fold.

We thank the reviewer for his comments, the assay set-up in the Methods section included the description "pYX017", which is the name given to this plasmid by the An lab, which was also cited and acknowledged. We now also added the note that this is a pCAG-driven L1RP element based on the reviewer's comment, which we agree can be useful to interpret the results.

We also completely agree with the reviewer's assessment and would be very happy to cite a primary research paper that could back these claims up, especially: "At a certain point, making

more L1 mRNA has no impact on jumping.", since this is also exactly what we are thinking, however to our knowledge no such systematic analysis has been carried out.

We again completely agree with the reviewer's conclusion that it would indeed be very unexpected that SAFB (triple) KD would boost L1RP mobility by a further 3-fold, *if* SAFB proteins were acting as L1Hs transcriptional repressors.

Indeed, we do not think that SAFB proteins are transcriptional regulators of L1Hs, and this was not a claim in the paper.

Our results in the original submission strongly suggest that SAFB proteins are primarily post-transcriptional regulators: they bind RNA, they regulate splicing/termination/export, these are obvious signs of post-transcriptional regulation.

We also now provide data utilising ORFeus and the HUSH complex to strengthen the idea that SAFB proteins are post-transcriptional, RNA-based regulators that is distinct from the HUSH complex (discussed in more detail below).

Technical points: data for the L1 RT- control were produced but not shown.

As indicated in the methods section, the L1 RT- control in this assay is the "pYX015" from Wenfeng An's lab, which is actually a mutant of ORF1p that prevents retrotransposition of L1. This plasmid is used as a "background" control, against which all the values are normalised to. More clarification for this is added to the Methods.

The replicates are said to be biological (which is normally considered experiments conducted on different days), but appear to instead be technical (multiple wells on the same day) and so the SD would likely be substantially larger if replicated twice more.

We removed the word "biological" from the figure legend.

The L1 probe in Fig. 2h here is designed against L1HS ORF2, which is highly conserved amongst L1s and therefore this probe is very unlikely to be specific to L1HS, which may explain why the signal is so strong. It would be prudent to test these probes in mouse cells, and in mouse cells expressing the L1RP plasmid, as controls.

This assessment is true, and it is exactly the reason why the plot is labelled as "L1 RNA". However the title on the top of the figure stated "RNA FISH (endogenous L1Hs)" which can indeed be misleading, thus was changed to "RNA FISH (endogenous L1)".

We now also carried out northern blots in cells that were transfected with L1Hs or untransfected (Fig. 2k), and additionally knocked-down SAFB proteins and the HUSH complex to determine how the signal changes when one depletes transcriptional regulators (HUSH), post-transcriptional regulators (SAFB) or both. As one would expect, removal of both transcriptional and post-transcriptional inhibitors lead to the most dramatic increase in L1 expression (Fig. 2i,k)

Fig. 2d: How was this plot generated? The FLASH reads would not align to L1HS loci uniquely and so this must be generated by aligning them directly to an L1HS sequence? It is conspicuous that no L1HS loci are presented in IGV-style figure panels; they're all older L1s, and so presumably FLASH reads rarely appear on L1HS sequences displayed in this way.

We apologise for not being explicit enough when describing how this plot was generated. The motivation behind this figure is pretty simple: when it comes to older L1 elements we are able to uniquely map FLASH reads directly on the elements, which allows for direct visualisation and analysis. However, as the reviewer also indicates, L1HS is an active transposon, its insertions are recent, making it difficult (i.e. nearly impossible) to map reads uniquely onto them.

There is no “ominous” reason about the paucity of L1HS screenshots: for SAFB proteins to bind an L1HS element inside an intron, (a) the gene must be expressed (b) there must be an L1HS insertion within the gene (c) the insertion must be on the same strand as the gene (which on average is less likely than the opposite strand, see Extended Data Fig 4a where each line is an L1 within a gene, or see Han et al. 2004). In short, there are few such cases, here’s a full-length L1HS element (6032 bp), inserted on the same strand as the RASEF gene:

L1 elements are coloured by their strand of insertion; blue: + strand (Watson), red: - strand (Crick). RASEF gene is on the - (Crick) strand.

In case the labels are too small to read: Top 4 traces are coverages of the FLASH data, from top to bottom: FLASH data of ectopically expressed SAFB1, endogenously tagged SAFB1, ectopically expressed SAFB2 and ectopically expressed SLTM. The bottom two are the normalised RNA-seq coverage of control siRNA-treated 293 cells and triple siRNA treated 293 cells.

We think that the most parsimonious explanation here is that SAFB proteins bind to this L1HS element, as they do other L1PA, L1PB and many L1M elements, all of which have very similar, sometimes identical ORF1p and ORF2p sequences. The binding appears “patchy” most likely due to the inability to uniquely map FLASH reads all across this recent insertion, as alluded to by the reviewer as well. The RNA-seq data is also extremely informative: Even with paired-end 2x100bp sequencing, it is very difficult to find unique reads mapping on this element, resulting in a patchy pattern, resembling the FLASH coverage, suggesting that the same SNPs allow unique mapping around these regions.

Looking at the RNA-seq coverage, it is also clear that the depletion of SAFB proteins leads to a dramatic increase in the coverage seen on this L1Hs element. Note that this increase is concomitant with a reduction of coverage for all the exons downstream of this element. To highlight this fact, we drew pink dotted lines parallel to the maximum height of the first exon.

There are two hypotheses that could explain these results: (a) L1Hs element is autonomously expressed, i.e. it is expressed from its own promoter independently from the promoter of the RASEF gene. SAFB proteins bind to its RNA and prevent its nuclear export, and their depletion releases the L1 RNA into the cytoplasm, stabilising it, leading to an increase in the coverage we observe, or (b) The "expression" of this L1Hs element is driven by the RASEF gene, i.e. the first exon of the RASEF gene is spliced into a cryptic splice acceptor upstream of this L1Hs element, which is suppressed by SAFB proteins. Depletion of SAFB proteins increases the likelihood of this splicing event, increasing the coverage of the L1Hs gene.

If the first hypothesis is true, up-regulation of the L1Hs element should not have anything to do with the expression of the exons downstream of it, since the expression of the L1 would simply be driven by its own promoter.

If the second hypothesis is true, this amounts to an "alternative splicing" event which means that the first exon of RASEF is either spliced to the second exon of RASEF gene, or it is spliced to a cryptic splice site upstream of the L1Hs element. This means that, in this scenario, an increase in this particular L1Hs element should be coupled to a decrease in the expression of the exons of RASEF downstream of this element. As can be seen in this example, the apparent up-regulation of the L1Hs element is coupled to the downregulation of all the exons of the RASEF gene downstream of this L1Hs element, indicating that this is a form of alternative splicing.

The Sashimi plot of the locus illuminates this case even further. Sashimi plots show the count of "gapped alignments", which are encountered when a part of a read is mapped to one exon, and another part of the read is mapped to another exon. These alignments therefore show where splicing has occurred on the nascent transcript, and can be used to map splice sites and quantify their usage, which we have done extensively in this manuscript.

When or if an exon is spliced to multiple exons, multiple lines emanate from this particular exon, indicating that this exon can engage in "alternative splicing". Even though these lines are represented all together on such figures, it is physically impossible for them to happen on the same molecule, hence the term "alternative" in alternative splicing. As we can see below, in control cells $69 / (69 + 257) * 100 \sim 21\%$ of gapped-alignments are mapped to a cryptic splice site ~ 110 nt upstream of the L1Hs element. Upon depletion of SAFB proteins, $256 + 23 / (256 + 23 + 164) * 100 \sim 63\%$ are mapped to two cryptic splice sites upstream of this L1Hs element, note that a new cryptic splice site that is not detected in control samples emerges in SAFB depleted cells.

In other words, in control samples, approximately 1/5th of host gene transcription is "consumed" by the L1Hs element, whereas upon triple depletion almost 2/3rd of transcription is hijacked by this element, culminating in a decrease in the expression downstream exons of RASEF. Overall, a log2-fold change of -0.38 (padj = 1.68E-20) is reported by DESeq2.

In short, the plot was generated with these two thoughts in mind:

First, the few examples we see with intronic L1Hs elements fit the rest of the phenotypes we see with older insertions i.e. splicing/termination defects upon SAFB depletion, therefore the most likely explanation is that we are looking at the same underlying molecular events, thus it would be reasonable to show alignments on a "canonical" L1Hs element.

Second, the binding we observed more clearly on older elements, since we can uniquely map more reads, seemed to concentrate on the open reading frames of L1 elements. Fusing these ideas into one, we mapped FLASH reads directly onto a full-length L1Hs element where we annotated the positions of ORFs and UTRs. We also did the same to a reconstructed Tigger DNA transposon, and observed the same exact pattern: binding is concentrated on the sense-strand of both transposons, specifically on the coding sequences. We hope that the motivation behind these plots, and their scientific purpose are more clear.

In this figure, the use of sense/antisense as a comparison is then hard to follow, and one would imagine most TEs have stronger FLASH signal on the sense strand.

We think that what confuses the reviewer is this thought: if SAFB proteins bind to L1 RNA, well shouldn't it always bind the sense strand? The answer is "yes", but the reads from FLASH originate overwhelmingly from intronic regions which is actually an abundant source of both sense and anti-sense TE RNA. Therefore, it is difficult to overstate how incredibly specific SAFB proteins are in their binding when they bind to sense-TE-derived RNA in the nucleus. This is what this plot is showing, together with the fact that their binding is extremely concentrated on the ORFs. This result is the motivation behind looking into retrotransposition: if SAFB proteins are so incredibly specific to sense-L1 RNA, even binding to extremely old elements that were inserted millions of years ago into the ancestral genome (L1M family for example) as well as recently inserted L1Hs elements, do they actually have anything to do with L1 life-cycle? Which we then interrogated using the retrotransposition assays.

Fig. 2: The siRNA experiments lack detail. Was one siRNA designed per protein? Immunoblots are not shown to indicate efficiency of protein KD. It is surprising that the triple KD gives such an effect on retrotransposition in Fig. 2g when KD against each of the SAFB proteins singly gives opposing results.

We apologise for omitting immunoblots that show effectiveness of siRNA treatments, which was criticised by multiple reviewers, but the siRNA information was already available in the Methods section. We now provide a large amount of immunoblots, which not only show the effectiveness of the siRNA treatments but how in some cases SAFB proteins regulate each other's expression as well, justifying depletion of all three as a reasonable way to uncover their full potential in regulating gene expression as well as retrotransposition of L1 elements (Extended Data Fig. 6b-f).

We think that this statement of the reviewer: "(...) when KD against each of the SAFB proteins singly gives opposing results" is a mischaracterisation of the data presented in the manuscript. We can only assume that the misinterpretation of the data presented in Fig 2g must have led to the reviewer's dramatic statement in the preamble of his review: "Strong claims are made

around SAFB proteins repressing L1 retrotransposons (...)" . We believe we made very reasonable claims about SAFB proteins repressing L1 transposons, which is also supported by at least two other papers as well: SAFB1 is a hit in the L1-repressors screen by the lab of Joanna Wysocka, which has uncovered the role of HUSH complex in L1 biology (Liu et al. 2018). Importantly this screen used a different reporter (GFP instead of luciferase), a different cell line and different promoter. Similar results with SAFB1 was also reported by the lab of Wenbo Li (Xiong et al. 2021), where the more classical Neomycin resistance and colony counting is used to show that depletion of SAFB1 increases retrotransposition of L1. The claim we make is that these proteins are, in part, redundant with each other, therefore depletion of all three unleashes the actual repressive activity these proteins exert on L1. This claim is further supported by the RNA-seq data, which mirrors the results of the retrotransposition assay: among single depletions, most changes are scored in SAFB1 KD, dramatic changes are seen in triple KD, and mild changes are observed in SAFB2 and SLTM single depletions. We hope that the new data presented in this revised version, including immunoblots from single depletions, results from HCT116 cells where SAFB2 is not expressed, as well as RT-qPCR analysis of single, all double-combinations as well as triple-KD (Extended Data Fig. 6) will make it clear that SAFB1, as also seen in the retrotransposition assay, appears to be the most important factor among the three, but depletion of all SAFB proteins expressed in a given cell line is necessary to completely remove repressive activity of SAFB proteins.

Fig. 3a and Ext. Data Fig. 7 show that the impact of SAFB KD on endogenous L1HS elements, as measured by a non-standard TE analysis of RNA-seq, is not significant.

We apologise for not providing enough details about the TE analysis pipeline. It looks like we only indicated that it was carried out with "snakePipes noncoding-RNA-seq" pipeline in a figure legend, which we assume must have led to this short assessment from the reviewer.

Essentially, snakePipes is a flexible, genomics data pipeline with in-built quality controls, developed and maintained by the Manke lab (Bhardwaj et al. 2019), which is also the same team that developed and maintains deepTools. The "non-coding-RNA-seq" module uses Tetranscripts and DESeq2 to quantify repeat expression together with gene expression.

We assume that the reviewer knows how Tetranscripts works, but briefly, Tetranscripts counts reads that fall onto mRNAs, together with TE sequences derived from RepeatMasker (in fact, all repeats, such as centromeric repeats, simple repeats etc., not just TEs), and uses an EM-algorithm to assign reads to TEs, while counts on mRNAs are used to normalise the data, since reads from TEs, even in extreme cases, are sparse and not suitable for typical normalisation methods, or could be "normalised out" (i.e. interpreted as increased sequencing depth) in case of systematic increase of TE coverage. DESeq2 is then used for statistical analysis of the data, which then reports log2 fold-changes as well as associated (adjusted) p- values. We hope that this explanation clarifies the issue with the "non-standard TE analysis" part of the reviewer's comment.

As mentioned in the explanation of Tetranscripts, this analysis simultaneously analyses the expression of all repeat elements in the RepeatMasker output, which includes all SINEs, LTR elements, DNA transposons, and of course LINES. In this sense, neither Fig. 3a, nor Extended Data Fig. 7 (in this submission, Extended Data Fig. 8) shows "(..) the impact of SAFB KD on

endogenous L1HS elements (..)", as claimed by the reviewer. Fig. 3a shows the impact of SAFB KD on all repeat elements, where we see mainly LINES (red dots) getting "apparently upregulated", some of which are so significant that their adjusted p-values are reported to be 0, thus their $-\log_{10}$ values are capped at 300 by the "EnhancedVolcano" package, showing that they are highly significant. Extended Data Fig. 8a shows exactly the same analysis, this time focusing on the top20 most significantly changing repeats (this is a standard output from snakePipes), sorted by their adjusted p-values, are shown with their full names. This is to show that the LINES seen in Fig. 3a are a diverse family of L1 elements, including L1Hs and other L1PA elements, but also older L1P and L1M elements are clearly visible, oddly, together with two Tigger elements, which one can go back and verify that they are present on Fig. 3a.

We now also provide the DESeq2 output of the Tetranscripts analysis for all the datasets reported (Supplementary Data Tables 4-11).

We would also be happy to run any other analysis suggested by the reviewer on our data. We are confident that any well-explained data pipeline will find the events described in this manuscript very easily, and recapitulate our main findings.

More to the point, the use of Z-score for colouring in Ext. Data Fig. 7 is likely to mislead as to the strength of these effects. What justification is there for strongly colouring a Z-score of 1?

These are direct, unmodified outputs of the snakePipes software, using Tetranscripts at the back-end which can be easily and completely reproduced.

We find it valuable to simply provide these outputs as they are, but we also provide the same outputs as Volcano plots in Fig. 3 and Fig. 4 (in this submission, Fig. 5), where the effect sizes can be more directly judged as they report the \log_2 -fold changes on the x-axis and $-\log(p\text{-value})$ on the y-axis. We are also now providing the DESeq2 outputs of the Tetranscript analysis for a more in-depth look into the data.

We note that the z-scores in these figures result from row-normalisation of input values, which is a standard procedure to make heatmaps with high dynamic range across rows easier to read, and actually have nothing to do with the strength of the effect, which we now clarify in the figure legend.

As it can be seen in the figure, all replicates are plotted separately and all receive a z-score. The point of these plots is to quickly check the most significantly altered repeat elements (TE or others), and to see how well the replicates agree with each other, providing a measure of robustness of the output, which is really important when judging repeat expression since the data points are typically sparse.

Just to illustrate what these plots are doing more clearly: if all three replicates of control have the same value, say 10, and all three replicates of the treatment has the same value, say 100, the z-score for each replicate of control would be -1, and for each replicate for the treatment would be +1. The only way to get a z-score larger than +1 in these plots is to have a replicate that is dramatically different than the others, for example if replicate 3 of the treatment had a value of 1000, then the z-scores become (-0.545, -0.545, -0.545, -0.2935, -0.2935, 2.2219),

which would have a very striking colour combination. We therefore think that the colours are indeed functioning as intended, at least for this type of plot.

To reiterate: z-scores in this figure are not the z-scores used to determine p-values, these are only to look at agreement between replicates for the most significantly changing elements in the dataset.

The claim that SAFB KD upregulates L1HS is not based on robust data.

We hope that the additional clarifications we provided up to this point will lead the reviewer to re-evaluate this assessment.

We have also considered how else can we test our claims and reasoned that if SAFB KD actually upregulates L1Hs as we claim in the manuscript based on L1 retrotransposition assays, RNA-FISH and RNA-seq data, then the best way to test this claim further would be to compare the effect of SAFB depletion to the depletion of a well-known transcriptional regulator of L1Hs expression.

We chose the HUSH complex since it is not only a well-established transcriptional regulator but also an RNA-based transcriptional regulator that can perhaps be more directly compared to SAFB regulation.

As expected, depletion of the HUSH complex (we depleted both TASOR and MPP8 to completely remove all HUSH complex) indeed leads to a robust increase in ORF1p expression (Fig. 2i, compare lane 3 to lane 1).

Depletion of SAFB proteins also leads to an increase in ORF1p expression, which is similar in magnitude to HUSH depletion (Fig. 2i, compare lanes 2 to 1 and 2 to 3).

These results suggest that if HUSH complex is indeed a regulator of L1Hs expression, then SAFB proteins should also be considered as regulators of L1Hs expression.

Furthermore, co-depletion of SAFB and HUSH leads to an even further increase in ORF1p expression that is stronger than either depletion alone (Fig. 2i compare lane 4 to lanes 2 and 3), which we additionally verified at the RNA level by a northern blot using a probe against ORF2 (Fig. 2k).

These results indicate that, since HUSH complex is a robust transcriptional regulator of L1Hs, then SAFB proteins are most likely post-transcriptional regulators that interact with L1 RNA after transcription and prevent their passage to the nucleus, therefore removing these regulators together amplifies the effect of L1 upregulation.

Finally the "tuning" idea rests on the fact that L1 RNAs have evolved towards and maintain an adenine bias (we provide Fig. 2j to make this point more clear), SAFB proteins recognise this pattern, and since even the fruitfly orthologue of SAFB proteins, Saf-B, recognises the same RNA pattern, then it is likely that L1 RNAs are maintaining this bias, at least in part, to be repressed by SAFB in the soma, in case their transcriptional repression fails.

If this is true, then if we remove this A-bias from L1, then it should not be under the control of SAFB proteins anymore, which is a very stringent test for our hypothesis. Boeke lab has removed the A-bias from L1 RNA by recoding its ORFs and created a hyperactive transposon that they named ORFeus (Han and Boeke 2004).

Reassuringly, and proving our point, expression of ORFeus, unlike L1, is not dependent on SAFB proteins (Fig. 2i, compare lane 6 to lane 5).

Equally interestingly, HUSH complex, which is mentioned as a specific regulator of L1Hs expression, still represses ORFeus (Fig. 2i, compare lane 7 to 5), which indicates that the A-bias in L1 could not have evolved for HUSH repression, since HUSH can repress ORFeus equally well.

We conclude that SAFB KD can up-regulate L1Hs, if L1Hs escapes transcriptional regulation which can and does happen in a variety of physiological scenarios.

It is also interesting that YY2 is upregulated in this KD as YY2 perturbation may contribute to the mild change in L1 expression observed as YY2 (and YY1) can both bind the L1HS promoter.

This is an interesting point, but YY2 is **not** upregulated in SAFB depleted cells.

YY2 is "apparently upregulated" in SAFB depleted cells because more reads fall onto its transcript annotation upon depletion of SAFB proteins and is therefore reported as "upregulated" by DESeq2 and would be scored as such by any other software that analyses count data.

What is happening at the YY2 locus is the following: a cryptic splice site upstream of the YY2 retro-gene, which is bound by SAFB proteins, is activated in SAFB depleted cells which is then spliced to exon 5 of MBTPS2 gene, terminating expression at the functional pA site, creating a chimeric transcript that contains the first 5 exons of MBTPS2 and a part of the normal YY2 transcript.

Note how exons 1-5 of MBTPS2 are not down-regulated in SAFB depleted cells while exons 6-11, which are downstream of YY2 are severely down-regulated – a common theme in this manuscript.

Note also that MBTPS2 is one of most severely downregulated genes in SAFB-depleted cells and annotated on the same plot that shows "apparent" YY2 up-regulation.

These conclusions were obvious from Illumina data already, but directRNA sequencing using the ONT platform, where mRNAs are directly fed through a nanopore, provides a striking direct visualisation of these events:

As can be seen hopefully clearly on the resized image, there is no independent YY2 expression. In other words, YY2 is not using its own promoter for expression, which is in line with the fact YY2 is rather specifically expressed in the testis. The entire up-regulation scored in RNA-seq data is a result of YY2 essentially hijacking MBTPS2 transcription through a cryptic splice site that is normally suppressed by SAFB proteins.

We would not exclude a physiological link to L1 regulation, and we assume there must be a link: YY2 is expressed most notably in testis, and mouse Yy2 gene resides in the same intron of Mbtps2 gene, and SAFB depletion in mouse 3T3 cells causes the same exact phenotype. One possibility is that SAFB downregulation during spermatogenesis promotes YY2/Yy2 expression when it is expressed from its own promoter, however we have no evidence for this at this time.

We hope that the parallels between splicing/termination defects caused by L1, Tigger and pseudogenes are more clear with this interesting example.

Bottom line: There is a lack of support that SAFB is a repressor of L1HS retrotransposition, or L1 transcription in general, and that is a major problem for the main claims around host defence in somatic cells.

We hope that the additional data, direct biochemical comparisons to the HUSH complex, detailed explanations and visualisations provided in this letter will clarify these issues for the reviewer.

2) SAFB is a multifunctional protein beyond splicing.

The data in Figs. 2h and 3d give very good reason to entertain an alternative, perhaps more mundane, explanation for the TE part of this manuscript, namely that SAFB impacts nuclear export and that this phenomenon is not specific to TEs.

The examples of SAFB KD in mouse and fly (Fig. 4b,c) support this view, as they don't describe TEs, and for some of the examples (e.g. Ext. Data Fig. 6) there is appreciable signal on the L1 sequences highlighted in the control RNAi experiments.

Crucially, the polyA+ RNA-seq was prepared from total RNA, which for various reasons tends to be dominated by the cytoplasmic fraction. Therefore, if SAFB KD increases the export of previously nuclear-enriched transcripts, such as those containing intronic L1s, they would appear much more abundant in the RNA-seq data. However, this would not be the result of altered splicing, it actually would be the result of altered nuclear export. That would also be consistent with the retrotransposition assay data from Fig. 2g, where mobility increases much more than one would expect, because SAFB KD enhances export of the L1 mRNA, which is more rate-limiting than transcription.

We appreciate the reviewer's efforts to merge the role of SAFB in the inhibition of "nuclear export" of TEs, to the post-transcriptional phenotypes we observe in the RNA-seq data. We agree that this would indeed be a simple explanation, but unfortunately this would not be explaining the data, but "explaining away" the data. We have also entertained the "alternate hypothesis", however looking at the data carefully, which is to say beyond read counts at transcripts, it is clear that it fails to explain the molecular phenotypes observed.

There are two ways the "alternate hypothesis" could work:

(a) either TEs drive their own expression with their own promoter, and they are simply exported more efficiently in SAFB-depleted cells

or

(b) SAFB-bound pieces of TE elements form chimeric transcripts via splicing and/or termination with their host genes upon depletion of SAFB proteins and these transcripts are then stabilised since they are free to move out of the nucleus.

We hope the reviewer can agree that the first variant is rather absurd and does not require a response, since it cannot explain why virtually all of the affected TEs are on the same strand as the host gene: if TEs were transcribed from their own promoters, the strand of their insertion should not determine if they will be transcribed or not.

The second variant of the hypothesis requires splicing and/or termination at the TE pA site, which we surmise is the model put forward by the reviewer. In this version, TEs and their host genes form chimeric transcripts at the same rate with or without SAFB proteins, and only in the absence of SAFB proteins these chimeric transcripts escape the nucleus, are stabilised in the cytoplasm and now detected in our RNA-seq data.

If this version of the hypothesis were true, then if we were to isolate nuclear and cytoplasmic RNA separately from control and SAFB-depleted cells, then we would see that the nuclear/cytoplasmic ratio of L1-containing chimeric transcripts would decrease upon SAFB depletion, since according to this alternative model posited by the reviewer, no new chimeras would be generated, and the chimeric transcripts would simply be more efficiently exported out

of the nucleus. We rather see that when the abundance of L1-chimeras increase, they increase in the nucleus and in the cytoplasm, which strongly suggests that more of them are produced via enhanced or in some cases completely novel cryptic splicing and/or termination at the pA of L1 and other repeats.

There are other reasons why the "alternate hypothesis" doesn't work, which formed the basis of a large part of the manuscript and essentially the entirety of Fig. 3:

If "alternate hypothesis" were true, it would not be possible to explain why exons of the host genes downstream of the TE part of the chimeric transcript would be down-regulated (why would exons 6-11 of MBTPS2 be downregulated if YY2 is more efficiently exported?). Perhaps we were not able to make this point more clearer in the figures and the manuscript: the reduction in the expression of these downstream exons are not just relative to the upstream exons, which one could explain with the second variant of the "alternate hypothesis". The reduction is absolute, which is the reason why DESeq2 marks some of these genes as "down-regulated", since they are indeed down-regulated, all the while a large chunk in their introns are up-regulated. These effects are summarised in Fig. 3c transcriptome-wide, with an example gene, CENPQ underneath it. We have now added the text "pre-peak exons" and "post-peak exons" on the CENPQ gene to clarify this relationship to Fig. 3c.

Finally, it is a minor part of the reviewer's comment, but while trying to illustrate the common molecular phenotypes between human, mice and fly data, we seem to have omitted TEs producing chimeric transcripts via enhanced splicing in SAFB depleted mouse 3T3 cells. There was a summary plot on Extended Data Fig. 7i (in this submission, Extended Data Fig. 8i) that shows the enrichment of novel splice sites in mice on LTR and L1 elements upon SAFB depletion, but this is admittedly a bit too obscure. We now also provide a few of these examples on Extended Data Fig. 15a.

3) Much reliance is placed on the use of FLASH to assess RNA-binding by SAFB proteins. It would be useful to provide data from an orthogonal method showing SAFB binding to L1 sequences, to exclude any technical artefacts relating perhaps to the A-rich SAFB motif and their preponderance in L1 sequences.

We think that FLASH in the way it was carried out in this manuscript, namely using the same tag combination for >30 proteins, expressing all of them from the same locus using the same amount of doxycycline and carrying out a very stringent and standard purification that is identical for all the constructs is probably the least likely way to be affected by technical artefacts, which we think is also clear both from the UMAP (Fig. 1) and the more traditional correlation plot shown on Extended Data Fig. 1b. It is also important to mention that we were able to capture diverse binding preferences such as U-rich targets for hnRNPC, C-rich targets of hnRNPK, G-rich targets of hnRNPF, and that several SR-proteins, such as TRA2A, TRA2B and SRSF12 show virtually identical sequence preferences as SAFB proteins, and we do not see their enrichment on L1 or other TEs.

Nevertheless, it is always good to have an orthogonal method as suggested by the reviewer, thus we carried out a RIP-qPCR experiment using a monoclonal antibody specific to SAFB1, using HEK293 cells lightly crosslinked with 0.2% formaldehyde to suppress in-solution interactions as much as possible. We designed primers that specifically amplify the targeted

TE and no other loci in the genome, we included the giant-exons of ANK2, ANK3, CLIP1, and used ACTB and GAPDH which are very abundant mRNAs as negative controls. Results are consistent with FLASH data and summarised in (Extended Data Fig. 6a)

4) It's a more minor point, and yet it is striking and surprising that no mention is made of HUSH. HUSH is a very robust and specific repressor of L1 transcription and retrotransposition (PMID: 34794168), in a way that arguably SAFB does not appear to be.

We thank the reviewer for raising this point, which we discussed in several points above. We originally omitted the HUSH complex since it was clear to us that we were dealing with a post-transcriptional phenomenon and the comparison to a transcriptional regulator did not seem fair, but the comparisons to HUSH, especially in combination with ORFeus have actually been really helpful to improve the manuscript.

We have explicitly and directly tested the effect of SAFB and/or HUSH complex on L1Hs (vs ORFeus) expression, and we have observed a similar magnitude of L1Hs upregulation in either case using immunoblotting ORF1p (Fig. 2i) or northern blotting ORF2 as readouts (Fig. 2k).

We also would like to note that, HUSH is a robust but certainly not a specific repressor of L1 transcription, which is clearly demonstrated by recent Lehner work cited by the reviewer where HUSH represses lentivirally transfected Cas9 among many other examples, but this is also clear from previous work, since HUSH as a complex was discovered as a suppressor of an LTR reported by Paul Lehner (Tchasovnikarova et al. 2015). HUSH is a very remarkable and interesting complex that seems to target most intronless transcripts, and L1 happens to be one of those.

In a limited sense, we also explicitly tested this specificity by using ORFeus, which is a “de-tuned” L1 variant designed by Jef Boeke lab, which is no longer A-rich (Fig. 2j). As can be seen clearly in Fig. 2i, HUSH is perfectly capable of suppressing L1Hs and ORFeus, but SAFB can only suppress L1Hs, which makes it arguably a more specific repressor of L1. However, we do not make the claim that SAFB is “specific” for L1 RNA, in contrast we argue that SAFB recognizes a sequence motif and it is the L1 RNA that has evolved towards this sequence bias to keep itself inactive in somatic cells, which is what we refer to as “tuning”.

5) Another minor point: Many of the IGV examples of splicing into ancient L1s are spectacular. Even if I disagree as to their significance, with regards to tuning the expression of host genes via altered splicing, this does show the potential wealth of nuclear RNAs that we tend to miss with RNA-seq made from total RNA preps.

We thank the reviewer for this comment. The “tuning” we are talking about is the tuning of the L1 RNA through evolution towards an A-rich sequence, which has significant implications with regards to evolution of L1 elements, their activity in the germline vs soma, the role of SAFB proteins in their regulation, and how these TEs can disrupt gene expression in various scenarios that involve SAFB downregulation and/or mislocalization. We hope that at least some of these points become more clear with the experiments involving ORFeus and the HUSH complex.

Geoff Faulkner (University of Queensland)

Referee #3:

This manuscript by Ilik et al. describes how autonomous transposable elements (TEs) use their sequences to be recognized and masked by SAFB proteins and thus avoid their unnecessary splicing and exonization in target genes within somatic cells. They suggest the mechanism is in turn active in non-somatic cells such that TEs can rearrange the genome effectively and propagate their own distribution. The FLASH- and RNA-seq-based data are presented with a bunch of derivable information in them. Almost 30 RBPs had been tested in FLASH to identify clustering and reveal SAFBs as specific binders to TEs, likely counteracting (alternative) splicing enhancing by SRs. The authors also identify that especially giant exons are targeted by TEs (which has in a way been suggested before, see below). As a novel finding they prove that SAFB depletion affects the splicing of TEs, in particular L1, also taking into account the TE's strand location.

Altogether, this study provides a convincing mechanistic insight into how TEs are kept suppressed by SAFBs. Although it remains open what makes SAFBs different from previously described splicing silencers, it well showcases the possible co-evolution of TEs and a suitable set of host RBPs.

The abstract is written clearly and is accessible to readers beyond the ultimate expert level. The same can be said about the introduction in principle. As authors try to put a focus on the role of SAFBs as "immune system component" they might like to introduce SAFBs a bit more in light of their unclear dual nucleic binding capabilities and role on the transcriptional level.

The data, as provided, are sound and trustworthy with sufficient control experiments carried out in most of the cases. I do almost unlimitedly support the conclusions from the presented data, and I would definitely underline the strong value of mechanistic assumptions on the SAFB-TE interactions regarding the comparison between somatic cells and e.g. reproductive tissues with respect to the role of SAFBs as immune system-like regulators.

The identified target motifs of SAFBs are supported by previous studies based at least on CLIP data, while the general question remains as how exactly SAFBs interact with their RNA target sequences.

As far as I can judge it (see my comments where I may not do so sufficiently) the treatment of statistics seems appropriate. The references are very focused on the TE side, which I may not oversee in total. For SAFBs authors have not put much focus on its manifold involvements in RNA processing, also connecting to diseases and underlined by a growing number of strong literature, but this may be related to a somewhat different scope here. Still it can teach us a lot about RNA-binding by SAFB, which is little understood on the structural level.

What appears a bit puzzling to me is that the manuscript misses to link to a previously published study (Xiong et al., Cell Res., 2021) in which the concept of TE suppression by SAFB proteins has been introduced, albeit in a m6A-related context. While the two studies would not need to overlap and in fact do not mechanistically (seeing the final model provided by Xiong et al.: the role of m6A, the place and timepoint of physical engagement with RNA, no mentioning of exonization etc.; in other words: a somewhat different focus), it shows that SAFBs have been identified as regulators of TEs in maintaining host gene expression before.

Also, the SAFB-related target motifs are well in line and e.g. give good value to the still non-understood RNA target preferences for SAFBs (esp. supported by structural data). Still, I assume the herein presented study by Ilik et al. definitely adds a very strong value, a second, expanded perspective on SAFB-TE and its impact on gene regulation and will impact its research field, and on the longer run potentially beyond (seeing SAFBs as biomarkers, etc).

I, however, personally suggest to place the manuscript in its current form in a slightly more specific journal with a cell-biological and/or genetic focus seeing the above-mentioned.

Independently, I see a clear need of addressing the following issues:

- Connecting to the above mentioned: have the authors (meanwhile) looked at the role of methylation for SAFB localizing to TEs and the influence on TE activity in their setup? SAFBs are hardly canonical readers of m6A, but methylation may steer interactions indirectly. Please comment or add findings if available.

We are grateful to the reviewer for this very insightful and prescient comment, it instigated an interesting experiment with surprising results that explained several aspects of this work.

To address this comment, we decided to test the RRM of SAFB1 against the RRM of TRA2B in a gel-shift experiment using (AGA)_{x7} RNA as the probe, where adenosines are either all N6-methylated or unmethylated, which was achieved by replacing ATP with N6-ATP while setting up the in vitro transcription reaction. We chose TRA2B, since it has a well-studied RRM, precisely measured K_d values against its preferred substrate, AAGA, and an NMR structure where it is in complex with AAGA RNA (Cléry et al. 2011).

We carried out gel-shift experiments using Halo-tagged TRA2B-RRM, Halo-tagged SAFB1-RRM, and Halo tag alone as a negative control. We used Halo tag to increase the solubility of the RRM, which we had observed previously as an issue for SAFB1, at the same time we used Halo tag to fluorescently label the proteins we use, which we could then directly visualise in the gel-shift experiment on the same gel and simultaneously with the probe (Fig. 4a), which allowed direct comparison of protein loading.

Finally, the Halo tag is slightly negatively charged at neutral pH, which was crucial because the RRM of SAFB1 and TRA2B are both positively charged at neutral pH and therefore normally run towards the cathode during a gel-shift experiment (i.e. out of the gel). The results are summarised in Fig. 4b. As can be seen, Halo by itself does not bind either methylated or unmethylated probe (Fig 4b, lane 12, protein loading on the right). Halo-SAFB1-RRM and Halo-TRA2B-RRM both interact with unmethylated (AGA)_{x7}, latter with apparently higher affinity (unmethylated (AGA)_{x7}: TRA2B > SAFB1). Unexpectedly, TRA2B-RRM does not seem to interact at all with methylated (AGA)_{x7} probe, whereas SAFB1-RRM's binding is hardly affected (m6AGm6Ax7: SAFB1 >> TRA2B).

Since these experiments were carried out using the RRM domains of either protein involving no other auxiliary domain, it seems reasonable to speculate that TRA2B's RRM folds into a structure that provides a good fit for unmethylated AGA (Cléry et al. 2011), thus making it a relatively better interactor, which is probably also the reason why it cannot interact with m6AGm6A, since it likely does not fit into the RRM due to the bulky methyl groups. SAFB1's RRM

interacts weakly with AGA compared to TRA2B, likely because RRM loosely fits unmethylated RNA, but the binding is consequently not affected by the bulky methyl groups of m6AGm6A. These are of course speculations, and only an NMR or equivalent structure of SAFB1's RRM with methylated and unmethylated RNA can really show the mechanism behind these observations.

These results provide an unexpected explanation to one of the key observations of this manuscript: why do SAFB proteins have a peculiar specificity towards repressing splicing of long exons? Both early and recent studies mapping N6-methylated RNA in the human transcriptome report that one of the most clear patterns found in mRNAs is the enrichment of m6A on long exons (Ke et al. 2015; Ke et al. 2017; Murakami and Jaffrey 2022). Moreover, recent work has shown that deposition of EJC after splicing locally protects mRNA from methylation (Yang et al. 2022; He et al. 2023), suggesting that the longer nascent RNA is exposed to the nuclear environment without splicing, the more likely that it will be methylated.

Since nascent RNA emerges from RNAPII in an unmethylated state, TRA2B has a higher chance of interacting with it, increasing the likelihood of splicing. If splicing is delayed, or not possible due to lack of acceptor/donor sequences, nascent RNA will be progressively methylated, switching its preference from TRA2B to SAFB, and more likely to remain an intron. Since long exons are more likely to be methylated, they are more likely to be suppressed by SAFB, and consequently their splicing is boosted by depletion of SAFB proteins. Although our gel-shift assays only used TRA2B as an example, we also carried out FLASH experiments with an antibody against phospho-SR repeats in control vs SAFB-depleted cells. When we analysed recovery of repeat elements, we see a specific increase in sense-L1 RNA bound by p-SR proteins(Fig. 4a).

In summary, RNA methylation indeed seems to steer interactions away from SR-proteins and towards SAFB proteins, which, depending on the substrate, can lead to suppression of splicing or nuclear retention.

- The domain architecture of SAFB proteins is very similar. How are differences in effects, especially in k.d., between SAFBs explained? Is this based on expression levels or non-identical RNA affinities?

Again a very good point raised by the reviewer. We cannot completely exclude non-identical RNA affinities, although their RRMs are quite well-conserved, there are divergent residues that can have an effect on binding affinities. We have observed however that even between three cell lines, HEK293, HeLa and HCT116, there are quite significant differences between the expression levels of these proteins (Extended Data Fig. 6b). Moreover, SAFB proteins also seem to regulate each other's expression levels in a way that depends on their absolute expression levels in a given cell line (Extended Data Fig. 6c,d). For example, single depletion of SAFB1 increases SAFB2 expression in HEK293 cells, and vice versa, without much effect on SLTM, but in HCT116 cells where SAFB2 is not expressed, depletion of SAFB1 increases SLTM expression (Extended Data Fig. 6b-d). We also now provide an RT-qPCR experiment where we carried out single and double depletions, together with the triple-depletion in HEK293 cells (Extended Data Fig. 6e-f). These results show that expression levels of different SAFBs is an important factor in SAFB regulation, but also suggest that differences in affinities might

also be important since SAFB1 seems to be the dominant protein among the three, and is also the one that is evolutionarily most conserved.

- Speaking of that: seeing the apparently clear but short target RNA consensus, how would SAFBs interact with/mask binding sites? Just through their RRM domains, which would require a pretty strong Kd, also to compete out other RBPs with similar preferences e.g. hnRNPG and Tra2-b1 (Moursy et al., NAR, 2014)?

To address this point, and several related points, we carried out rescue experiments by expressing an siRNA-resistant SAFB1 construct in SAFB1, SAFB2 and SLTM depleted cells and analysing the extent of the rescue with the targets that we used to determine the hierarchy of SAFB proteins in HEK293 cells (Extended Data Fig. 6c-e). In this experiment, we also included SAFB1 constructs that lacked domains of interest such as the SAP, RRM, three different deletions targeting the C-terminus (Extended Data Fig. 12). In summary, we could rescue the effect to triple KD by expressing SAFB1, and the extent of the rescue was generally lower in both SAP-deletion, RRM-deletion and C-term deletions, which also depended on the target to some extent (Extended Fig. 12). The one mutant that clearly lost any ability to rescue triple KD was delE which deletes the entire arginine-rich domain at the C-terminus. Since we also carried out biochemical purifications using the same cell lines we used for FLASH, we also determined several high-confidence SAFB-interacting proteins (Extended Data Fig. 11a-d) and checked both the expression of the deletion constructs as well as their ability to co-purify these interaction partners we identified (Extended Data Fig. 11e). These immunoblots clearly show that the most severe SAFB1 mutant, delE, is also the mutant that does not interact with RBMX (hnNRPG), RBM12B and NCOA5 (Extended Data Fig. 11e). These results suggest that a plurality of hnRNP-like proteins likely compete with SR proteins to access the nascent RNA, which is additionally affected by the methylation status of the RNA.

We also would like to note a very recent publication from the Elliott lab that came out as we were working on the revision of this manuscript: Siachisumo et al., 2023 (Siachisumo et al. 2023), which is a continuation of their testis-specific RBMXL2 work, and in a way also an extension of their work on testis-specific functions of TRA2 proteins. They show that RBMX is a specific regulator of "ultra-long" exons, which they define as exons longer than 1kb. This work nicely complements our work (and vice versa) since they show that RBMX inhibits cryptic splicing events inside the "ultra-long" exons they bind, thus protecting the integrity of these coding exons, while we show that at least some of these exons can be spliced-in and out as cassette exons through SAFB regulation. It is of course also extremely curious that there is an apparent convergence towards the testis since RBMX paralogs RBMXL2 and RBMY (which is on the Y chromosome) have similar functions in testis and seemingly can complement RBMX in somatic cells (or at least in cell culture).

- How does masking work? The suggested motifs from FLASH do not fit e.g. the ENCODE-deposited RBnS data (SAFB2) and at least may not be explainable with RRM(only)-mediated binding. The authors should add data that give more insight into the roles of RNP complex formation from the protein side, e.g. using truncations, depletions, etc. The role of homo- and heterooligomeric SAFB for RNA processing e.g. has been discussed recently and may be relevant here. Else, SAFBs could also be recruited to their final binding sites by another RBP.

To avoid repetition we will not discuss the gel-shift assays which gives more credibility to the RRM domain of SAFB1 mediating direct interactions. The biochemical purification of SAFB1, SAFB2 and SLTM, the interaction partners we identified from those experiments, and the rescue experiments coupled immunoblots monitoring the interactions with these factors and SAFB hierarchy experiments suggest that SAFB proteins are not acting alone, and rather associate with proteins of similar functions to push for the intronization of their targets.

We think that SAFBs could still be recruited to some of their final binding sides through other hnRNP and hnRNP-like proteins they interact with, especially since we have recovered hnRNPG (typically recognises AAN, CAA) as a SAFB-interactor (Extended Data Fig. 11) which itself has been suggested to interact with m6A-modified RNA through its C-terminus (Liu et al. 2017), and RBM12B, which has four (q)RRM domains that are closely related to the hnRNPF/H family, which are well-known to interact with G-rich RNA (Dominguez et al. 2010). We think these proteins probably act as a biochemical complex with different binding modules that are likely targeting purine-rich RNAs, however calling them a complex would require more detailed biochemical work which we think is currently beyond the scope of this work.

- As critical also in other dual nucleic acid-binding proteins: the authors should provide a control experiment that undoubtedly excludes a potential influence on the transcriptional level, i.e. DNA-binding. For example, one could test effects of a SAFB with a knocked-out SAP domain. I do not doubt the post-transcriptional data presented here, but the proteins have originally been identified as chromatin regulators. The interplay between DNA- and RNA-binding is not clear and should not be ignored, e.g. in the statement of nuclear retention of TEs by SAFB.

We would not want to exclude the role of the SAP domain in SAFB function, especially since this part is extremely well-conserved through evolution together with the RRM domain.

Whether or not this part of any other part of SAFB proteins is really involved in transcription is hard to prove, since, to our knowledge, previous work never really excluded a post-transcriptional effect. In our own rescue experiments (Extended Data Fig. 12), we can see that deletion of the SAP domain can impede SAFB function, but we cannot at this point claim what exactly is happening one way or the other. Even though we can see how the SAP domain could be helpful to explain the nuclear retention function of SAFB proteins, it is also entirely possible that the role of the SAP domain, which has a very weak but detectable affinity towards DNA, could be to keep SAFB proteins close to chromatin such that they can have quick access to nascent RNA, and may have nothing to do with nuclear retention per se. Since we do not have a good experiment to differentiate these two scenarios, we would like to refrain from making potentially false claims regarding the role of the SAP domain.

We think we provide more direct evidence towards the role of the RRM domain with the gel-shift experiments. In addition, the experiments with ORFeus (Fig. 2i-j) could be interesting to the reviewer since we essentially modify the RNA side of the equation where we remove SAFB binding sites from L1 RNA which liberates the RNA from SAFB regulation, but not from the HUSH complex that seems to suppress most intronless constructs, which we think provides strong evidence for the idea that SAFB-RNA interactions at the core of SAFB-mediated regulation. Whether or not the SAP domain supports these functions, we do not know, but we would not want to exclude the possibility.

- Please comment (or even show) if in general mRNA export is affected in SAFB k.o. systems!

We have not noticed anything striking with pA+ RNA-FISH, which we didn't include in the manuscript. We have in the meantime generated RNA-seq data from nuclear and cytoplasm RNA separately (HEK293 cells, control vs triple KD), and we don't see a general export defect upon SAFB depletion (Fig. 3i).

- Related to the suggested concept of TE somatic suppression vs. activity in testes: can the authors comment on the apparent activity in bone marrow (Fig. 4f)?

Indeed, as also raised in the first comment by referee 1, for some genes we observe a specific shift towards usage of splice acceptors within SAFB peaks in bone marrow, muscle, liver, and to a lesser degree in other tissues. To address this question, we have further scrutinised our data, and decided to update the figure to contain an even more stringent subset of splice junctions, most notably only the junctions utilised in all 27 analysed tissues, rather than 20 or more as before (of note, this is Fig. 5f in the revised manuscript). Tissue-specific splice junction usage and relationships between tissues are generally unchanged. While we provide further insights into the SAFB activity in testis in the revised manuscript (Fig. 4g, Extended Data Fig. 18), at this point we can only report the SAFB-related isoforms in bone marrow and other tissues as-is, and hypothesise about the cause. In principle, isoforms induced by SAFB loss are within the range of possible pre-mRNA processing outcomes in all tissues. They are strongly suppressed by SAFB binding, but they could in some genes also be made more likely to be generated by other factors, such as RNAPolIII speed or differential RNA methylation. We have found SAFB proteins to be expressed at comparable (bulk) levels in all tissues, but effects on splicing could also be driven by niche-, cell- or timepoint-specific SAFB expression levels, as shown in our extended analysis in testis.

- Finally, I do not fully understand the possibility of SAFBs to selectively regulate giant protein-coding cassette exons. How exactly could they themselves recognize the size of exons and what differentiates them from canonical SR-antagonists like hnRNPs? Please comment or add additional data as this would valuably contribute to answering a central question of selective splicing suppression organized by RBPs. Do SAFBs interact with the splicing machinery in some way?

We were also not quite sure how giant exon alternative splicing could mechanistically work, since how can any protein measure the length of RNA? (Ago family do it, but the scales are completely different). We discussed how this could work in the gel-shift section: essentially the length seems to be measured by the time that it takes to transcribe the RNA by RNA polymerase II which determines the extent of m6A modification, which then determines which RBPs can interact with the RNA and which cannot. There is certainly more work to be done, especially regarding the extent of this regulation and perhaps other RNA-modifications pushing interaction to one way or another (or "steer" as the reviewer refers to it). For the last point, we could not detect a significant interaction with the splicing machinery by neither mass-spectrometry nor with immunoblotting a few key targets (Extended Data Fig. 11). If there is an interaction, we cannot detect it with our methods, but perhaps proximity-labelling methods such as BioID can identify such interactions.

- I strongly suggest to include a final model figure/panel that summarizes the findings.

We have now included a model figure in Extended Data Fig. 18.

Regarding methods, their description seems comprehensive, but some essentials are missing.

- As a non-expert in this, I miss data on the control over SAFB expression levels: are the transfected mammalian cells expected to express protein on the endogenous-like level? Is there SAFB endogenous background (which is to get rid of in the siRNA treatments)? Please provide experimental data, i.e. WB!

Upon revision we added western blot data addressing these points by the reviewer. The comparison of endogenous and transgene expression levels is presented in Extended Data Fig.11c, where the protein amount of SAFB1 and SAFB2 transgenes are very close to endogenous levels and SLTM is over-expressed. In addition, effectiveness of the siRNA is presented in Extended Data Fig.6c. FLASH experiments were always carried out with the endogenous proteins present to be consistent between all the experiments.

- For Flash: I do not fully understand: are all RBPs equipped with Biotin acceptor peptide affinity tag to parallelize the procedure? This is not explained, or at least is not sufficiently clear to me.

We apologise for not clarifying this in the first submission. The reviewer is right about the biotin acceptor peptide, it is always the same tag for all the proteins, which are purified the same way. In the revised manuscript the information is provided in the Methods section as:

“All transgenes were cloned with an N-terminal His₆-biotinylation sequence-His₆ tandem (HBH) tag that allows rapid and ultra-clean purifications without the use of antibodies. We also added a 3xFLAG tag right before the HBH tag to increase the versatility of the construct, which we refer to as the 3FHBH tag”

- It may make sense to comment on possible relayed effects of SAFB-k.d.-related changes in gene expression, i.e. indirectly through the lack of affected proteins to regulate others' expression, stabilities, etc (also on the transcript level).

We had the chance to look deeper into this by looking at the expression of SAFB proteins in different cell lines (Extended Data Fig. 6b) where we noticed that SAFB2 is not expressed in HCT116 cells. This we have also verified by depleting SAFB1 or SAFB1 and SAFB2 in HeLa cells (Extended Data Fig. 6d, bottom) or depleting SAFB1 in HCT116 cells (Extended Data Fig. 6d, top) and using the monoclonal antibody called HET which recognises both SAFB1 and SAFB2 (this is to make sure that the lack of SAFB2 signal in HCT116 is not some antibody artefact). And it can be seen that in HeLa cells, HET signal diminishes only when SAFB1 and SAFB2 is depleted at the same time, while in HCT116 cells HET just follows SAFB1 signal.

While doing these experiments, and also checking single, double and triple depletions (Extended Data Fig. 6c), we also noticed the feedback loop between SAFB proteins, which we

think the reviewer wanted us to explore further. Essentially, in HEK293 cells depletion of SAFB1 results in an increase in SAFB2, and depletion of SAFB2 increases SAFB1 expression, but neither effect SLTM, nor SLTM effects SAFB1 or SAFB2. In HCT116 cells, where SAFB2 is absent, SAFB1 depletion increases SLTM levels.

We see these effects at the RNA level as well, but the magnitude of the effect is somewhat low: In SAFB1 KD, SAFB2 is up 0.77 (log2fold-change), SLTM is up 0.25 (log2fold-change); In SAFB2 KD SAFB1 is up 0.31 (log2fold-change), SLTM does not change; In SLTM KD SAFB1 does not change, SAFB2 is up by 0.13 (log2fold-change).

Finally, some minor things:

- In many panels of the current figure versions (esp. the supplement) the axis labelling is very small and hard to read, which is complicated for the non-expert reader less familiar with the type of data presentation.

We have changed the small font size and hope that everything is clearly visible.

- Statements of correlation in lines 130-132 are to my understanding not shown as a figure panel.

We extended the description of chi-square test presented in the main text, and added a reference to the supplementary table with detailed counts of genes in different categories, at two log2-fold-change (LFC) cutoffs (Supplementary table 15). Please note this analysis is also discussed in response to Reviewer#4, point 5.

- Line 129: ...compared SAFB-bound regions (remove "of").

Fixed.

- Line 142: new terminal exon? (not "now").

We thank the reviewer for spotting these mistakes. We have corrected them in the revised manuscript.

- Domain color coding in Ext. Data Fig. 1a is not very helpful using very similar types of blue and violet. As a side note, SR proteins usually contain a pseudo RRM as the second one if in tandem.

We have added a note to the figure legend indicating that some of the RRM domains could be qRRMs. We did not want to restrict it to SR proteins because we realised that the RRM domains of hnRNPF are actually qRRMs as well. We would be happy to label them more accurately as RRM and qRRM but we could not find a good resource for this.

Referee #4:

Transposable elements (TEs) form a major component of the human genome that, when active, can be expressed and inserted within protein-coding genes, thereby in principle affecting host gene expression. However, it is well known that most intronic transposable elements (TEs) are never exonized and are simply spliced out, suggesting that cellular factors can a) discriminate TE- from “host”-derived splice sites, and b) suppress activity of the former. However, this is not well understood on the mechanistic level.

In their article “Autonomous transposons tune their sequences to ensure somatic suppression“, Ilik et al. follow up on this observation and set out to identify novel factors that regulate splicing of intronic TEs in the human genome. Exploiting their own CLIP-like FLASH protocol, they generated and thoroughly analyzed RNA-protein interaction datasets for in total 33 splicing regulators, such as SR/SR-like and HnRNP proteins. This led to the identification of the SAFB family, comprising SAFB1, SAFB2 and SLTM, as specifically enriched at sense L1 and Tigger DNA transposon transcripts. Intriguingly, knockdown of all SAFB members enhanced L1 retrotransposition in a luciferase-based reporter assay, suggesting that this binding is functionally relevant. To investigate the role of SAFB proteins on TE (and general gene) expression on a global level, the authors compared RNA-seq profiles in two SAFB-proficient and -knockdown cell lines and observed upregulation of L1 and Tigger elements upon SAFB loss. Mechanistically, this upregulation was mainly due to the activation of cryptic splice sites embedded within or in close proximity to the 5' TE region, which resulted in the exonization of intronic TEs and activated cryptic polyadenylation sites at the 3' end of the TE. Consequently, this led to reduced expression/coverage of coding exons downstream of this novel splice site. Interestingly, this exonization phenomenon was not restricted to TEs, but was also found for sense pseudogenes and giant protein-coding cassette exons, all of which turned out to be relatively long compared to conventional exons and were enriched for adenosine-based, purine-rich GAAGAA sequences, the prototypic SAFB binding motif. Moreover, this mechanism appeared to be conserved in mice and even in flies, possibly indicating that suppression of TEs and other purine-rich motifs constitutes the evolutionary original function of SAFB proteins.

Hypothesizing about the cellular and even organismal role of this SAFB-mediated suppression, the authors in the end speculate that TE activation may be some type of stress response to allow for better adaptations under strenuous conditions. In fact, 90 min of heat-shock – which sequesters SAFB proteins into stress bodies and thus potentially depletes them for other functions – activates alternative splicing events and TE upregulation that largely recapitulate the SAFB knockdown phenotype. In addition, the authors provide evidence that SAFB expression seems to be rather low in post-meiotic spermatids, and this likewise leads to upregulation of similar splicing events as upon SAFB KD. From an evolutionary point of view, this SAFB low condition may allow TE activity in a short window that allows inheritance of all changes to the next generation, but at the same time does not affect the host.

Together, this description of SAFB proteins as evolutionary conserved suppressors of TEs and other TE-like elements such as pseudogenes and giant protein-coding exons is very interesting and overall well supported by the presented data. However, I would like to ask for a few

additional experiments to further support this novel role of SAFB proteins, and to also provide more mechanistic insight into how SAFB proteins perform this task.

1. What is the individual role of the different SAFB protein domains the observed effects? Is TE binding and suppression (both of exonization and retrotransposition) mediated only by the RNA binding domain, or are other functionally relevant proteins recruited into the complex via additional domains? Can the SAFB KD effect be rescued by reconstitution of the RNA domain alone?

We looked for an answer to this question by introducing either an siRNA-resistant full-length SAFB1, or SAFB1 with various deletions targeting its SAP domain, RRM domain or sections of its low-complexity C-terminal domain into cells that are depleted of SAFB proteins. We judged how these constructs complement splicing defects introduced by the triple-KD by RT-qPCR (Extended Data Fig. 12). The results show essentially that each domain is important for SAFB function, which is perhaps expected since these are the regions that are evolutionarily conserved. The most debilitating deletion was the one that removed a large part of the arginine-rich C-terminal domain, and this exact deletion was also the one that completely lost all interactions with the other proteins that we identified as SAFB-interactors (new data, IP-MS, Supplementary Table 3 and Extended Data Fig. 11). These results suggest that, just like SR-proteins that collectively and by interacting with each other on a stretch of RNA promote exonization of their targets, groups and sub-groups of hnRNP and hnRNP-like proteins do the same to promote intronization of their targets. Our new results with methylated vs non-methylated RNA engaging in differential interactions with TRA2B-RRM and SAFB1-RRM (new data, Fig. 4b) also suggest that RNA modifications can also push this equilibrium into one direction or the other.

2. How is expression of giant protein-coding exons regulated in normal tissues? Are the respective cells, in case of ANK2 and -3, MAP2 and MAPT for example of the nervous system, specifically depleted of SAFB proteins? Can giant protein-coding exon expression therein be suppressed by exogenous SAFB protein?

Since we can force the inclusion of these exons in a variety of completely unrelated cell lines (HEK293, HeLa, HCT116, mouse 3T3 fibroblasts) upon SAFB depletion, we reason that SAFB proteins are responsible for suppressing "accidental" expression of these large isoforms in tissues where they should not be expressed. The reviewer is right in asking how this repression is lifted then in the specialised tissues where they are and they must be expressed. We think that at least in testis, this is achieved by programmed downregulation of SAFB1/2 during spermatogenesis, which we now verified by staining Safb1 in cryo-sections of a P50 mouse testis (Fig. 5g and Extended Data Fig. 18a). We have not found evidence of a similar downregulation of SAFB proteins in neuronal tissue. New single, double-depletions and immunoblots showing SAFB protein levels in different cell lines (Extended Data Fig. 6), suggest that RNA-seq may not be completely reliable in inferring actual SAFB protein levels in cells, however lab of James Uney has carried out SAFB stainings in mouse brains and do report decent staining throughout the CNS (Norman et al. 2016), although they do report cytoplasmic SAFB2 staining, which we also observed in cell culture (data not shown). In their more recent work with human brain material, they do report differential SAFB expression, but mostly in the context of pathological conditions (Buckner et al. 2020). The results of the heat-shock experiment, the fact that SAFB proteins can be localised to the cytoplasm in certain cases,

suggest possible temporal control over these giant exon splicing through removing SAFB proteins away from nascent RNA. However, at this point in time, we do not have a definite answer. We did however try to naively overexpress SAFB1 in mouse N2A cells, which express detectable amounts of the giant exon of ANK3 as well as CLIP1. As a control, we overexpressed Cas9, we have not scored a specific reduction in ANK3's giant exon splicing, but the splicing of the giant exon of CLIP1 was significantly reduced in SAFB1 over expressing cells (Extended Data Fig. 18b). Overexpression of SAFB1 also reduced the endogenous expression of SAFB proteins, consistent with the new immunoblotting data we provided in the revised version (Extended Data Fig. 6) which might indicate that CLIP1 is simply more sensitive to SAFB levels, or activators of the ANK3's giant exon splicing are more dominating in N2A cells.

3. Is there any evidence for high activity of TE retrotransposition in post-meiotic spermatids? If so, can this be counteracted by SAFB proteins?

The very short answer to the first part, to our knowledge, is no. But the answer would also be "no" if the questions was directed at spermatocytes, spermatogonia, PGCs and so on, because retrotransposition is normally a very rare event, however recent scRNA-seq experiments show a peak of LINE expression coinciding with spermatids (<https://www.nature.com/articles/ng1022z/figures/3>). Researchers also typically report "TE activity" rather than "retrotransposition" since TE activity can be measured by expression of ORF1p and/or L1 RNA, but formally proving retrotransposition is quite difficult since it is a "needle in a haystack" problem. There are methods to specifically pull-down genomic loci with oligo targeting 5'- and/or 3'-ends of TE sequences, but this does not eliminate > 1 million TE insertion one must keep on sequencing. Applying these to testis, in such a way to unequivocally distinguish the origin of the retrotransposition event (could happen in ES cells, PGCs, Spermatogonia, or Spermatocytes before spermatids) is extremely difficult, and we do not see a feasible way to do it yet. One way around would be to use ORFeus, which is a hyperactive L1 which is not A-rich anymore since its coding sequence has been recoded, but ORFeus is not controlled by SAFB as we show with additional data (Fig. 2i,j), therefore we cannot use it to understand the role of SAFB in a natural context.

Although we cannot answer the reviewer's question directly, we decided to at least do some ground work to see if it can be feasible in the future, and tested a related question. One important precondition for SAFB to act on L1 RNA is that L1 is expressed, which is normally strongly repressed by a plethora of epigenetic factors. In order to first verify that SAFB is expressed in mouse testis, in a way predicted from the RNA-seq data, we carried out an IF staining on the testis of a 7-week, wild-type male mouse to capture a wide variety of stages in the seminiferous tubule. These stainings show that SAFB1 is expressed in Spermatogonia, its expression peaks at Spermatocytes, then drops in round spermatids and essentially becomes undetectable at the later stages (Fig. 5g, Extended Data Fig. 18a), coinciding with the splicing of the giant exon of CLIP1 (Akhmanova et al. 2005). These tissues were also co-stained with ORF1p as an indicator of TE activity, in line with expectations, no specific signal could be detected. We then used Dnmt3 KO mice, which is reported to specifically induce expression of younger L1 elements in mice testes. In this tissue, we could score a very robust ORF1p expression, which appears mutually exclusive with SAFB1 staining (Fig. 5h, Extended Data Fig. 18a). These experiments support the idea that L1 retrotransposition is suppressed by

SAFB proteins in the testis. We hope to create tools that can be used to test these models more explicitly in the future.

4. For the majority of the presented data, the authors show mapped IGV snapshots for SAFB binding and RNA-seq coverage to demonstrate changes in downstream exon expression and splicing. I think it would be beneficial to confirm these findings by qPCR assays, also in order to get a better quantitative idea of the implied alterations.

We have now provided RT-qPCR analysis of several splicing events within the context of all single and double-depletion conditions in addition to triple KD (Extended Data Fig. 6f).

5. The authors provide RNA-seq data of SAFB knockdown cells (HEK293 and HeLa) to demonstrate their TE-suppressing function in normal somatic tissues. Not surprising given the described role of SAFB proteins in diverse RNA-centric processes, SAFB KD results in prominent changes in gene expression, in particular in the triple KD. Can the authors comment on how many genes become up- and downregulated in the respective experiments (Ext. Data Fig. 6a-g), and what percentage of these genes contain intronic SAFB target elements such as TEs, pseudogenes or giant exons? Vice versa, how many genes that are expressed beyond a certain threshold in 293 and HeLa cells do not become deregulated despite being enriched for SAFB binding and TEs? Can the authors also please provide data for the extent of the knockdown, e.g. a western blot?

We now included western blots showing the extent of knock-down in new Extended Data Fig.6.

Based on RNA-seq and DESeq2 analysis, all siRNA-targeted transcripts in HEK293 and HeLa cells were below 25% of their levels measured in the control libraries, with some (e.g. SAFB1 in triple knockdowns) closer to 15%. In HCT116 double knockdown, SAFB1 was around 20%, and SLTM slightly above 30% of their control levels. The results are visualised in ED Fig.7h. To define a gene as differentially expressed, we used the adjusted P-value cutoff of 0.05, and the log2FC threshold of 1. Supplementary Table 15 summarises this information (also shown here):

cell_line	siRNA target	DE genes (padj < 0.05, lfc > 1)				nonDE genes		chi-squared	df	pval	odds ratio
		UP	UP w/ SAFB peaks	DOWN	DOWN w/ SAFB peaks	total	w/ SAFB peaks				
HEK293	SAFB1	3	1	1	1	27156	8705				
	SAFB2	0	0	0	0	27160	8707				
	SLTM	0	0	0	0	27160	8707				
	SAFB1+SAFB2+SLTM	88	28	65	42	27005	8635	12.632	1	0.00038	1.79
HeLa	SAFB1+SAFB2+SLTM	12	3	20	17	17537	7495	4.321	1	0.03765	2.23
HCT116	SAFB1+SLTM	63	16	54	38	23183	8063	6.143	1	0.0132	1.58

When all SAFB proteins expressed in a given cell line are knocked down together, genes bound by SAFB are highly enriched among the deregulated genes. This is reported in the manuscript for HEK293, and was recapitulated in HeLa and HCT116 cells. This result suggests that the majority of differentially expressed genes are direct SAFB targets, although of course we cannot exclude (and should expect) indirect effects as well. We repeated the same analysis at a lower LFC threshold (second worksheet in Supplementary Table 15), and observed even stronger enrichment (in terms of odds ratio) of SAFB targets in DE genes, even in single-

protein KDs. This suggests that many genes with mildly affected expression at the gene level in SAFB KD are also regulated by SAFB.

Of note, total numbers of differentially expressed genes reported by DESeq2 can depend on variability between experimental replicates, some of which is certainly technical, so higher number of DE genes does not necessarily imply stronger effect of SAFB loss. Moreover, as the effects of SAFB loss are post-transcriptional, differential gene expression analysis based on short reads counted on annotated gene models has limited detection power. For example, if a SAFB peak is spliced in and causes premature transcription termination in one of the last introns of a very long gene, total read count across the exons of that gene will not be markedly different between KD and control, and DESeq2 will not report such gene as differentially expressed. This was our motivation for the analysis presented in Figure 3c.

Reviewer Reports on the First Revision:

Referees' comments:

Referee #1:

The authors have done a good job addressing my comments and concerns.

Referee #2:

I appreciate the clarifications and additional data provided by the authors. They have largely addressed one of my major concerns, regarding interpretation of the RNA-seq analysis (although there are no methods included that I could see for how the nuclear vs cytoplasmic RNA-seq was done - please fix this). Each of my minor concerns have also been addressed, and it is good to see the HUSH literature now being included and considered.

However, the major concern I noted around the presented evidence that SAFB proteins prevent L1 retrotransposition remains.

1) In their response, the authors acknowledged that the biological replicates claimed for Fig. 2g were actually technical replicates. In other words, this experiment was done once. There is a headline claim in the abstract that SAFB proteins prevent retrotransposition. It would be reasonable to perform the L1 reporter assay two more times to be confident of this result and present data points representing biological triplicates, or remove the claim. This is not an onerous request given the simplicity of the assay. The main text and figure legend should also note that this L1 is pCAG-driven as that could influence its potential regulation by SAFB proteins. The reference the authors requested, with respect to the maximal impact of L1 RNA abundance on retrotransposition efficiency is PMID: 21320307, Fig. 2.

2) A straightforward control experiment for the RNA-FISH experiment presented in Fig. 2h was requested and not done. This control would involve a human L1 plasmid expressed in mouse cells (where there is no background L1Hs to worry about) to test the RNA FISH probe. If there is a technical explanation for why the revision experiment did not work please provide it. The Northern blot shown instead now in Fig. 2k is from a human cell line, using an L1 plasmid that isn't clearly identified, and the 28S signal is clearly higher in lane 8 than the other lanes. I don't think this Northern actually adds anything, although I appreciate how much work it would have been to prepare. The result looks inconclusive. It would be better to do the requested control experiment.

3) I admire the comparison shown in Fig. 2i between the GC-rich ORFeus and an AT-rich wild-type L1. Clear difference between lanes 6 and 7. The methods do not say which wild-type L1 was used and whether the two plasmids involved L1 being driven by the same promoter, which could also impact SAFB regulation. There are major caveats to comparing the two L1s, not least of which because ORFeus has a much higher baseline expression (as shown). It is necessary to clarify these methods details (as elsewhere in the manuscript) before being able to fully interpret the result, despite its

conceptual appeal.

Geoff Faulkner (University of Queensland)

Referee #3:

In their revised version of the manuscript, Ilik et al. have undertaken great efforts to add valuable data and close some of the earlier gaps in their study. This refers both to my comments raised and also to comments raised by the other reviewers as far as can judge it.

I highly appreciate the novel findings of SAFBs interacting with “opponents” of SR-proteins, while apparently competing with the very ones via the distinctive ability to interact with m6A-modified RNA. Mapping the interaction to a significantly confined region in the SAFB1 C-term is interesting and something that has been suggested earlier but never been shown with such a functional impact. Of note, this region appears somewhat different in SAFBs and may be a fine-tuning element.

Regarding the potential to recognize m6A-RNA (whilst per se the RRM shows relatively weak RNA binding), I may be allowed to add that potential homo- and hetero-oligomerization of SAFBs will possibly increase the effect in vivo, if so, as it may enrich the local concentration of SAFB RRMs.

Altogether, I feel the inclusion of new data justifies suggesting the manuscript for publication in this journal, given the following things will be added:

1) EMSAs of SAFB vs. TRA2B RRMs are convincing, but at least one replicate should be included. If EMSAs have been run in replicates, this should be stated.

2) Further, as authors say themselves, one needs to treat this observation with care with respect to the situation in vivo, i.e. full-length proteins. At least SR proteins are known to exploit more than their RRM, and for SAFBs this is still not fully understood, but possible.

Could the authors use a simple pulldown using immobilized native and methylated RNA (like in the EMSA) to detect levels of TRA2b and SAFB(s) from full lysates? This will strengthen their conclusions even more and reveal that native fl-proteins incl. their possible PTMs would still differentially bind m6A-modified RNA as a proof of principle. I do see the challenge in trying to isolate individual RBPs from complex RNPs such that the findings may be less black-white than in EMSAs with purified domains. Nonetheless, I suggest including this experiment.

Also, not all SR proteins have purine-rich motifs as their consensus target sequences; or they can bind to combined motifs via tandem RRMs (e.g. SRSF1). This does not per se weaken the findings and hypotheses, but may make aware of the need to distinguish SRSFs, as to my knowledge none of them has been tested in direct interaction with methylated RNAs so far. Just as a comment.

3) As minor edits I suggest:

- Ext. Data Fig. 1: Please double-check all domains! E.g. SRSF7 contains a ZnF domain (for which RNA binding is debated) and also U2AF1 should have one. I assume, all potential RNA-binding relevant domains (thus no SAPs?) are supposed to be listed as it is about domain architectures.

- Ext. Data Fig. 6a: Please define errors (in figure legend)! In panel f, the statistical background of the depicted should also be added (likely median of the triplicate?)!

- For the IPs and WBs in Ext. Data Fig. 11, I assume no replicates have been made. This should be stated at least, while full views of the respective Blots will have to be included.

- Ext. Data Fig. 12 (left): Please label y-axis!

- Ext. Data Fig. 18b: Please define error bars in the legend!

- I suggest working over the sentence ranging from lines 275-279, it reads very complicated to follow!

Referee #4:

The authors have done an excellent job in addressing each of the points and questions that were raised - also by the other reviewers - in the original submission. They have now clarified a number of important issues and have added substantial new data (in particular the comparison of the SAFB-mediated regulation of TEs with that mediated by the HUSH complex, the identification of SAFB interaction partners and the surprising insensitivity of SAFB1 to N6-methylated RNA (in contrast to TRA2B)) that provide mechanistic insight and further support their line of argumentation. Overall very responsive and convincing.

Response to Referees

We thank all the reviewers for their comments and constructive criticism. As the reviewers #1 and #4 had no more questions or concerns on our revised manuscript, we addressed all the major and minor points of the reviewers #2 and #3. We used blue text to indicate new text in the article file and our responses here.

Point-by-Point Response

Referee #2:

I appreciate the clarifications and additional data provided by the authors. They have largely addressed one of my major concerns, regarding interpretation of the RNA-seq analysis (although there are no methods included that I could see for how the nuclear vs cytoplasmic RNA-seq was done - please fix this). Each of my minor concerns have also been addressed, and it is good to see the HUSH literature now being included and considered.

We apologize for the omission of these technical details from Methods, which are now added.

However, the major concern I noted around the presented evidence that SAFB proteins prevent L1 retrotransposition remains.

1) In their response, the authors acknowledged that the biological replicates claimed for Fig. 2g were actually technical replicates. In other words, this experiment was done once. There is a headline claim in the abstract that SAFB proteins prevent retrotransposition. It would be reasonable to perform the L1 reporter assay two more times to be confident of this result and present data points representing biological triplicates, or remove the claim. This is not an onerous request given the simplicity of the assay. The main text and figure legend should also note that this L1 is pCAG-driven as that could influence its potential regulation by SAFB proteins. The reference the authors requested, with respect to the maximal impact of L1 RNA abundance on retrotransposition efficiency is PMID: 21320307, Fig. 2.

We thank the reviewer for this comment. We have now repeated this experiment and present the results as a combined figure in Fig. 2e. We have also added the statement that L1 is pCAG-driven both to the text and the figure legend, as requested.

2) A straightforward control experiment for the RNA-FISH experiment presented in Fig. 2h was requested and not done. This control would involve a human L1 plasmid expressed in mouse cells (where there is no background L1Hs to worry about) to test the RNA FISH probe. If there is a technical explanation for why the revision experiment did not work please provide it. The Northern blot shown instead now in Fig. 2k is from a human cell line, using an L1 plasmid that isn't clearly identified, and the 28S signal is clearly higher in lane 8 than the other lanes. I don't think this Northern actually adds anything, although I appreciate how

much work it would have been to prepare. The result looks inconclusive. It would be better to do the requested control experiment.

We indeed had some simple technical issues with this experiment, mostly with respect to transfection efficiencies in mouse cell lines –we could see that the L1 FISH probes against human L1 ORF2 did not show any staining in mouse cells, suggesting that they are specific, but we could not unambiguously identify cells expressing human L1 RNA to be completely sure of the results. To overcome these problems, we cloned a construct that expresses GFP and human L1-ORF2 from the same bi-directional promoter, enabling us to find mouse cells that express the human L1 RNA. We have done the FISH experiment as requested, and present it in Extended Data Fig. 4g. As can be seen, the FISH probes are specific to human L1 RNA.

The plasmid that drives L1 expression is expressing L1_{RP}, which is the same L1 used in the retrotransposition assays.

3) I admire the comparison shown in Fig. 2i between the GC-rich ORFeus and an AT-rich wild-type L1. Clear difference between lanes 6 and 7. The methods do not say which wild-type L1 was used and whether the two plasmids involved L1 being driven by the same promoter, which could also impact SAFB regulation. There are major caveats to comparing the two L1s, not least of which because ORFeus has a much higher baseline expression (as shown). It is necessary to clarify these methods details (as elsewhere in the manuscript) before being able to fully interpret the result, despite its conceptual appeal.

We again apologize for omitting the details, which we agree are important to interpret the experiment. Indeed the constructs are driven by the same promoter and are exactly the same overall except the coding regions ORF1 and ORF2. The wild-type L1 used is the same sequence used for retrotransposition assays (L1_{RP}), to ensure comparability. We added the full sequence of these plasmids to Supplementary Table 2 “Cloning_other_oligos”, under the tab L1vsORFeus_plasmids with annotations pointing to the important elements in the plasmid.

Referee #3:

In their revised version of the manuscript, Ilik et al. have undertaken great efforts to add valuable data and close some of the earlier gaps in their study. This refers both to my comments raised and also to comments raised by the other reviewers as far as can judge it.

I highly appreciate the novel findings of SAFBs interacting with “opponents” of SR-proteins, while apparently competing with the very ones via the distinctive ability to interact with m6A-modified RNA. Mapping the interaction to a significantly confined region in the SAFB1 C-term is interesting and something that has been suggested earlier but never been shown with such a functional impact. Of note, this region appears somewhat different in SAFBs and may be a fine-tuning element.

Regarding the potential to recognize m6A-RNA (whilst per se the RRM shows relatively weak RNA binding), I may be allowed to add that potential homo- and

hetero-oligomerization of SAFBs will possibly increase the effect in vivo, if so, as it may enrich the local concentration of SAFB RRM.

Altogether, I feel the inclusion of new data justifies suggesting the manuscript for publication in this journal, given the following things will be added:

1) EMSAs of SAFB vs. TRA2B RRM are convincing, but at least one replicate should be included. If EMSAs have been run in replicates, this should be stated.

We had several versions of these gels with each protein on its own gel (Halo alone, Halo-SAFB1-RRM and Halo-TRA2B-RRM) while setting up the experiment, thus confident that the results are robust and reproducible. We have however defrozed the reagents and repeated the experiment with the gel set up (see below), and the results look very reproducible. We now added the statement to the Figure 4 legend saying: "Gel images shown here are representative of two replicates."

Figure 1 Repeat EMSA experiment gel images same conditions as in Fig. 4b

2) Further, as authors say themselves, one needs to treat this observation with care with respect to the situation in vivo, i.e. full-length proteins. At least SR proteins are known to exploit more than their RRM, and for SAFBs this is still not fully understood, but possible.

Could the authors use a simple pulldown using immobilized native and methylated RNA (like in the EMSA) to detect levels of TRA2b and SAFB(s) from full lysates? This will strengthen their conclusions even more and reveal that native fl-proteins incl. their possible PTMs would still differentially bind m6A-modified RNA as a proof of principle. I do see the challenge in trying to isolate individual RBPs from complex RNPs such that the findings may be less black-white than in EMSAs with purified domains. Nonetheless, I suggest including this experiment.

Also, not all SR proteins have purine-rich motifs as their consensus target sequences; or they can bind to combined motifs via tandem RRMs (e.g. SRSF1). This does not per se weaken the findings and hypotheses, but may make aware of the need to distinguish SRSFs, as to my knowledge none of them has been tested in direct interaction with methylated RNAs so far. Just as a comment.

We thank the reviewer for this comment, and the suggestions. We tried this experiment by using a biotinylated L1 fragment (biotinylated at the 3'-end with a single biotin group) either unmethylated or N6-methylated which we have introduced through in vitro transcription, and used a nuclear extract prepared from HCT116 cells as the lysate. We could detect a remarkable enrichment of SAFB1

on methylated RNA, together with its interaction partners, as well as YTHDC1, which decided to include as a positive control, but we couldn't detect SLTM binding to either methylated or unmethylated RNA, perhaps due to its low expression levels. On the SR-proteins side, we could detect TRA2b enriched on the unmethylated RNA, though it is detectable on the methylated RNA as well, perhaps through SR-SR interactions with other SR-proteins. We could observe for example an enrichment of SRSF3 on the methylated RNA. SRSF1 seemed quite specific to unmethylated RNA, and we could see a single band on 1H4 blots at around 37kDa that appear also very specific to unmethylated RNA, though this could just be SRSF1. The results of this experiment are shown in Extended Data Fig. 7g.

As the reviewer alluded to, this experiment is not as black-and-white as the EMSAs since it includes many proteins, at unknown concentrations containing various PTMs that affect their interactions with the RNA and with each other, but we agree that it provides quite valuable additional information, and we thank again for this suggestion.

3) As minor edits I suggest:

- Ext. Data Fig. 1: Please double-check all domains! E.g. SRSF7 contains a ZnF domain (for which RNA binding is debated) and also U2AF1 should have one. I assume, all potential RNA-binding relevant domains (thus no SAPs?) are supposed to be listed as it is about domain architectures.

All checked again and updated. There were indeed several ZnF domains missing, including SRSF7 and U2AF1, we thank the reviewer for these suggestions. We tried to list all relevant RNA-binding domains, but some IDRs are still left out, because it is already a quite crowded figure with too many colors.

- Ext. Data Fig. 6a: Please define errors (in figure legend)! In panel f, the statistical background of the depicted should also be added (likely median of the triplicate?)!

Legend is updated (Extended Data Fig. 4a in this submission).

- For the IPs and WBs in Ext. Data Fig. 11, I assume no replicates have been made. This should be stated at least, while full views of the respective Blots will have to be included.

These were done twice, with slightly different set-ups (for example, just "mock" transfection instead of GFP as a control in panel e). Full blots are provided in the Supplementary Information with this submission.

- Ext. Data Fig. 12 (left): Please label y-axis!

Added (Extended Data Fig. 7f in this submission).

- Ext. Data Fig. 18b: Please define error bars in the legend!
Explanation is added to the legend (Extended Data Fig. 10h in this submission).

- I suggest working over the sentence ranging from lines 275-279, it reads very complicated to follow!

We removed the last part of the sentence to simplify it, we thank the reviewer for the suggestion.

Reviewer Reports on the Second Revision:

Referees' comments:

Referee #2:

The authors have addressed my remaining comments, thank you.

Geoff Faulkner (University of Queensland)

Referee #3:

In this revised version, the authors have taken care of the mentioned issues from before to my satisfaction.

I do now recommend the manuscript for publication.

Author Rebuttals to Second Revision:

Response to Referees

We thank all the reviewers for their comments and constructive criticism. As the reviewers #1 and #4 had no more questions or concerns on our revised manuscript, we addressed all the major and minor points of the reviewers #2 and #3. We used blue text to indicate new text in the article file and our responses here.

Point-by-Point Response

Referee #2:

I appreciate the clarifications and additional data provided by the authors. They have largely addressed one of my major concerns, regarding interpretation of the RNA-seq analysis (although there are no methods included that I could see for how the nuclear vs cytoplasmic RNA-seq was done - please fix this). Each of my minor concerns have also been addressed, and it is good to see the HUSH literature now being included and considered.

We apologize for the omission of these technical details from Methods, which are now added.

However, the major concern I noted around the presented evidence that SAFB proteins prevent L1 retrotransposition remains.

1) In their response, the authors acknowledged that the biological replicates claimed for Fig. 2g were actually technical replicates. In other words, this experiment was done once. There is a headline claim in the abstract that SAFB proteins prevent retrotransposition. It would be reasonable to perform the L1 reporter assay two more times to be confident of this result and present data points representing biological triplicates, or remove the claim. This is not an onerous request given the simplicity of the assay. The main text and figure legend should also note that this L1 is pCAG-driven as that could influence its potential regulation by SAFB proteins. The reference the authors requested, with respect to the maximal impact of L1 RNA abundance on retrotransposition efficiency is PMID: 21320307, Fig. 2.

We thank the reviewer for this comment. We have now repeated this experiment and present the results as a combined figure in Fig. 2e. We have also added the statement that L1 is pCAG-driven both to the text and the figure legend, as requested.

2) A straightforward control experiment for the RNA-FISH experiment presented in Fig. 2h was requested and not done. This control would involve a human L1 plasmid expressed in mouse cells (where there is no background L1s to worry about) to test the RNA FISH probe. If there is a technical explanation for why the revision experiment did not work please provide it. The Northern blot shown instead now in Fig. 2k is from a human cell line, using an L1 plasmid that isn't clearly identified, and the 28S signal is clearly higher in lane 8 than the other lanes. I don't think this Northern actually adds anything, although I appreciate how much work it would have been to prepare. The result looks inconclusive. It would be better to do the requested control experiment.

We indeed had some simple technical issues with this experiment, mostly with respect to transfection efficiencies in mouse cell lines –we could see that the L1 FISH probes against human L1 ORF2 did not show any staining in mouse cells, suggesting that they are specific, but we could not unambiguously identify cells expressing human L1 RNA to be completely sure of the results. To overcome these problems, we cloned a construct that expresses GFP and human L1-ORF2 from the same bi-directional promoter, enabling us to find mouse cells that express the human L1 RNA. We have done the FISH experiment as requested, and present it in Extended Data Fig. 4g. As can be seen, the FISH probes are specific to human L1 RNA.

The plasmid that drives L1 expression is expressing L1_{RP}, which is the same L1 used in the retrotransposition assays.

3) I admire the comparison shown in Fig. 2i between the GC-rich ORFeus and an AT-rich wild-type L1. Clear difference between lanes 6 and 7. The methods do not say which wild-type L1 was used and whether the two plasmids involved L1 being driven by the same promoter, which could also impact SAFB regulation. There are major caveats to comparing the two L1s, not least of which because ORFeus has a much higher baseline expression (as shown). It is necessary to clarify these methods details (as elsewhere in the manuscript) before being able to fully interpret the result, despite its conceptual appeal.

We again apologize for omitting the details, which we agree are important to interpret the experiment. Indeed the constructs are driven by the same promoter and are exactly the same overall except the coding regions ORF1 and ORF2. The wild-type L1 used is the same sequence used for retrotransposition assays (L1_{RP}), to ensure comparability. We added the full sequence of these plasmids to Supplementary Table 2 “Cloning_other_oligos”, under the tab L1vsORFeus_plasmids with annotations pointing to the important elements in the plasmid.

Geoff Faulkner (University of Queensland)

Referee #3:

In their revised version of the manuscript, Ilik et al. have undertaken great efforts to add valuable data and close some of the earlier gaps in their study. This refers both to my comments raised and also to comments raised by the other reviewers as far as can judge it.

I highly appreciate the novel findings of SAFBs interacting with “opponents” of SR-proteins, while apparently competing with the very ones via the distinctive ability to interact with m6A-modified RNA. Mapping the interaction to a significantly confined region in the SAFB1 C-term is interesting and something that has been suggested earlier but never been shown with such a functional impact. Of note, this region appears somewhat different in SAFBs and may be a fine-tuning element.

Regarding the potential to recognize m6A-RNA (whilst per se the RRM shows relatively weak RNA binding), I may be allowed to add that potential homo- and hetero-oligomerization of SAFBs will possibly increase the effect in vivo, if so, as it may enrich the local concentration of SAFB RRMs.

Altogether, I feel the inclusion of new data justifies suggesting the manuscript for publication in this journal, given the following things will be added:

1) EMSAs of SAFB vs. TRA2B RRMs are convincing, but at least one replicate should be included. If EMSAs have been run in replicates, this should be stated.

We had several versions of these gels with each protein on its own gel (Halo alone, Halo-SAFB1-RRM and Halo-TRA2B-RRM) while setting up the experiment, thus confident that the results are robust and reproducible. We have however defrozen the reagents and repeated the experiment with the gel set up (see below), and the results look very reproducible. We now added the statement to the Figure 4 legend saying: “Gel images shown here are representative of two replicates.”

Figure 1 Repeat EMSA experiment gel images same conditions as in Fig. 4b

2) Further, as authors say themselves, one needs to treat this observation with care with respect to the situation in vivo, i.e. full-length proteins. At least SR proteins are known to exploit more than their RRM, and for SAFBs this is still not fully understood, but possible.

Could the authors use a simple pulldown using immobilized native and methylated RNA (like in the EMSA) to detect levels of TRA2b and SAFB(s) from full lysates? This will strengthen their conclusions even more and reveal that native fl-proteins incl. their possible PTMs would still differentially bind m6A-modified RNA as a proof of principle. I do see the challenge in trying to isolate individual RBPs from complex RNPs such that the findings may be less black-white than in EMSAs with purified domains. Nonetheless, I suggest including this experiment.

Also, not all SR proteins have purine-rich motifs as their consensus target sequences; or they can bind to combined motifs via tandem RRM (e.g. SRSF1). This does not per se weaken the findings and hypotheses, but may make aware of the need to distinguish SRSFs, as to my knowledge none of them has been tested in direct interaction with methylated RNAs so far. Just as a comment.

We thank the reviewer for this comment, and the suggestions. We tried this experiment by using a biotinylated L1 fragment (biotinylated at the 3'-end with a single biotin group) either unmethylated or N6-methylated which we have introduced through in vitro transcription, and used a nuclear extract prepared from HCT116 cells as the lysate. We could detect a remarkable enrichment of SAFB1 on methylated RNA, together with its interaction partners, as well as YTHDC1, which decided to include as a positive control, but we couldn't detect SLTM binding to either methylated or unmethylated RNA, perhaps due to its low expression levels. On the SR-proteins side, we could detect TRA2b enriched on the unmethylated RNA, though it is detectable on the methylated RNA as well, perhaps through SR-SR interactions with other SR-proteins. We could observe for example an enrichment of SRSF3 on the methylated RNA. SRSF1 seemed quite specific to unmethylated RNA, and we could see a single band on 1H4 blots at around 37kDa that appear also very specific to unmethylated RNA, though this could just be SRSF1. The results of this experiment are shown in Extended Data Fig. 7g.

As the reviewer alluded to, this experiment is not as black-and-white as the EMSAs since it includes many proteins, at unknown concentrations containing various PTMs that affect their interactions with the RNA and with each other, but we agree that it provides quite valuable additional information, and we thank again for this suggestion.

3) As minor edits I suggest:

- Ext. Data Fig. 1: Please double-check all domains! E.g. SRSF7 contains a ZnF domain (for which RNA binding is debated) and also U2AF1 should have one. I assume, all potential RNA-binding relevant domains (thus no SAPs?) are supposed to be listed as it is about domain architectures.

All checked again and updated. There were indeed several ZnF domains missing, including SRSF7 and U2AF1, we thank the reviewer for these suggestions. We tried to list all relevant RNA-binding domains, but some IDRs are still left out, because it is already a quite crowded figure with too many colors.

- Ext. Data Fig. 6a: Please define errors (in figure legend)! In panel f, the statistical background of the depicted should also be added (likely median of the triplicate?)!

Legend is updated (Extended Data Fig. 4a in this submission).

- For the IPs and WBs in Ext. Data Fig. 11, I assume no replicates have been made. This should be stated at least, while full views of the respective Blots will have to be included.

These were done twice, with slightly different set-ups (for example, just “mock” transfection instead of GFP as a control in panel e). Full blots are provided in the Supplementary Information with this submission.

- Ext. Data Fig. 12 (left): Please label y-axis!

Added (Extended Data Fig. 7f in this submission).

- Ext. Data Fig. 18b: Please define error bars in the legend!

Explanation is added to the legend (Extended Data Fig. 10h in this submission).

- I suggest working over the sentence ranging from lines 275-279, it reads very complicated to follow!

We removed the last part of the sentence to simplify it, we thank the reviewer for the suggestion.